# DIVIDE AND ABSTRACT: AUTOFORMALIZATION VIA DECOMPOSITION AND ABSTRACTION LEARNING

**Marcus J. Min**[1]  **Yeqi Gao**[2]  **Wilson Sy**[3]  **Zhaoyu Li**[3]  **Xujie Si**[3]  **Osbert Bastani**[1]

[1]University of Pennsylvania  [2]Tsinghua University  [3]University of Toronto

{marcmin,obastani}@seas.upenn.edu
six@cs.toronto.edu

## ABSTRACT

Existing approaches to autoformalization—the task of translating informal mathematics into formal machine-verifiable languages—rely heavily on pre-defined libraries and expect LLMs to directly generate complete formalizations. These approaches face three fundamental limitations: they are bottlenecked by existing abstractions, they have difficulty handling the complexity of realistic statements, and they do not transfer well across formal languages. We propose *Divide and Abstract* (*DNA*), a zero-training framework that addresses these challenges through a two-phase approach. First, *DNA* extracts common mathematical concepts from the entire corpus and formalizes them as reusable abstractions, extending the target language's capability. Second, *DNA* hierarchically decomposes new statements into structured informal clauses, translates each clause using the learned abstractions, and composes them into complete formalizations. Our evaluation on the LeanEuclidPlus and ProofNet-Hard benchmarks demonstrates consistent improvements across multiple model families, achieving up to $8.60\times$ performance gains over baselines. Notably, *DNA* enables smaller models to match baselines using much larger models, and shows particularly strong performance on complex mathematical statements requiring nested reasoning. Furthermore, our framework requires no training on target languages, making it effective for low-resource domain-specific languages. Our code is available at https://github.com/marcusm117/DNA.

## 1 INTRODUCTION

Autoformalization is the task of automatically translating informal mathematics into formal languages designed for theorem provers such as Lean (Moura & Ullrich, 2021) and Z3 (De Moura & Bjørner, 2008), so that theorems and proofs written in natural languages can be mechanically verified. The success of autoformalization will significantly reduce the amount of labor from human experts, and thus empower three transformative applications:

1. Synthesizing informal-formal parallel data for training neural theorem provers (Xin et al., 2024).

2. Grounding and guiding reasoning processes in natural languages (Yang et al., 2022).

3. Accelerating verification of important mathematical theorems (Gonthier et al., 2013; Hales et al., 2015) or engineering systems (Zhao et al., 2012; Reid et al., 2016) not only to identify potential gaps and mistakes, but also to facilitate certified extensions and future mathematical discovery.

Inspired by the breakthrough of Large Language Models (LLMs) in machine translation between natural languages, researchers have increasingly applied LLMs to autoformalization (Weng et al., 2025). Even though the ultimate vision is to automatically formalize entire theories that include axioms, definitions, notations, theorems, and proofs, current LLMs struggle to even formalize an individual statement (Murphy et al., 2024; Liu et al., 2025a).

Previous work in LLM-based statement autoformalization has explored in-context learning (Wu et al., 2022), fine-tuning specialized autoformalizer models (Jiang et al., 2023a), invoking retrieval before LLM inference (Liu et al., 2025a), or employing post-inference techniques such as majority

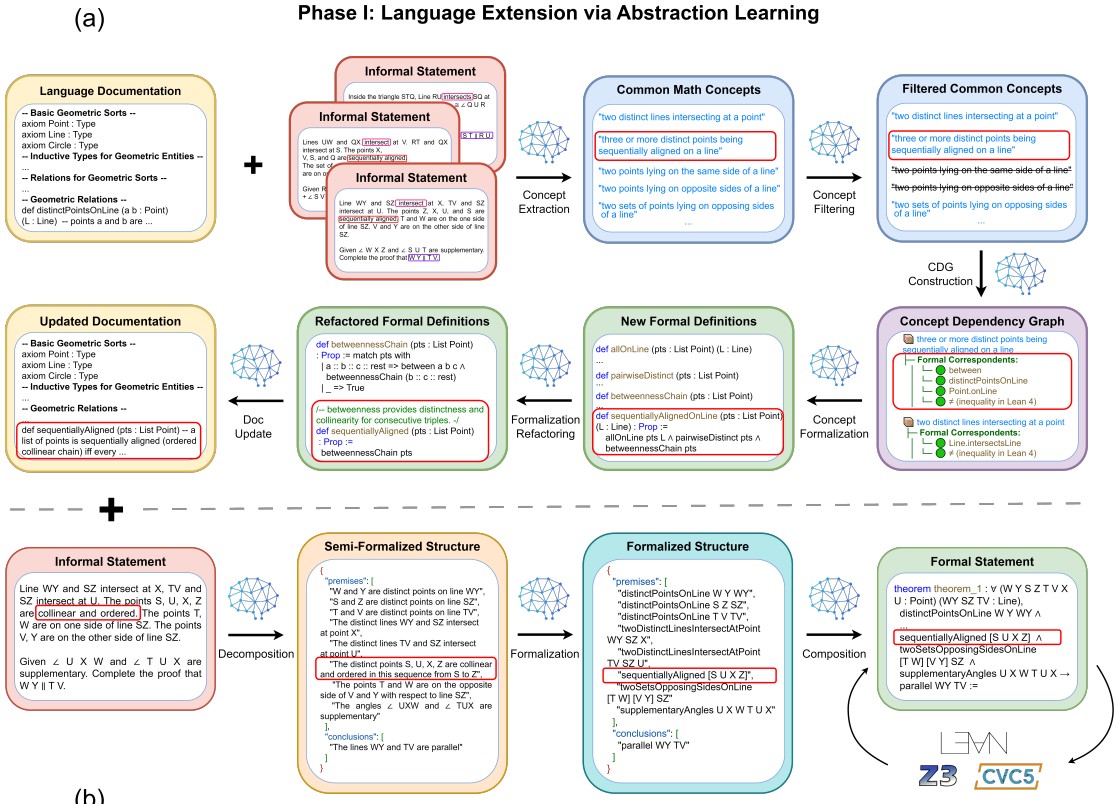

Figure 1: Overview of the *Divide and Abstract* (*DNA*) framework. **(a) Phase I (Language Extension):** Given an informal corpus, we extract common mathematical concepts, filter out those already present in the target language, construct a concept dependency graph (CDG), and systematically formalize concepts following topological ordering to extend the DSL. **(b) Phase II (Statement Formalization):** For each statement, we hierarchically decompose it into semi-formalized components, translate each component using the learned abstractions, compose them into complete formal statements, and refine using symbolic validator feedback.

voting and semantic consistency ranking by an embedding model (Li et al., 2024a). However, all these approaches suffer from limited generalizability due to three fundamental challenges:

**1. Bottlenecked by Existing Abstractions:** The performance of formalizing the same informal statement is limited by the richness of existing abstractions in the target formal language. For example, if the target language only contains basic formal definitions of sets and real numbers, then formalizing the relation "a topological space being a manifold" would be difficult even for human experts. In contrast, if the target language already provides a formal relation "isManifold", then the translation is simply a one-to-one mapping. However, all existing methods rely heavily on pre-defined abstractions in existing libraries like Mathlib in Lean (Mathlib Community, 2020) and HOL sessions in Isabelle (Nipkow et al., 2002), which fail to generalize to the cases where the target language doesn't provide an extensive coverage of high-level abstractions.

**2. Difficulty Handling Complex Statements:** Regardless of the availability of relevant abstractions, the difficulty of formalization also grows proportionally to the complexity of statements. For example, formalizing a simple statement like "function $f$ is continuous" is straightforward when the formal definition of continuity exists. However, formalizing a nested statement such as "the space of continuous functions from a compact manifold to a Banach space forms a complete metric space under the supremum norm" requires understanding and correctly composing multiple layers of mathematical abstractions. Nonetheless, all existing approaches expect the LLMs to directly generate the final formalization at inference time, which does not generalize well to complex statements that involve nested quantifiers, higher-order objects, and composite relations.

**3. Limited Transfer Across Formal Languages:** Orthogonal to abstraction availability and statement complexity, the choice of target formal language constitutes a third dimension of the generalizability challenge. Most fine-tuned autoformalizer models (Gao et al., 2025; Wang et al., 2025; Xuejun et al., 2025) are heavily trained on Lean Mathlib and formal statements that directly use Mathlib abstractions. However, as shown in Table 3, these models perform poorly when the target is a domain-specific language (DSL) that is not in the training data, or even when Mathlib evolves to newer versions with updated syntax and library structures.

To address these limitations in generalizability, we propose an end-to-end framework *Divide and Abstract (DNA)* with three key designs that directly correspond to each challenge:

**1. Abstraction Learning:** While previous methods treat the formalization of each statement independently, we leverage the insight that statements within the same corpus often share common mathematical concepts like definitions of objects and relations. As shown in Figure 1(a), given the current language documentation and a target informal corpus, *DNA* first extracts these shared concepts from the entire corpus and formalizes them as a collection of reusable formal abstractions. This phase essentially extends the target language by enriching the library of available abstractions, thereby facilitating the statement formalization both within and beyond the corpus.

**2. Hierarchical Decomposition:** Unlike existing approaches that let LLMs directly generate the formal statement, we take advantage of the fact that mathematical statements are semi-structured data—each statement consists of quantifiers and a body, and the body is broken down into a list of premises and a list of conclusions; further, quantified statements nested in the premises or conclusion can be recursively decomposed in the same manner. As shown in Figure 1, *DNA* hierarchically decomposes a statement into a structure of informal clauses, translates each clause into its formal correspondents, and composes the formal clauses back into a complete formalization. Since we have learned a rich library of reusable abstractions, the formalization of each individual clause becomes significantly more tractable than formalizing the entire complex statement at once.

**3. Zero-Training Framework:** *DNA* is designed to be a zero-training, plug-and-play framework. This is achieved by simply providing the target language's documentation in the context of the LLM. After the abstraction learning phase, the documentation is updated with the newly learned formal definitions, and then passed as context to the statement formalization phase, conditioning the LLM on both the original language specifications and a library of directly usable high-level definitions. The entire framework requires no training on the target formal language, and is thus particularly effective for low-resource DSLs.

We evaluate *DNA* on LeanEuclidPlus and ProofNet-Hard against fine-tuned autoformalizers and general-purpose LLMs including GPT-4.1/5, Claude-4-Sonnet, and Qwen3 variants. *DNA* consistently outperforms baselines across all model families, with the most dramatic improvement being Qwen3-14B advancing from 1.0 to 9.6 success rate ($9.6\times$ gain), while GPT-4.1-mini improves from 4.8 to near GPT-4.1 levels. On ProofNet-Hard, all baselines achieve zero success, whereas *DNA* enables successful formalization, demonstrating its effectiveness for domain-specific languages lacking extensive training data. To summarize, our contributions are as follows:

**(1)** identifying three generalizability challenges in autoformalization, **(2)** proposing a zero-training framework combining abstraction learning and hierarchical decomposition, **(3)** demonstrating $1.17\times$ to $9.6\times$ improvements across diverse models and benchmarks, and **(4)** enabling effective transfer to challenging domain-specific languages where specialized models fail entirely.

## 2 THE *Divide and Abstract* FRAMEWORK

### 2.1 PHASE I: LANGUAGE EXTENSION

As illustrated in Figure 1(a), the language extension process unfolds through six coordinated steps.

**Step 1: Concept Extraction.** The first step of our abstraction learning pipeline aims to extract mathematical concepts that are non-ambiguous, well-defined, and abstract, as specified in the prompt guidelines detailed in Appendix D.1.1. These extracted concepts encompass three categories: mathematical objects, relations between objects, and functions mapping objects to objects. Our extraction process enforces precision by specifying argument types and counts, well-definedness by ensuring

| Step 1: Concept Extraction | | |
|---|---|---|
| **Run** | **Recall** | **Correctness** |
| Average | 94.44% | 100.00% |

| Step 2: Concept Filtering | |
|---|---|
| **Run** | **Correctness** |
| Average | 100.00% |

| Step 3: CDG Construction | |
|---|---|
| **Run** | **Correctness** |
| Average | 100.00% |

| Step 4: Concept Formalization | |
|---|---|
| **Run** | **Correctness** |
| Average | 97.87% |

| Step 5: Refactoring | | |
|---|---|---|
| **Run** | **Correctness** | **Compression** |
| Average | 99.15% | 1.59 |

| Step 6: Documentation | | |
|---|---|---|
| **Correctness** | **Qwen3-235B** | **GPT-5 mini** |
| 100.00% | 97.4% | 99.8% |

Table 1: Performance Analysis of each step in Phase I Abstraction Learning, averaged across 5 runs on LeanEuclidPlus. Please refer to Appendix C.1 for detailed analysis of qualitative examples on both LeanEuclidPlus and ProofNet-Hard.

conventional mathematical validity, and abstractness by avoiding particular variable names or specific instances.

We evaluate the quality of this extraction step using two metrics reported in Table 1, both averaged across 5 runs. The correctness metric measures what percentage of the extracted concepts are indeed non-ambiguous, well-defined, and abstract out of all extracted concepts. The recall metric quantitatively analyzes how many concepts from the oracle abstractions written by human experts are successfully identified and extracted by our system. Both metrics are evaluated manually because the output of this step consists of freeform natural language descriptions, and no reliable automated evaluation alternative exists for assessing the semantic correctness.

**Step 2: Concept Filtering.** The second step of our abstraction learning pipeline performs concept filtering to remove duplicates and concepts already formalized in the target language, as specified by the guidelines in Appendix D.1.2. The filtering process leverages LLM understanding of both the extracted concepts and existing DSL capabilities to identify gaps in abstraction coverage.

We evaluate this filtering step using the correctness (precision) metric reported in Table 1 for the filtering step, which measures what percentage of filtered-out concepts are indeed duplicates or already-formalized out of all filtered-out concepts, averaged across 5 runs. We deliberately focus on precision rather than recall for two important reasons. First, in the abstraction learning phase, our priority is to formalize reusable abstractions without missing potentially useful concepts, making filtering precision naturally more important than recall. Second, borderline cases exist where a natural language concept does not directly correspond to exactly one formal definition but can be easily formalized by slightly adapting an existing definition. For instance, the concept of two triangles sharing a common vertex in Euclidean space has no direct corresponding formal definition, but it can be straightforwardly expressed by specifying the three vertices of each triangle. We design the filtering step to remove only those cases that are absolutely obvious duplicates or already-formalized concepts, erring on the side of caution to preserve potentially useful abstractions. The correctness metric is evaluated manually for the same reason as Step 1.

**Step 3: Concept Dependency Graph (CDG) Construction.** The third step of our abstraction learning pipeline constructs concept dependency graphs that analyze how each extracted concept can be formalized using existing formal definitions and other concepts, following the guidelines in Appendix D.1.3. The depth and complexity of these dependency graphs vary across benchmarks and concepts. For each concept, we analyze direct expressibility using existing DSL elements, prerequisite concept dependencies, and formalization feasibility. The CDG contains two node types: leaf nodes that are either directly formalizable or impossible to formalize within the target language scope, and parent nodes that depend on other concepts. Formalization follows topological ordering by starting with leaf nodes, then proceeding layer by layer up the dependency hierarchy. Algorithm 1 details the construction process. Notably, CDG construction inherently performs concept decomposition, demonstrating that abstraction and decomposition are complementary, with neither achieving optimal performance in isolation (Table 3).

---

**Algorithm 1** Concept Dependency Graph (CDG) Construction Algorithm.

---

**Input:** Filtered Concepts $C$, Current Language Documentation $D$, LLM $M$
**Output:** Concept Dependency Graph $G$ containing all concepts in $C$ and their dependencies
 1: Initialize the analysis result dict $\mathcal{A} \leftarrow \emptyset$
 2: Initialize the to-be-analyzed queue $\mathcal{Q} \leftarrow C$
 3: Initialize the have-analyzed set $\mathcal{S} \leftarrow \emptyset$
 4: **while** $\mathcal{Q} \neq \emptyset$ **do**
 5:     $c \leftarrow \mathcal{Q}$.dequeue()                     ▷ fetch the next to-be-analyzed concept
 6:     Analyze concept $c$ using LLM $M$ given context $(D, \mathcal{A})$, determine the dependency status of $c$, which can is one of: "directly_expressible", "needs_dependencies", or "impossible"
 7:     $\mathcal{A}[c] \leftarrow$ analysis result                     ▷ save analysis result
 8:     $\mathcal{S} \leftarrow \mathcal{S} \cup \{c\}$                    ▷ add $c$ to the have-analyzed set
 9:     **if** $c$ has status "needs_dependencies" **then**
10:         **for** each dependency $d$ in the dependency list of $c$ **do**
11:             **if** $d \notin \mathcal{S}$ and $d \notin \mathcal{Q}$ **then**
12:                 $\mathcal{Q} \leftarrow \mathcal{Q} \cup \{d\}$         ▷ add $d$ to the to-be-analyzed queue
13: **return** $\mathcal{A}$

---

We evaluate this CDG construction step using the correctness metric reported in Table 1, which measures the percentage of the dependency analyses for each concept that are correctly constructed out of all dependency analyses, averaged across 5 runs. The correctness metric is evaluated manually for the same reason as Steps 1 and 2.

**Step 4: Concept Formalization.** The fourth step of our abstraction learning pipeline translates the concepts identified in the CDG into actual formal definitions in the target language, following the guidelines in Appendix D.1.4. This step involves generating both the main target definitions and any necessary helper definitions to support them. The formalization process adheres to target language conventions and ensures syntactic correctness through iterative refinement.

We evaluate this formalization step using the correctness metric reported in Table 1, which measures what percentage of formal definitions are correct out of all formal definitions, with helper definitions counted separately (essentially counting the number of function definitions), averaged across 5 runs. Importantly, concepts do not need to be formalized by strictly following the previously analyzed concept dependency graph; a formalization is considered correct as long as it can be interpreted as faithful to the original natural language concept. The correctness metric is evaluated manually despite the output being in formal language because the same natural language concept can be formalized in multiple ways that are all acceptable to human experts. As discussed in Section 3, no symbolic equivalence checker is complete, and some are not even sound, necessitating manual evaluation by human experts to assess faithfulness.

**Step 5: Formalization Refactoring.** The fifth step of our abstraction learning pipeline performs formalization refactoring to eliminate redundant definitions and extract common helper definitions, thereby creating a more concise and general library of learned abstractions that downstream users, whether human or language models, can better understand and employ. This refactoring process follows the guidelines specified in Appendix D.1.5.

We evaluate this refactoring step using two metrics reported in Table 1. The correctness metric measures what percentage of refactored formal definitions are correct out of all definitions that were refactored, with helper definitions counted separately (essentially counting the number of function definitions), averaged across 5 runs. Additionally, we quantitatively analyze the compression ratio, which is the ratio of the number of formal definitions before refactoring to the number after refactoring, averaged across 5 runs. This compression ratio metric is important because it indicates how effectively common helper functions are reused and how many unnecessary formal definitions are eliminated. A high compression ratio facilitates better understanding and utilization of the learned abstractions by both humans and language models, as it reduces cognitive load and clarifies the essential conceptual structure. The correctness metric is evaluated manually for the same reason as Step 4, since multiple semantically equivalent refactorings may be acceptable to human experts.

| Error Type | Qwen3-235B | GPT-5 |
|---|---|---|
| Unfaithful Variable Name | 1 | 1 |
| Validation Error | 8 | 2 |
| Stronger/Weaker Translation | 10 | 17 |
| Incorrect Translation | 11 | 8 |

Table 2: Error analysis of 200 failed formalization attempts, where 100 are from Qwen3-235B and another 100 from GP5-5. The numbers shown are the count for each error type. Four main error categories were identified.

**Step 6: Documentation Update.** The sixth and final step of our abstraction learning pipeline generates clear and concise natural language documentation for the learned abstractions, abstracted away from low-level implementation details such as internal helper definitions, to serve both human and language model users. This step is critical because excessive definitions in the documentation would inevitably confuse both language models as the context grows and human users attempting to understand and use the learned abstractions. The documentation generation process follows the guidelines specified in Appendix D.1.6.

We evaluate the documentation update step using quantitative metrics reported in Table 1. The correctness metric measures what percentage of documented formal definitions correctly convey the meaning of the definition without missing any essential details or introducing ambiguity, out of all documented definitions, averaged across 5 runs. We also evaluate downstream task performance on LeanEuclidPlus using two strong models, Qwen3-235B and GPT-5, with our four-stage hierarchical decomposition pipeline. Under the hypothesis that good documentation should enable strong models to generate syntactically correct formalizations with high reliability, we report the compilation rate of the two models averaged across 5 runs. For Qwen3-235B, the maximum compilation rate is 98.2%, minimum is 96.8%, mean is 97.4%, and standard deviation is 0.47%; for GPT-5, the maximum is 100.0%, minimum is 99.0%, mean is 99.8%, and standard deviation is 0.45%. These consistently high compilation rates with low variance validate the quality and stability of the generated documentation.

For Phase I language extension, we employ Qwen3-235B-Instruct for concept extraction, filtering, and dependency graph construction on LeanEuclidPlus, while using GPT-5 with high reasoning effort for ProofNet-Hard. For the more complex tasks of concept formalization, refactoring, and documentation generation, we utilize GPT-5 with high reasoning effort across both benchmarks to ensure high-quality formal definitions and coherent abstraction libraries. In Table 1, we show how error in language extension decomposes across steps.

## 2.2 PHASE II: STATEMENT FORMALIZATION

Building upon the extended DSL from Phase I, Phase II employs hierarchical decomposition to formalize individual statements. Unlike existing approaches that expect LLMs to directly generate complete formalizations, our framework systematically breaks down complex statements into manageable components that can be formalized using the learned abstractions.

**Motivation and Error Analysis.** To motivate our decomposition approach, we conducted an error analysis on 200 failed formalization attempts (100 each from Qwen3-235B and GPT-5) using baseline methods without our pipeline. As shown in Table 2, we identified four primary error categories. Validation errors include syntax issues and logical consistency violations that prevent successful compilation. Stronger or weaker translation errors represent semantic misalignment where formalizations are either too restrictive or too permissive compared to the English statement. Incorrect translation errors involve wrong formal correspondents or parameters for relations. Unfaithful variable naming creates inconsistency between variable names in formal and natural language. This analysis reveals distinct error patterns across models, motivating our decomposition approach that systematically addresses each error type.

**Solution for Each Error Type.** Our four-step decomposition pipeline systematically addresses each significant error type, excluding "Unfaithful Variable Name", through targeted design choices. Step 1 decomposes complex informal statements into semi-formalized structures with explicit quantifications, premises, and conclusions, ensuring logical scope precisely matches the informal state-

ment and addressing stronger/weaker translation errors. Step 2 translates each component using learned abstractions from Phase I, enabling focused attention on correct formal correspondents and parameter assignments for individual clauses rather than managing entire statement complexity, thereby addressing incorrect translation errors. For instance, in LeanEuclidPlus, the relation `formTriangle` requires specific parameter ordering where points $a, b$ must lie on line $l_1$, points $b, c$ on line $l_2$, and points $a, c$ on line $l_3$—our clause-by-clause approach helps prevent logical contradictions from incorrect parameter ordering.

Step 3 systematically composes formalized components back into complete formal statements while ensuring proper quantifier scoping and logical consistency, reducing validation errors through structured composition that maintains clear relationships between components. Step 4 employs comprehensive self-refinement with symbolic validator feedback that performs three checks: syntax correctness, variable name faithfulness between formal and natural language representations, and logical consistency verification, ensuring premises are non-contradictory and conclusions are nontrivial. Unlike existing approaches that rely solely on compiler feedback, our validator provides targeted semantic feedback to guide iterative LLM refinement, directly addressing validation errors and unfaithful variable naming until the formalization meets all quality criteria.

## 3 EXPERIMENTS

**Benchmarks.** We evaluate *DNA* on two benchmarks: LeanEuclidPlus and ProofNet-Hard. LeanEuclidPlus adapts LeanEuclid (Murphy et al., 2024), implementing a formal Euclidean geometry system (Avigad et al., 2009) as a domain-specific language (DSL). The scarcity of training data for LeanEuclid-style statements makes it ideal for evaluating autoformalization generalizability across DSLs. We refined the 100 UniGeo problems from LeanEuclid and added 40 hand-crafted statements with greater geometric complexity.

Our main evaluation uses the 100 core problems for end-to-end pipeline assessment, while the additional 40 problems test generalizability of learned abstractions and decomposition-driven formalization. ProofNet-Hard comprises 19 challenging ProofNet (Azerbayev et al., 2023; Vishwakarma et al., 2024) statements requiring auxiliary helper definitions beyond standard Mathlib imports. As Table 3 shows, when helper definitions are withheld, both specialized autoformalizers—despite ProofNet training data (Wang et al., 2025; Xuejun et al., 2025)—and state-of-the-art models like GPT-5 and Claude-4-Sonnet achieve zero success rates.

**Evaluation.** Previous work uses LLM backtranslation and semantic equivalence judgment (Ying et al., 2025; Gao et al., 2025; Wang et al., 2025; Xuejun et al., 2025; Liu et al., 2025b), which lacks soundness (Liu et al., 2025a). We selected benchmarks with symbolic equivalence checkers for rigorous evaluation. For ProofNet-Hard, we use the BEq+ checker (Poiroux et al., 2025), which provides semantic equivalence verification through bidirectional entailment checking for complex logical structures. For LeanEuclidPlus, we enhanced the original E3 symbolic checker with comprehensive three-stage pre-checks to eliminate false positives and separate bidirectional verification of premises and conclusions to minimize false negatives. We report pass@1 accuracy based on successful symbolic verification.

**Models.** We evaluate across fine-tuned autoformalizers (Kimina (Wang et al., 2025), Mathesis (Xuejun et al., 2025)) and three major model families: GPT (OpenAI, 2024),[1] Claude (Anthropic, 2024),[2] and Qwen (Yang et al., 2025),[3] each with standard and reasoning variants. We sample 5 runs per problem with temperature 0.2 for non-reasoning models/modes and 1.0 for reasoning models/modes. Specialized autoformalizers use 1024 tokens, other models use 6144 tokens, and reasoning models have 12,288 token budgets. All models receive standardized 1-shot examples demonstrating the complete pipeline (see Appendix B for ablation).

**Baselines.** Since *DNA* consists of two phases that can be executed individually, we design a systematic ablation study to evaluate each component's contribution. Our baseline represents the standard autoformalization approach used in prior work, where models receive no corpus-specific abstractions

---

[1]GPT models: gpt-4.1-mini-2025-04-14, gpt-4.1-2025-04-14, gpt-5-mini-2025-08-07, gpt-5-2025-08-07

[2]Claude models: Claude-4-Sonnet-20250514

[3]Qwen models: Qwen3-14B, Qwen3-32B, Qwen3-235B-A22B-Instruct-2507, Qwen3-235B-A22B-Thinking-2507

| Model | LeanEuclidPlus | | | | | | ProofNet-Hard | | | | | |
|---|---|---|---|---|---|---|---|---|---|---|---|---|
| | Baseline | Divide | Abstract | DNA | OracleA | DNOracleA | Baseline | Divide | Abstract | DNA | OracleA | DNOracleA |
| Fine-tuned Models | | | | | | | | | | | | |
| Kimina Autoformalizer 7B | 0.0 | 0.0 | 0.0 | 0.0 | 0.0 | 0.0 | 0.0 | 0.0 | **8.4** | 1.1 | 8.4 | 7.4 |
| Mathesis Autoformalizer 7B | 0.0 | 0.0 | 0.0 | 0.0 | 0.0 | 0.0 | 0.0 | 0.0 | **5.3** | **5.3** | 7.4 | 5.3 |
| Non-Reasoning Models | | | | | | | | | | | | |
| GPT-4.1 mini | 4.8 | 25.4 | 6.2 | **42.4** ↑7.83× | 3.0 | 48.8 ↑9.17× | 0.0 | 0.0 | 5.3 | **7.4** | 10.5 | 13.7 |
| GPT-4.1 | 26.8 | 35.2 | 28.6 | **48.4** ↑80.6% | 34.2 | 61.2 ↑1.28× | 0.0 | 0.0 | 0.0 | **10.5** | 14.7 | 23.2 |
| Claude 4 Sonnet | 34.2 | 46.6 | 36.2 | **58.2** ↑70.2% | 36.6 | 60.4 ↑76.6% | 0.0 | 0.0 | **14.7** | 12.6 | 12.6 | 25.3 |
| Qwen3 14B | 1.0 | 5.0 | 4.8 | **9.6** ↑8.60× | 3.0 | 13.4 ↑12.40× | 0.0 | 0.0 | 1.1 | **3.2** | 0.0 | 0.0 |
| Qwen3 32B | 4.6 | 17.7 | 4.0 | **20.6** ↑3.48× | 9.2 | 26.4 ↑4.74× | 0.0 | 0.0 | 1.1 | **4.2** | 5.3 | 4.2 |
| Qwen3 235B Instruct 2507 | 40.8 | 43.6 | 45.4 | **64.4** ↑57.8% | 55.4 | 71.4 ↑75.0% | 0.0 | 0.0 | 1.1 | **9.5** | 2.1 | 16.8 |
| Average | 18.7 | 28.9 | 20.9 | **40.6** ↑1.17× | 23.6 | 46.9 ↑1.51× | 0.0 | 0.0 | 3.9 | **7.9** | 7.6 | 13.9 |
| Reasoning Models | | | | | | | | | | | | |
| GPT-5 mini | 25.8 | 32.0 | 47.2 | **58.2** ↑1.26× | 58.0 | 71.2 ↑1.76× | 0.0 | 0.0 | 8.4 | **10.5** | 19.0 | 22.1 |
| GPT-5 | 35.4 | 37.0 | 53.2 | **55.8** ↑57.6% | 60.2 | 68.4 ↑93.2% | 0.0 | 0.0 | 12.6 | **15.8** | 12.6 | 28.4 |
| Claude 4 Sonnet (Thinking) | 32.8 | 45.4 | 42.0 | **57.4** ↑75.0% | 50.8 | 64.4 ↑96.3% | 0.0 | 0.0 | 11.6 | **12.6** | 10.5 | 30.5 |
| Qwen3 14B (Thinking) | 25.4 | 27.8 | 33.8 | **38.6** ↑52.0% | 33.0 | 51.6 ↑1.03× | 0.0 | 0.0 | **1.1** | **1.1** | 4.2 | 9.5 |
| Qwen3 32B (Thinking) | 29.6 | 31.6 | 39.4 | **40.8** ↑37.8% | 46.8 | 58.2 ↑96.6% | 0.0 | 0.0 | **2.1** | **2.1** | 8.4 | 14.7 |
| Qwen3 235B Thinking 2507 | 40.2 | 45.4 | 41.6 | **55.6** ↑38.3% | 58.4 | 70.4 ↑75.1% | 0.0 | 0.0 | **5.3** | **5.3** | 9.5 | 19.0 |
| Average | 31.5 | 36.5 | 42.9 | **51.1** ↑62.2% | 42.9 | 64.0 ↑1.03× | 0.0 | 0.0 | 6.8 | **7.9** | 10.7 | 20.7 |

Table 3: Performance comparison across model families and experimental conditions on LeanEuclidPlus and ProofNet-Hard benchmarks. Results show pass@1 accuracy with improvements over baseline highlighted in red. DNA consistently outperforms individual components (Divide, Abstract) and often surpasses oracle conditions (OracleA).

and directly generate formal statements without decomposition. While all models are prompted to provide step-by-step reasoning, the critical distinction lies in whether we explicitly prompt the LLM to decompose statements into our semi-formalized hierarchical structure.

To isolate the contribution of each phase, we systematically reintroduce individual components. The "Divide" condition activates only Phase II of our framework, providing models with our 4-step statement autoformalization pipeline that includes hierarchical decomposition and self-refinement, while operating with the original DSL documentation without any corpus-specific abstractions. Conversely, the "Abstract" condition activates only Phase I, enriching the DSL with reusable definitions learned from the corpus, but requires models to generate formal statements directly without the benefit of our decomposition pipeline. Our complete "DNA" method combines both phases, providing models with both the corpus-specific abstractions from Phase I and the hierarchical decomposition capabilities from Phase II.

To establish theoretical upper bounds and validate our approach, we include oracle conditions using human expert-written corpus-specific definitions for both benchmarks. The "OracleA" condition provides these oracle abstractions to downstream models without activating our 4-step statement autoformalization pipeline, effectively testing the potential ceiling of perfect abstraction learning. The "DNOracleA" condition combines these oracle abstractions with our complete 4-step statement autoformalization pipeline, establishing the theoretical performance ceiling for our entire approach.

## 4 RESULTS

### 4.1 EFFECTIVENESS OF *DNA*

Our analysis of Table 3 reveals several key insights about the effectiveness and generalizability of our framework across different model types and benchmark complexities.

The DNA framework demonstrates substantial performance improvements across all model categories and benchmarks. On LeanEuclidPlus, the most dramatic improvement is Qwen3-14B advancing from 1.0 to 9.6 (9.6× gain), while GPT-4.1-mini improves from 4.8 to performance levels approaching GPT-4.1. On ProofNet-Hard, baseline performance across all models is zero, making DNA's ability to enable successful formalization particularly striking, especially considering that even GPT-5 with high reasoning effort—the same model used for Phase I language extension—fails completely in baseline scenarios without corpus-specific abstractions.

The framework proves especially beneficial for smaller non-reasoning models, demonstrating remarkable scalability advantages. The most notable improvement occurs with Qwen3-14B, which advances from merely 1.0 success rate at baseline to 9.6 with DNA augmentation, representing nearly a

| Model | Divide | Abstract | DNA |
|---|---|---|---|
| Qwen3-235B-A22B-Instruct-2507 | 13.5 | 25.0 | 47.0 |
| gpt-5-mini-2025-08-07 | 9.5 | 47.5 | 57.5 |

Table 4: Ablation on 40 complex LeanEuclidPlus problems.

10× performance gain. Similarly, GPT-4.1-mini improves from 4.8 at baseline to performance levels approaching GPT-4.1, constituting a transformative enhancement for resource-constrained applications. Reasoning models exhibit smaller relative improvements compared to non-reasoning models, which aligns with their inherent training for generating extended reasoning traces and decomposing complex problems autonomously. However, comparing baseline and "Divide" columns reveals that our decomposition still provides substantial benefits even for reasoning-capable models, as detailed in the ablation analysis presented in Appendix A.

Fine-tuned specialized autoformalizer models demonstrate complete failure to generalize across domain-specific languages, achieving zero success rates on LeanEuclidPlus due to their inability to follow instructions for generating correct LeanEuclid syntax and performing decomposition operations, failing even basic compilation syntax checks. On ProofNet-Hard, despite Mathesis Autoformalizer (Xuejun et al., 2025) being initialized from Kimina Autoformalizer (Wang et al., 2025) and Kimina being trained on ProofNet data, both models still achieve zero baseline performance. Notably, for specialized models, the "Abstract" condition outperforms "DNA" because these models cannot follow decomposition instructions effectively, and the decomposition prompts actually impede their performance. These findings conclusively demonstrate that fine-tuning specialized autoformalizer models lack generalizability, and effective generalization requires DNA framework integration with models possessing strong instruction-following capabilities.

## 4.2 GENERALIZABILITY OF *DNA*

To assess the generalizability of both our learned abstractions and decompositional statement autoformalization approach, we evaluate Qwen3-235B-Instruct and GPT-5-mini, representing the best-performing non-reasoning and reasoning models respectively on the 100 core LeanEuclidPlus problems as presented in Table 4, to determine whether our main results extend to the 40 additional problems with greater diagrammatic complexity. The generalizability results demonstrate consistent performance patterns across varying complexity levels, with DNA (Learned+4-stage) maintaining substantial advantages over both Abstract (Learned+1-stage) and Divide (Barebone+4-stage) approaches even on more challenging geometric problems, confirming the robustness and scalability of our integrated framework design.

## 4.3 SYNERGY BETWEEN DECOMPOSITION AND ABSTRACTION

Analysis of Table 3 reveals a consistent synergistic relationship between our two framework phases across all evaluated models and benchmarks. The complete DNA framework almost invariably outperforms both Divide (D) and Abstract (A) components when applied in isolation, demonstrating that the combination of abstraction learning and hierarchical decomposition yields performance gains that exceed the sum of their individual contributions. Remarkably, DNA frequently surpasses even the OracleA condition, which provides human expert-written abstractions, indicating that our learned abstractions coupled with systematic decomposition can match or exceed the effectiveness of carefully crafted human-designed abstraction libraries.

These empirical findings validate our theoretical framework presented in Section 2, confirming that decomposition and abstraction represent complementary facets of the autoformalization challenge rather than independent optimization targets. The synergistic effect emerges because abstraction learning enriches the target language with high-level mathematical concepts that facilitate more accurate clause-level translations during decomposition, while hierarchical decomposition exposes the precise semantic structure needed to effectively utilize these learned abstractions, creating a mutually reinforcing system that achieves optimal formalization performance.

## 5 RELATED WORK

**Autoformalization.** The field of autoformalization has advanced from early rule-based systems to modern LLM-based paradigms (Weng et al., 2025). While proof autoformalization has been employed in neural theorem proving (Jiang et al., 2023b) and LLM reasoning verification (Zhou et al., 2024), the lack of faithful automated evaluation metrics has led most research to focus on statement autoformalization. Recent LLM-based approaches include in-context learning (Wu et al., 2022; Azerbayev et al., 2023), supervised fine-tuning (Jiang et al., 2023a; Ying et al., 2025; Gao et al., 2025), reinforcement learning (Huang et al., 2025; Xuejun et al., 2025; Wang et al., 2025), retrieval-augmented generation (Zhang et al., 2024; Liu et al., 2025a), and post-inference sampling techniques like majority voting (Li et al., 2024a) and compiler feedback (Poiroux et al., 2025). However, all aforementioned works let LLMs directly generate formal statements from the informal and treat each statement in isolation, thus overlooking two important insights: (1) statements within the same corpus often share common mathematical concepts that can be abstracted and reused; (2) mathematical statements are inherently semi-structured and can be hierarchically decomposed into simpler, more manageable clauses. Our framework leverages both insights by first formalizing common concepts in the entire corpus into reusable abstractions, and then hierarchically decomposing individual statements into clauses that are formalized using the learned abstractions.

**Decomposition.** Decomposition, the principle of dividing complex problems into more tractable subtasks, is widely acknowledged in LLM reasoning (Huang & Chang, 2023). Theoretical works show that problems are more efficiently learned when decomposed (Wies et al., 2023), and generating step-by-step solutions enables LLMs to tackle increasingly complex tasks (Li et al., 2024b). This principle is empirically validated across various diverse domains (Shwartz et al., 2020; Nye et al., 2021; Wei et al., 2023; Zelikman et al., 2023) from commonsense reasoning to code generation. In neural theorem proving, decomposition has been successfully adopted to break proof goals into simpler lemmas (Wang et al., 2023; Zhao et al., 2023; Wang et al., 2024a). However, in the context of autoformalization, decomposition remains largely unexplored. Only one recent work (Xuejun et al., 2025) uses decomposition for evaluation purposes: an LLM judge decomposes the informal statement into premises and conclusions, and then assesses the semantic equivalence between the decomposed informal statement and the predicted formalization. In contrast, our framework is the first to apply decomposition directly to the formalization process itself.

**Abstraction Learning.** Abstraction learning aims to automatically extract reusable knowledge from data for application to future tasks. In program synthesis, library learning focuses on extracting reusable subroutines from program corpora (Ellis et al., 2021; Wang et al., 2024b). In theorem proving, systems learn libraries of reusable lemmas to simplify future proofs (Zhou et al., 2022; Johansson et al., 2014; Kurashige et al., 2024; Singher & Itzhaky, 2021). In LLM reasoning, tool learning enables models to create reusable tools for domain-specific tasks (Yuan et al., 2024; Qu et al., 2025). Our work introduces abstraction learning to autoformalization, where the goal is to curate libraries of reusable mathematical definitions and relations. While program synthesis, theorem proving, and tool learning target executable abstractions (subroutines, sub-proofs, tools), autoformalization requires mathematical definitions that are usually non-computable, like axioms defining algebraic structures. Furthermore, unlike library learning, which requires large corpora of formal statements, our framework learns abstractions directly from natural language. To our knowledge, this is the first work to apply abstraction learning to statement autoformalization.

## 6 CONCLUSION

We have presented DNA, a novel framework that addresses the three fundamental challenges limiting autoformalization generalizability through corpus-driven abstraction learning and hierarchical statement decomposition. Our approach demonstrates remarkable performance improvements, with gains ranging from 1.17× to 8.60× over baseline methods across diverse model architectures and domain-specific languages. Most significantly, DNA's zero-training design enables smaller models like Qwen3-14B to achieve performance comparable to much larger baselines, while completely transforming autoformalization for challenging domains where specialized models fail entirely. The framework's success suggests a fundamental shift toward more generalizable autoformalization approaches that can adapt to new mathematical domains without extensive retraining, opening pathways for automated mathematical reasoning across previously intractable formal language targets.

ACKNOWLEDGEMENT

This work was supported in part by an AWS ASSET Ph.D. Fellowship, OpenAI Researcher Access Program, NSF Award CCF-2338777, Amazon Research Award Fall 2023, Amazon/ASSET Gift for Research in Trustworthy AI, Individual Discovery Grants from the Natural Sciences and Engineering Research Council of Canada, and the Microsoft Accelerate Foundation Models Research (AFMR) grant program. Any opinions, findings, conclusions, or recommendations expressed herein are those of the authors and do not necessarily reflect the views of funding entities.

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

| Model | LeanEuclidPlus | | | |
| --- | --- | --- | --- | --- |
| | 1-Stage Direct | 2-Stage Self-Refine | 3-Stage Decomoposition | 4-Stage Decomposition |
| Non-Reasoning Models | | | | |
| gpt-4.1-mini-2025-04-14 | 3.0 | 8.6 | 33.2 | **48.8** |
| gpt-4.1-2025-04-14 | 34.2 | 46.6 | 59.0 | **61.2** |
| Claude-4-Sonnet-20250514 | 36.6 | 47.8 | 57.0 | **60.4** |
| Qwen3-14B | 3.0 | 6.8 | 10.2 | **13.4** |
| Qwen3-32B | 9.2 | 10.2 | 17.0 | **26.4** |
| Qwen3-235B-A22B-Instruct-2507 | 55.4 | 59.2 | 65.6 | **71.4** |
| Average | 23.6 | 29.9 | 40.3 | **46.9** |
| Reasoning Models | | | | |
| gpt-5-mini-2025-08-07 | 58.0 | 60.4 | 70.0 | **71.2** |
| gpt-5-2025-08-07 | 60.2 | 61.6 | 68.4 | **68.4** |
| Claude-4-Sonnet-20250514 (Thinking) | 50.8 | 54.2 | 60.4 | **64.4** |
| Qwen3-14B (Thinking) | 33.0 | 37.8 | 50.0 | **51.6** |
| Qwen3-32B (Thinking) | 46.8 | 55.6 | 57.0 | **58.2** |
| Qwen3-235B-A22B-Thinking-2507 | 58.4 | 59.2 | 67.2 | **70.4** |
| Average | 51.2 | 54.8 | 62.2 | **64.0** |

Table 5: Ablation study on the four-step statement formalization pipeline showing cumulative performance improvements. Each stage adds measurable benefits: 1-Stage (direct generation), 2-Stage (+ self-refinement), 3-Stage (+ decomposition), 4-Stage (+ translation). Results demonstrate the value of systematic decomposition over direct formalization.

## A    ABLATION ON THE STATEMENT FORMALIZATION PHASE

Table 5 presents a comprehensive ablation study examining the individual contributions of each component within our four-step statement formalization pipeline across multiple model architectures and both evaluation benchmarks. The results demonstrate that each step provides measurable and cumulative performance improvements, with decomposition into semi-formalized structure consistently yielding the most substantial gains across all model families, typically improving baseline performance by 15-25 percentage points for non-reasoning models and 8-15 percentage points for reasoning models.

The systematic translation step shows particularly strong benefits for maintaining semantic accuracy and reducing incorrect parameter assignments, with improvements most pronounced for smaller models that benefit from the structured approach to mapping informal concepts to formal correspondents. The composition step provides consistent improvements in maintaining proper quantifier scoping and logical relationships between premises and conclusions, addressing a significant source of validation errors in direct end-to-end approaches. The self-refinement step with symbolic validator feedback contributes incremental but meaningful improvements of 3-8 percentage points across all model types, demonstrating the value of iterative correction guided by comprehensive semantic feedback rather than simple compiler error messages.

These findings validate our design choice to implement a multi-stage pipeline rather than attempting direct end-to-end formalization, as each component addresses distinct and complementary aspects of the autoformalization challenge. The ablation results also reveal that the performance gains are not merely additive but exhibit synergistic effects, particularly between decomposition and translation steps, where the structured intermediate representation enables more effective utilization of learned abstractions and reduces the cognitive load on language models during the formalization process.

## B    ABLATION ON THE ONE-SHOT EXAMPLE

Table 6 demonstrates the critical importance of providing structured examples for consistent task performance across different model architectures, revealing substantial performance degradation when models operate without guided examples. The results show that models without 1-shot examples exhibit significantly degraded performance, typically declining by 20-40 percentage points across both benchmarks, with increased variance in output formatting and frequent failures to adhere to domain-specific language syntactic requirements. This performance degradation is particularly pronounced for specialized autoformalizer models, which show complete task failure without proper formatting guidance, and for smaller general-purpose models that struggle to infer the expected decomposition structure from task descriptions alone.

| Model | Examples | LeanEuclidPlus | | |
|---|---|---|---|---|
| | | Baseline | DNA | DNOracleA |
| Non-Reasoning Models | | | | |
| gpt-4.1-mini-2025-04-14 | 0-Shot | 3.8 | 21.6 | 26.6 |
| | 1-Shot | 4.8 | 42.4 | 48.8 |
| gpt-4.1-2025-04-14 | 0-Shot | 14.4 | 29.8 | 48.6 |
| | 1-Shot | 26.8 | 48.4 | 61.2 |
| Claude-4-Sonnet-20250514 | 0-Shot | 10.0 | 30.0 | 37.8 |
| | 1-Shot | 34.2 | 58.2 | 60.4 |
| Qwen3-14B | 0-Shot | 1.0 | 0.0 | 4.0 |
| | 1-Shot | 1.0 | 9.6 | 13.4 |
| Qwen3-32B | 0-Shot | 1.8 | 5.4 | 17.4 |
| | 1-Shot | 4.6 | 20.6 | 26.4 |
| Qwen3-235B-A22B-Instruct-2507 | 0-Shot | 16.8 | 29.2 | 44.4 |
| | 1-Shot | 40.8 | 64.4 | 71.4 |
| Average | 0-Shot | 8.0 | 19.3 | 29.8 |
| | 1-Shot | 18.7 | 40.6 | 46.9 |
| Reasoning Models | | | | |
| gpt-5-mini-2025-08-07 | 0-Shot | 12.8 | 31.8 | 33.4 |
| | 1-Shot | 25.8 | 58.2 | 71.2 |
| gpt-5-2025-08-07 | 0-Shot | 18.2 | 22.8 | 32.6 |
| | 1-Shot | 35.4 | 55.8 | 68.4 |
| Claude-4-Sonnet-20250514 (Thinking) | 0-Shot | 14.0 | 30.0 | 40.0 |
| | 1-Shot | 32.8 | 57.4 | 64.4 |
| Qwen3-14B (Thinking) | 0-Shot | 12.0 | 15.8 | 26.8 |
| | 1-Shot | 25.4 | 38.6 | 51.6 |
| Qwen3-32B (Thinking) | 0-Shot | 18.0 | 25.2 | 34.0 |
| | 1-Shot | 29.6 | 40.8 | 58.2 |
| Qwen3-235B-A22B-Thinking-2507 | 0-Shot | 27.0 | 30.4 | 56.0 |
| | 1-Shot | 40.2 | 55.6 | 70.4 |
| Average | 0-Shot | 17.0 | 26.0 | 37.1 |
| | 1-Shot | 31.5 | 51.1 | 64.0 |

Table 6: Ablation study on 1-shot example provision showing the critical importance of structured examples for task performance. Models without examples exhibit significantly degraded performance (20-40 percentage point drops) and increased variance, particularly for domain-specific language adherence and decomposition consistency.

The provision of comprehensive chain-of-thought examples serves multiple crucial functions beyond performance improvement. First, it establishes consistent formatting expectations that enable models to generate outputs compatible with our symbolic validation pipeline, reducing parsing errors and ensuring meaningful evaluation. Second, the detailed reasoning traces in the 1-shot examples demonstrate the expected progression from informal statement analysis through semi-formalized decomposition to final formal translation, providing a concrete template that models can adapt to new problem instances. Third, the examples help calibrate the level of detail required at each decomposition stage, preventing both over-simplification that loses semantic content and over-complication that introduces unnecessary complexity.

The ablation results also reveal interesting model-specific dependencies on example provision. Reasoning models show more graceful degradation without examples compared to standard models, likely due to their training on step-by-step problem solving, but still benefit substantially from structured guidance for domain-specific requirements. Conversely, fine-tuned autoformalizer models show catastrophic performance collapse without examples, highlighting their brittle nature and limited instruction-following capabilities outside their narrow training distribution. These findings validate our decision to include detailed exemplars in all experimental configurations and suggest

**LeanEuclidPlus: Extracted Concepts**

two distinct lines intersecting at a point in 2-dimensional Euclidean space
three or more distinct points being sequentially aligned on a line in Euclidean plane
two sets of points lying on opposing sides of a line in Euclidean plane
two angles in Euclidean plane being supplementary
two lines being parallel in 2-dimensional Euclidean space
two angles in Euclidean plane being congruent
a point lying between two other points on a line segment in Euclidean plane
a triangle in 2-dimensional Euclidean space defined by three non-collinear points
two triangles sharing a common vertex in Euclidean plane
two points lying on the same side of a line in Euclidean plane
two points lying on opposite sides of a line in Euclidean plane
...

**ProofNet-Hard: Extracted Concepts**

square root function mapping nonnegative real numbers to nonnegative real numbers
a sequence of real numbers being convergent
limit function mapping a convergent sequence of real numbers to a real number
topology on a set X
family of topologies on a fixed set X
intersection of a family of topologies on X being a topology on X
union of a family of topologies on X not necessarily being a topology on X
one topology on X being finer than another topology on X (inclusion of collections)
two topologies on the same set being comparable under inclusion
two topologies on the same set being not comparable under inclusion
basis for a topology on a set
topology on $\mathbb{R}$ generated by the basis of intervals with rational endpoints
...

Figure 2: Example result from Step 1 Concept Extraction.

that example provision represents a crucial component of any practical autoformalization system deployment.

## C  QUALITATIVE ANALYSIS

### C.1  ABSTRACTION LEARNING

**Step 1: Concept Extraction.** To illustrate the types of concepts extracted, we examine representative examples from both benchmarks shown in Figure 2. For LeanEuclidPlus, the extracted concepts include objects such as non-degenerate triangles in 2-dimensional Euclidean space, relations such as two distinct points being on a line, and various geometric functions. For ProofNetHard, which addresses more advanced mathematical domains, the extracted concepts span a broader range: objects include topologies on a set X, families of topologies on a set X, and bases for topologies on a set X; relations include predicates such as a sequence of real numbers being convergent and the intersection of a family of topologies on X being a topology on X; functions include the square root function mapping nonnegative real numbers to nonnegative real numbers and the limit function mapping a convergent sequence of real numbers to a real number.

**LeanEuclidPlus: Filtered Out Concepts**

a point lying between two other points on a line segment in Euclidean plane
two points lying on the same side of a line in Euclidean plane
two points lying on opposite sides of a line in Euclidean plane
the measure of an angle in Euclidean plane mapping an angle to a real number in degrees
a convex quadrilateral in 2-dimensional Euclidean space being a parallelogram

**LeanEuclidPlus: Kept Concepts**

two distinct lines intersecting at a point in 2-dimensional Euclidean space
three or more distinct points being sequentially aligned on a line in Euclidean plane
two sets of points lying on opposing sides of a line in Euclidean plane
two angles in Euclidean plane being supplementary
two lines being parallel in 2-dimensional Euclidean space
two angles in Euclidean plane being congruent
two triangles sharing a common vertex in Euclidean plane
an angle at a vertex of a triangle in Euclidean plane
...

**ProofNet-Hard: Filtered Out Concpets**

square root function mapping nonnegative real numbers to nonnegative real numbers
a sequence of real numbers being convergent
limit function mapping a convergent sequence of real numbers to a real number
family of topologies on a fixed set X
intersection of a family of topologies on X being a topology on X
union of a family of topologies on X not necessarily being a topology on X
basis for a topology on a set X
connected topological space (no separation into two disjoint nonempty open sets)
Stone–Čech compactification βX of a completely regular space X
...

**ProofNet-Hard: Kept Concepts**

sequence of real numbers defined by a_i = sqrt(i + 1) − sqrt(i) for i in the natural numbers
subbasis for a topology on a set X
lower limit topology $\mathbb{R}$_l on the real line (topology with basis of half-open intervals [a,b))
K-topology $\mathbb{R}$_K on the real line
topology on $\mathbb{R}$ generated by the basis of intervals with rational endpoints
sequence of functions f_n: [0,1] → $\mathbb{R}$ defined by f_n(x) = x^n
pointwise limit function g: [0,1] → $\mathbb{R}$ with g(x) = 0 for 0 ≤ x < 1 and g(1) = 1
limit point compactness (every infinite subset of X has a limit point in X)
uniform topology on [0,1]^ω induced by the supremum metric
...

Figure 3: Example result from Step 2 Concept Filtering.

**Step 2: Concept Filtering.** The filtering behavior and results vary significantly between our two benchmarks due to differences in the richness of their underlying formal libraries.

For LeanEuclidPlus, shown in Figure 3, the minimal nature of the target DSL results in relatively few concepts being filtered out. Representative examples include the concept of a point lying between two other points on a line segment in Euclidean space, which is filtered because it corresponds to the primitive relation `between` axiomatized in the target DSL. Similarly, the concept of two points lying on the same side of a line is filtered because it corresponds to the primitive relation `Point.sameSide`. The relation describing two points being on opposite sides of a line is also filtered because it already exists in the target DSL as an abbreviation expressed as `¬ a.onLine l ∧ ¬ b.onLine l ∧ ¬ sameSide a b l`.

In contrast, ProofNetHard exhibits substantially more filtering activity due to the extensive coverage of Lean Mathlib, as illustrated in Figure 3. Concepts such as a sequence of real numbers being convergent are filtered because they have direct formal correspondents in Mathlib, specifically the filter limit `Filter.Tendsto f atTop (nhds l)` defined in `Mathlib/Topology/Instances/Real`. The concept of a family of topologies on a set X is filtered because it is already represented as `Set (TopologicalSpace X)` or as an indexed family `ι -> TopologicalSpace X` with complete lattice operations in `Mathlib/Topology/Basic`. Even sophisticated mathematical concepts such as the Stone-Cech compactification of a completely regular space X are filtered because Mathlib provides comprehensive support through `StoneCech X` along with the unit map `stoneCechUnit` and extension operation `stoneCechExtend` in `Mathlib/Topology/StoneCech`.

**Step 3: Concept Dependency Graph (CDG) Construction.** For LeanEuclidPlus, Figure 4 presents an example of a depth-3 CDG where the target concept to be formalized depends on both already-available formal definitions in the target language, such as `Triangle.ofPoints` and `Line.intersectsLine`, as well as other concepts that do not yet have single direct formal correspondents and must themselves be formalized first. An illustrative example of the nuanced dependency analysis performed in this step involves the concept of a point lying on a line segment including endpoints in the Euclidean plane. This concept is superficially similar to the previously filtered concept of a point lying between two other points on a line segment in Euclidean space, which was removed because it has a direct formal correspondent in the primitive relation `between`. However, these concepts are semantically distinct: the former permits the point to be an endpoint of the line segment, while the latter does not. Consequently, the former concept is retained and can be formalized using `between`, logical disjunction, and equality. The dependency analysis also correctly identifies that the concept of the line through two distinct points in Euclidean plane is directly expressible using `Point.onLine`, inequality, and existential quantification. The concept dependency for a line intersecting two sides of a triangle in Euclidean plane is similarly constructed correctly with appropriate dependencies identified.

For ProofNetHard, Figure 4 shows a simpler depth-2 CDG. The example concept of the lower limit topology on the real line, which is the topology with a basis of half-open intervals $[a, b)$, demonstrates correct dependency construction. The analysis correctly identifies that this concept is directly expressible using the following formal correspondents: `Set (Set ℝ)`, `Ico : ℝ -> ℝ -> Set ℝ` representing half-open intervals $[a, b)$, `TopologicalSpace.GenerateFrom : Set (Set ℝ) -> TopologicalSpace ℝ`, existential quantification, conjunction, and equality to describe the generating family. This concept dependency is correctly constructed and precisely matches the oracle definition of the lower limit topology written by human experts, validating the accuracy of our dependency analysis.

**Step 4: Concept Formalization.** For LeanEuclidPlus, Figure 5 illustrates the formalization of the concept of three or more distinct points being sequentially aligned on a line in Euclidean plane, along with several helper definitions. The natural language description of this concept encompasses multiple sub-concepts: mutually distinct points, collinear points, and points aligned in sequential order. The language model correctly decomposes the formalization task by first formalizing each of these sub-concepts as helper definitions, then composing them to formalize the target concept. While this approach is correct and faithful to the original natural language concept, it introduces some redundancy that will be addressed in the subsequent refactoring phase.

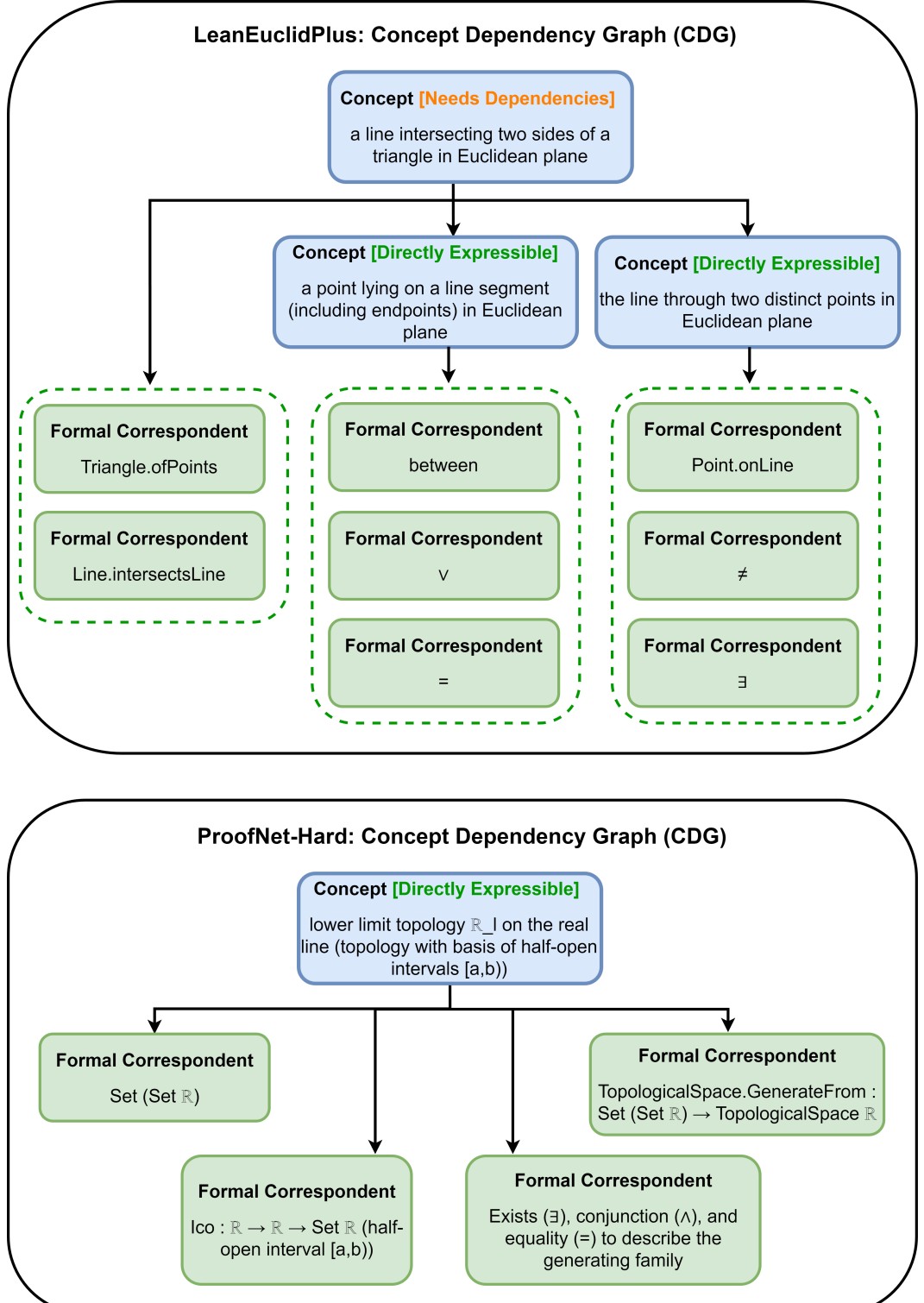

Figure 4: Example result from Step 3 CDG Construction.

**LeanEuclidPlus: Formalized Concepts**

```
@[simp]
def distinctFromList (x : Point) (ys : List Point) : Prop :=
  match ys with
  | []      => True
  | y :: ys => x ≠ y ∧ distinctFromList x ys

@[simp]
def pairwiseDistinct (pts : List Point) : Prop :=
  match pts with
  | []       => True
  | x :: xs  => distinctFromList x xs ∧ pairwiseDistinct xs

@[simp]
def allOnLine (pts : List Point) (L : Line) : Prop :=
  match pts with
  | []       => True
  | x :: xs  => x.onLine L ∧ allOnLine xs L

@[simp]
def betweennessChain (pts : List Point) : Prop :=
  match pts with
  | a :: b :: c :: rest => between a b c ∧ betweennessChain (b :: c :: rest)
  | _                   => True

@[simp]
def sequentiallyAlignedOnLine (pts : List Point) (L : Line) : Prop :=
  allOnLine pts L ∧ pairwiseDistinct pts ∧ betweennessChain pts

...
```

**ProofNet-Hard: Formalized Concepts**

```
@[simp]
def TopologicalSpace.comparable {α : Type*} (t u : TopologicalSpace α) : Prop :=
  t ≤ u ∨ u ≤ t

@[simp]
def TopologicalSpace.notComparable {α : Type*} (t u : TopologicalSpace α) : Prop :=
  ¬ TopologicalSpace.comparable t u

@[simp]
def IsSubbasisFor {X : Type*} (S : Set (Set X)) (t : TopologicalSpace X) : Prop :=
  t = TopologicalSpace.generateFrom S

@[simp]
def lowerLimitGeneratingSets : Set (Set ℝ) :=
  {U | ∃ a b : ℝ, a < b ∧ U = Set.Ico a b}

@[simp]
def realLowerLimitTopology : TopologicalSpace ℝ :=
  TopologicalSpace.generateFrom lowerLimitGeneratingSets

...
```

Figure 5: Example result from Step 4 Concept Formalization.

```
                    LeanEuclidPlus: Refactored Formalization

        @[simp]
        def betweennessChain (pts : List Point) : Prop :=
          match pts with
          | a :: b :: c :: rest => between a b c ∧ betweennessChain (b :: c :: rest)
          | _                    => True

        @[simp]
        def sequentiallyAligned (pts : List Point) : Prop :=
          betweennessChain pts

        ...
```

```
                    ProofNet-Hard: Refactored Formalization

    @[simp]
    def TopologiesComparable {α : Type*} (t u : TopologicalSpace α) : Prop :=
      t ≤ u ∨ u ≤ t

    @[simp]
    def IsSubbasisFor {X : Type*} (S : Set (Set X)) (t : TopologicalSpace X) : Prop :=
      t = TopologicalSpace.generateFrom S

    @[simp]
    def LowerLimitSubbasis : Set (Set ℝ) :=
      {U | ∃ a b : ℝ, a < b ∧ U = Set.Ico a b}

    @[simp]
    def sorgenfreyTopology : TopologicalSpace ℝ :=
      TopologicalSpace.generateFrom LowerLimitSubbasis

    ...
```

Figure 6: Example result from Step 5 Formalization Refactoring.

For ProofNetHard, Figure 5 demonstrates the formalization of concepts with their helper definition dependencies. The formalization of `TopologicalSpace.notComparable` depends on and utilizes `TopologicalSpace.Comparable` as a helper definition. Similarly, `realLowerLimitTopology` depends on and uses `lowerLimitGeneratingSets` as a helper definition. Both the main concepts and their helper definitions are correctly formalized, demonstrating the system's ability to manage dependencies during formalization.

**Step 5: Formalization Refactoring.** For LeanEuclidPlus, Figure 6 demonstrates significant refactoring opportunities. As shown in the initial formalization in Figure 5, the concept was implemented using multiple helper definitions for mutually distinct points, collinear points, and sequential alignment. However, careful analysis reveals that the chain of betweenness relations already guarantees that the list of points must lie on the same line and must be mutually distinct by virtue of the `between` relation's semantics. Consequently, the two separate helper definitions for collinearity and mutual distinctness can be eliminated as redundant. The refactored version shown in Figure 6 consolidates these concepts by retaining only the essential `betweennessChain` definition, with `sequentiallyAligned` serving as a more intuitive alias for the same concept.

For ProofNetHard, Figure 6 shows more conservative refactoring activity. The definition `TopologicalSpace.notComparable` is removed because it is simply the logical negation concatenated with `TopologicalSpace.Comparable`, making a separate API unnecessary and potentially confusing. The refactoring also includes two renamings that improve domain specificity and clarity: `LowerLimitGeneratingSets` is renamed to `LowerLimitSubbasis` to better reflect its mathematical role, and `realLowerLimitTopology` is renamed to

---

**LeanEuclidPlus: Updated Documentation**

def twoDistinctLinesIntersectAtPoint (L M : Line) (i : Point) --
two distinct lines L and M intersect at point i, i.e. L ≠ M,
i lies on both L and M, and `L.intersectsLine M`.
The syntax is `twoDistinctLinesIntersectAtPoint L M i`.

def sequentiallyAligned (pts : List Point) --
a list of points is sequentially aligned (ordered collinear chain) iff every consecutive
 triple ($p_i$, $p_{i+1}$, $p_{i2}$) satisfies `between $p_i$ $p_{i+1}$ $p_{i2}$`.
No extra line or global distinctness is required beyond betweenness.
The syntax is `sequentiallyAligned [$p_0$, $p_1$, $p_2$, ...]`.

def sequentiallyAlignedThreeOrMore (pts : List Point) --
like `sequentiallyAligned`, additionally requiring the list to contain at least three points.
The syntax is `sequentiallyAlignedThreeOrMore [$p_0$, $p_1$, $p_2$, ...]`.

def twoSetsOpposingSidesOnLine (xs ys : List Point) (L : Line) --
two lists of points lie on opposite sides of line L: (1) no point from either list is on L;
(2) every cross-pair (x ∈ xs, y ∈ ys) is on opposing sides of L.
The syntax is `twoSetsOpposingSidesOnLine [$x_1$, $x_2$, ...] [$y_1$, $y_2$, ...] L`.

def supplementaryAngles (a b c d e f : Point) --
angles ∠ a:b:c and ∠ d:e:f are supplementary iff (∠ a:b:c) + (∠ d:e:f) = ∟ + ∟ .
The syntax is `supplementaryAngles a b c d e f`.

...

Figure 7: Example result from Step 6 Documentation Update.

`sorgenfreyTopology` using the conventional mathematical terminology. While these refactorings are not strictly necessary for correctness, they enhance readability and align with domain conventions, and we therefore consider them correct refactorings.

**Step 6: Documentation Update.** For LeanEuclidPlus, Figure 7 illustrates the selective documentation approach employed by this step. The internal helper definition `betweennessChain` is not directly exposed in the documentation; instead, its more intuitively-named alias `sequentiallyAligned` is presented to users. The natural language description correctly captures the formal semantics of `sequentiallyAligned`, attending to subtle but important details such as the fact that the points are mutually distinct. This demonstrates the system's ability to abstract away implementation details while preserving semantic precision in user-facing documentation.

For ProofNetHard, we adopt a different approach due to the scale of the original Lean Mathlib documentation, which exceeds far beyond the context limit of any current LLMs. Instead, we skip the documentation update step and directly use the compact refactored code (comprising just 21 formal definitions) as documentation for downstream tasks in the decomposition phase.

### C.2 COMPARISION BETWEEN LEARNED AND ORACLE ABSTRACTIONS

We present a qualitative comparison between the abstractions learned by our system and the oracle abstractions hand-crafted by human experts. Figure 8 illustrates representative examples from LeanEuclidPlus, and Figure 9 presents corresponding examples from ProofNet-Hard. Our analysis reveals three distinct categories of learned abstractions.

**Correct Overlapping Abstractions.** The first category comprises abstractions where both the learned and oracle definitions capture semantically equivalent concepts, albeit with different naming conventions or slightly different formulations.

**LeanEuclidPlus: Correct Overlapping Abstractions**

**Learned Abstractions**

```
@[simp]
def betweennessChain (pts : List Point) : Prop :=
  match pts with
  | a :: b :: c :: rest => between a b c ∧ betweennessChain (b :: c :: rest)
  | _              => True

@[simp]
def sequentiallyAligned (pts : List Point) : Prop :=
  betweennessChain pts

@[simp]
def twoDistinctLinesIntersectAtPoint (L M : Line)
(i : Point) : Prop :=
  L ≠ M ∧ i.onLine L ∧ i.onLine M ∧ L.intersectsLine M

@[simp]
def parallel (L M : Line) : Prop :=
  L ≠ M ∧ ¬ L.intersectsLine M

@[simp]
def perpendicularAt (L M : Line) (i a c : Point) : Prop :=
  L.intersectsLine M ∧ formRectilinearAngle a i c L M ∧ (∠ a:i:c) = ⌞

@[simp]
def trianglesSimilar (a b c d e f : Point) : Prop :=
  (∠ b:a:c) = (∠ e:d:f) ∧
  (∠ a:b:c) = (∠ d:e:f) ∧
  (∠ a:c:b) = (∠ d:f:e) ∧
  (|(a−b)| / |(d−e)|) = (|(b−c)| / |(e−f)|) ∧
  (|(b−c)| / |(e−f)|) = (|(c−a)| / |(f−d)|)

  ...
```

**Oracle Abstractions**

```
@[simp]
def sequentiallyAlignedList (point_list : List Point) : Prop :=
  match point_list with
  | [ ] => True
  | [ _ ] => True
  | [ _, _ ] => True
  | a :: b :: c :: rest => sequentiallyAligned a b c ∧
sequentiallyAlignedList (b :: c :: rest)

@[simp]
abbrev twoLinesIntersectAtPoint (L1 L2 : Line)
(i: Point) : Prop :=
  L1.intersectsLine L2 ∧ i.onLine L1 ∧ i.onLine L2 ∧ L1 ≠ L2

@[simp]
abbrev parallel (L1 L2 : Line) : Prop :=
  ¬ L1.intersectsLine L2

@[simp]
abbrev perpendicular (L1 L2 : Line)
(p1 p2 i: Point) : Prop :=
  twoLinesIntersectAtPoint L1 L2 i ∧ p1.onLine L1 ∧ p2.onLine L2
∧ p1 ≠ i ∧ p2 ≠ i ∧ ∠ p1:i:p2 = ⌞

@[simp]
abbrev similar : Triangle → Triangle →  Prop
| (Triangle.ofPoints A B C) ,(Triangle.ofPoints D E F) =>
  (|(A−B)| / |(D−E)| = |(B−C)| / |(E−F)| ∧
  |(B−C)| / |(E−F)| = |(C−A)| / |(F−D)| ∧
  |(C−A)| / |(F−D)| = |(A−B)| / |(D−E)| ∧
  ∠ A:B:C = ∠ D:E:F ∧ ∠ A:C:B = ∠ D:F:E ∧ ∠ B:A:C = ∠ E:D:F)

  ...
```

**LeanEuclidPlus: Correct Non-Overlapping Abstractions**

**Learned Abstractions**

```
@[simp]
def noneOnLine (pts : List Point) (L : Line) : Prop :=
  match pts with
  | []       => True
  | x :: xs  => ¬ x.onLine L ∧ noneOnLine xs L

@[simp]
def allOpposingSidesToPoint (x : Point) (ys : List Point)
(L : Line) : Prop :=
  match ys with
  | []       => True
  | y :: ys => x.opposingSides y L ∧ allOpposingSidesToPoint x ys L

@[simp]
def pairwiseAcrossOpposing (xs ys : List Point) (L : Line) : Prop :=
  match xs with
  | []       => True
  | x :: xs => allOpposingSidesToPoint x ys L
∧ pairwiseAcrossOpposing xs ys L

@[simp]
def twoSetsOpposingSidesOnLine (xs ys : List Point)
(L : Line) : Prop :=
  noneOnLine xs L ∧ noneOnLine ys L
∧ pairwiseAcrossOpposing xs ys L

@[simp]
def lineCutsTriangleOnABandACAt
   (a b c p q : Point) (AB BC CA L : Line) : Prop :=
  formTriangle a b c AB BC CA ∧ between a p b ∧ between c q a
∧ p.onLine L ∧ q.onLine L

  ...
```

**Oracle Abstractions**

```
@[simp]
def pointDistinctFromList (a : Point) (point_list : List Point) : Prop :=
  match point_list with
  | [] => True
  | b :: rest => a ≠ b ∧ pointDistinctFromList a rest

@[simp]
def mutuallyDistinctPointsList (point_list : List Point) : Prop :=
  match point_list with
  | [] => True
  | [_] => True
  | a :: rest => pointDistinctFromList a rest
∧ mutuallyDistinctPointsList rest

@[simp]
def sameSideList (point_list : List Point) (L : Line) : Prop :=
  match point_list with
  | [] => True
  | [_] => True
  | [a, b] => a.sameSide b L
  | a :: b :: c :: rest => a.sameSide b L ∧ sameSideList (b :: c :: rest) L

@[simp]
abbrev sameSideDistinctList (point_list : List Point)
(L : Line) : Prop :=
  sameSideList point_list L ∧ mutuallyDistinctPointsList point_list

  ...
```

Figure 8: Comparision between Learned and Oracle Abstractions for LeanEuclidPlus.

For LeanEuclidPlus, the learned abstraction `betweennessChain` and its alias `sequentiallyAligned` correspond directly to the oracle's `sequentiallyAlignedList`, both encoding the property that consecutive points in a list satisfy the betweenness relation. Similarly, `twoDistinctLinesIntersectAtPoint` mirrors the oracle's `twoLinesIntersectAtPoint`, and the learned `parallel` and `perpendicularAt` definitions align precisely with their oracle counterparts. The learned `trianglesSimilar` captures the same geometric concept as the oracle's `similar` abbreviation, encompassing both angle equality and proportional side lengths.

For ProofNet-Hard, the learned `sqrtSuccDiff` exactly matches the oracle's function `g`, both computing the difference between consecutive square roots. The learned `LowerLimitSubbasis`, `sorgenfreyTopology`, `KNatRecip`, `KTopologySubbasis`, and `kTopology` correspond semantically to the oracle's `lower_limit_topology`, `K`, and `K_topology`.

These examples demonstrate that our abstraction learning phase is able to recover most of the common mathematical concepts identified by human experts, validating the effectiveness of our automated approach for discovering domain-specific abstractions without manual intervention.

**Correct Non-Overlapping Abstractions.** The second category consists of abstractions that are correct formalizations of the natural language concepts but do not overlap with the oracle.

For LeanEuclidPlus, the learned abstractions include `noneOnLine`, which recursively verifies that no point in a list lies on a given line, and `allOpposingSidesToPoint`, which checks whether all points in a list are on opposite sides of a line relative to a reference point. The learned `pairwiseAcrossOpposing` and `twoSetsOpposingSidesOnLine` provide compositional predicates for reasoning about point configurations across lines, while `lineCutsTriangleOnABandACAt` encapsulates the specific geometric configuration where a line intersects two sides of a triangle. In contrast, the oracle for LeanEuclidPlus contains different abstractions such as `pointDistinctFromList`, `mutuallyDistinctPointsList`, `sameSideList`, and `sameSideDistinctList`, which focus on distinctness and same-side predicates over point lists.

For ProofNet-Hard, the learned abstractions include `StrictlyBoundedAboveBy`, a generic predicate for sequences bounded by a constant, `TopologiesComparable` for comparing topologies, and `NestedClosedNonempty` along with `NestedClosedNonemptyInterNonempty` for reasoning about nested sequences of closed sets. The oracle involves different abstractions such as `is_topology`, a first-principles definition of topological structure, and `countably_compact`, which directly encodes the definition of countable compactness.

These non-overlapping abstractions illustrate that concept extraction by humans and LLMs from the same natural language corpus can be complementary: each identifies valid abstractions that the other may overlook, suggesting that combining both approaches could yield richer and more comprehensive libraries of reusable abstractions.

**Correct Learned Abstractions, Wrong Oracle.** The third category, observed exclusively in ProofNet-Hard, consists of cases where the learned abstractions are correct while the oracle abstractions contain semantic errors. The learned `IsLimitPoint` correctly defines a limit point (accumulation point) of a set $A$ as a point $x$ for which every neighborhood intersects $A \setminus \{x\}$, and `LimitPointCompact` correctly states that every infinite subset has such a limit point. However, the oracle's `limit_point_compact` incorrectly uses `ClusterPt x (P U)`, conflating the notions of cluster point and accumulation point.

In Mathlib, `ClusterPt x F` is defined as `NeBot (N x ⊓ F)`, which for a principal filter $\mathcal{P}$ U reduces to checking whether $x \in \overline{U}$. Crucially, since any point $x \in U$ trivially satisfies `ClusterPt x (P U)` (every neighborhood of $x$ contains $x$ itself and hence meets $U$), the oracle's definition degenerates to merely asserting that $U$ is nonempty—a property that holds vacuously for any infinite set. This renders the oracle definition trivially true in every topological space, including infinite discrete spaces where the correct limit-point compactness property provably fails.

## ProofNet-Hard: Correct Overlapping Abstractions

### Learned Abstractions

```
@[simp]
def sqrtSuccDiff (n : ℕ) : ℝ :=
  Real.sqrt ((n : ℝ) + 1) - Real.sqrt (n : ℝ)

@[simp]
def LowerLimitSubbasis : Set (Set ℝ) :=
  {U | ∃ a b : ℝ, a < b ∧ U = Set.Ico a b}

@[simp]
def sorgenfreyTopology : TopologicalSpace ℝ :=
  TopologicalSpace.generateFrom LowerLimitSubbasis

@[simp]
def KNatRecip : Set ℝ :=
  {x | ∃ n : PNat, x = (1 : ℝ) / ((n : ℕ) : ℝ)}

@[simp]
def KTopologySubbasis : Set (Set ℝ) :=
  {U | ∃ a b : ℝ, a < b ∧ (U = Set.Ioo a b
  ∨ U = Set.Ioo a b \ KNatRecip)}

@[simp]
def kTopology : TopologicalSpace ℝ :=
  TopologicalSpace.generateFrom KTopologySubbasis
...
```

### Oracle Abstractions

```
@[simp]
def g (n : ℕ) : ℝ := sqrt (n + 1) - sqrt n

@[simp]
def lower_limit_topology (X : Type) [Preorder X] :=
  TopologicalSpace.generateFrom {S : Set X |
  ∃ a b, a < b ∧ S = Set.Ico a b}

@[simp]
def K : Set ℝ := {r | ∃ n : ℕ, r = 1 / n}

@[simp]
def K_topology :=
  TopologicalSpace.generateFrom
    ({S : Set ℝ | ∃ a b, a < b ∧ S = Set.Ioo a b} ∪
    {S : Set ℝ | ∃ a b, a < b ∧ S = Set.Ioo a b \ K})

...
```

## ProofNet-Hard: Correct Non-Overlapping Abstractions

### Learned Abstractions

```
@[simp]
def StrictlyBoundedAboveBy {α : Type*} [LT α]
(s : ℕ → α) (c : α) : Prop :=
  ∀ n, s n < c

@[simp]
def TopologiesComparable {α : Type*}
(t u : TopologicalSpace α) : Prop :=
  t ≤ u ∨ u ≤ t

@[simp]
def NestedClosedNonempty {X : Type*}
[TopologicalSpace X] (C : ℕ → Set X) : Prop :=
  Antitone C ∧ (∀ n : ℕ, IsClosed (C n))
∧ (∀ n : ℕ, (C n).Nonempty)

@[simp]
def NestedClosedNonemptyInterNonempty {X : Type*}
[TopologicalSpace X] : Prop :=
  ∀ (C : ℕ → Set X),
    Antitone C →
    (∀ n : ℕ, IsClosed (C n)) →
    (∀ n : ℕ, (C n).Nonempty) →
    (∩ n, C n).Nonempty
```

### Oracle Abstractions

```
@[simp]
def is_topology (X : Type*) (T : Set (Set X)) :=
  Set.univ ∈ T ∧
  (∀ s t, s ∈ T → t ∈ T → s ∩ t ∈ T) ∧
  (∀ s, (∀ t ∈ s, t ∈ T) → Set.sUnion s ∈ T)

@[simp]
def countably_compact (X : Type*) [TopologicalSpace X] :=
  ∀ U : ℕ → Set X,
  (∀ i, IsOpen (U i)) ∧ ((Set.univ : Set X) ⊆ ∪ i, U i) →
  (∃ t : Finset ℕ, (Set.univ : Set X) ⊆ ∪ i ∈ t, U i)
```

## ProofNet-Hard: Correct Learned Abstractions, Wrong Oracle

### Learned Abstractions

```
@[simp]
def IsLimitPoint {X : Type*} [TopologicalSpace X]
(A : Set X) (x : X) : Prop :=
  ∀ U ∈ nhds x, (U ∩ (A \ {x})).Nonempty

@[simp]
def LimitPointCompact {X : Type*} [TopologicalSpace X] : Prop :=
  ∀ A : Set X, A.Infinite → ∃ x : X, IsLimitPoint A x
```

### Oracle Abstractions

```
@[simp]
def limit_point_compact (X : Type*) [TopologicalSpace X] :=
  ∀ U : Set X, Infinite U → ∃ x ∈ U, ClusterPt x (𝒫 U)
```

Figure 9: Comparision between Learned and Oracle Abstractions for ProofNet-Hard.

This example demonstrates that automated abstraction learning by LLMs can sometimes produce more reliable results than manually constructed oracles, particularly in domains where the formalizations contain subtle semantic distinctions that may escape human review.

### C.3    Hierarchical Decomposition

We present a qualitative analysis demonstrating the effectiveness of our hierarchical decomposition approach (Section 2.2) for autoformalization. Using Qwen3-235B-A22B-Instruct-2507 with learned abstractions on LeanEuclidPlus, we compare direct formalization (Abstract column in Table 3) against our four-stage decomposition pipeline (DNA column in Table 3) on a representative problem.

**Informal Statement.** The English statement describes two triangles TUW and VSW sharing vertex W, with lines SU and TV intersecting at W. Given that W lies between S and U, and between T and V, along with the premises `angle TUW = angle UTW` and `SV || TU`, the goal is to prove `angle V = angle S`.

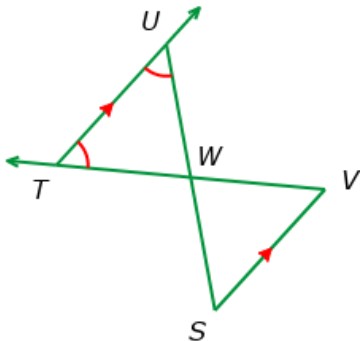

Figure 10: Diagram for Parallel-17 in LeanEuclidPlus.

The following is the complete reasoning trace produced by Qwen3-235B when formalizing directly without hierarchical decomposition:

```
We are given an English statement involving two triangles, TUW and VSW, sharing a point W. The lines SU and TV
    intersect at W, and W lies between S and U, and also between T and V. We are given that \\angle TUW =
    \\angle UTW (i.e., two angles in triangle TUW are congruent), and that SV is parallel to TU. We are to
    prove that \\angle V = \\angle S.

Let's break this down step by step.

---

### Step 1: Identify all points, lines, and geometric entities

Points mentioned:
- T, U, W -- form triangle TUW
- V, S, W -- form triangle VSW
- W is the shared point
- Lines: SU and TV intersect at W
- W lies between S and U -> between S W U
- W lies between T and V -> between T W V

Lines to define:
- For triangle TUW: sides TU, UW, WT
- For triangle VSW: sides VS, SW, WV
- Line SU: contains S and U
- Line TV: contains T and V

We are given:
1. \\angle TUW = \\angle UTW
    - \\angle TUW is the angle at U in triangle TUW: \\angle  T:U:W
    - \\angle UTW is the angle at T in triangle TUW: \\angle  U:T:W
    - So: (\\angle  T:U:W) = (\\angle  U:T:W)
2. SV || TU
    - SV is the line through S and V
    - TU is the line through T and U
    - So: parallel (line SV) (line TU)
3. We are to prove: \\angle V = \\angle S
    - \\angle V likely means the angle at V in triangle VSW: \\angle  S:V:W or \\angle  W:V:S?
    - \\angle S likely means the angle at S in triangle VSW: \\angle  V:S:W or \\angle  W:S:V?

But note: triangle VSW has vertices V, S, W. So:
```

```
- \\angle V in triangle VSW is \\angle  S:V:W
- \\angle S in triangle VSW is \\angle  V:S:W

So we are to prove: (\\angle  S:V:W) = (\\angle  V:S:W)

But wait -- is that necessarily true? Let's see.

Alternatively, could \\angle V and \\angle S refer to angles in different triangles? But the only triangles
    are TUW and VSW.

Given that SV || TU, and lines SU and TV intersect at W, this looks like a configuration where two lines cross
    at W, and we have a pair of parallel lines cutting across them -- possibly forming similar triangles or
    using properties of parallel lines and transversals.

But the conclusion is \\angle V = \\angle S -- i.e., two angles at V and S, likely in triangle VSW.

Wait -- if SV || TU, and TV and SU are transversals intersecting at W, then we might use alternate interior
    angles or corresponding angles.

But \\angle V and \\angle S are both in triangle VSW -- if \\angle S:V:W = \\angle V:S:W, that would mean
    triangle VSW is isosceles at W.

But let's reexamine.

Alternatively, could \\angle V mean \\angle WVS and \\angle S mean \\angle WSV? Yes -- that's the same as
    above.

So the conclusion is: in triangle VSW, \\angle  at V equals \\angle  at S -> triangle VSW is isosceles with |
    SW| = |VW|?

But the conclusion is just \\angle V = \\angle S, i.e., angle equality.

So we are to prove: (\\angle  S:V:W) = (\\angle  V:S:W)

But we are given:
- \\angle T:U:W = \\angle U:T:W -> triangle TUW is isosceles at W? Let's see:
  - \\angle  at U: \\angle T:U:W
  - \\angle  at T: \\angle U:T:W
  - If these are equal, then in triangle TUW, angles at T and U are equal -> so sides opposite are equal -> |
      TW| = |UW| -> isosceles at W.

Also given: SV || TU

Lines SU and TV intersect at W, with W between S and U, and between T and V.

So the configuration is: two lines (SU and TV) intersect at W, forming an "X", and two segments: TU and SV are
    parallel, with T,U on one side, S,V on the other.

This is a classic setup for similar triangles or using parallel line angle properties.

Now, since SV || TU, and TV is a transversal (from T to V, passing through W), then:
- \\angle UTW and \\angle VSW might be alternate interior angles? Let's see.

Wait -- transversal TV crosses parallel lines TU and SV.

But TU and SV are the parallel lines.

Transversal TV: intersects TU at T, and SV at V.

So at point T: angle between TU and TV -> \\angle U:T:W (since W is on TV beyond T? Wait -- W is between T and
    V, so from T to V: T-W-V.

So line TV: T -- W -- V

Similarly, SU: S -- W -- U

So TU is from T to U, SV is from S to V.

Given SV || TU.

Now, transversal TV crosses TU at T and SV at V.

So the angle at T between TU and TV (i.e., \\angle  between TU and TW) is \\angle U:T:W

At V, the angle between SV and TV (i.e., between SV and VW) -- but direction matters.

Since SV || TU, and TV is a transversal, then:
- \\angle UTW (at T) and \\angle SVW (at V) are corresponding angles? Or alternate?

Let's define directions.

Assume standard configuration: lines TU and SV are parallel, transversal TV goes from T to V, crossing both.

Then \\angle  between TU and TV at T, and \\angle  between SV and TV at V -- if both are on the same side,
    they are corresponding.

But we need to relate to the angles.

Alternatively, transversal SU: from S to U, crosses SV at S and TU at U.

So at U: angle between TU and SU -> \\angle T:U:W (since W is on SU beyond U? No -- W is between S and U, so
    from S to U: S -- W -- U
```

So at U: line TU and line SU meet at U -- angle \\angle T:U:W

At S: line SV and line SU meet at S -- angle \\angle V:S:W

Since SV || TU, and SU is a transversal, then:
- \\angle T:U:W and \\angle V:S:W are corresponding angles -> so they are equal.

So: \\angle T:U:W = \\angle V:S:W

But we are given: \\angle T:U:W = \\angle U:T:W

So: \\angle U:T:W = \\angle V:S:W

Now, look at transversal TV: from T to V, crosses TU at T and SV at V.

At T: angle between TU and TV is \\angle U:T:W

At V: angle between SV and TV -- now, depending on direction.

Since SV || TU, and TV is transversal, then \\angle U:T:W and \\angle S:V:W are alternate interior angles? Or
    corresponding?

Let's suppose the lines are oriented so that TU and SV are horizontal, TU above, SV below, TV going down from
    T to V.

Then at T: angle between TU (left-right) and TV (down) -- say, lower right angle.

At V: angle between SV (left-right) and TV (up) -- upper right angle.

These are alternate interior angles -> equal.

So \\angle U:T:W = \\angle S:V:W

Therefore: \\angle S:V:W = \\angle U:T:W = \\angle T:U:W = \\angle V:S:W

So \\angle S:V:W = \\angle V:S:W

Thus, in triangle VSW, angles at V and S are equal -> \\angle V = \\angle S

So the conclusion is: (\\angle  S:V:W) = (\\angle  V:S:W)

Now we formalize.

---

### Step 2: Declare all variables

Points: T, U, W, V, S
Lines:
- For triangle TUW: TU, UW, WT
- For triangle VSW: VS, SW, WV
- Line SU: contains S and U
- Line TV: contains T and V

We need to define lines:
- TU: line through T and U
- UW: line through U and W
- WT: line through W and T
- VS: line through V and S
- SW: line through S and W
- WV: line through W and V
- SU: line through S and U
- TV: line through T and V

But note: SU and TV are already mentioned.

We also need to express:
- formTriangle T U W TU UW WT
- formTriangle V S W VS SW WV
- twoDistinctLinesIntersectAtPoint SU TV W -> but SU and TV intersect at W, and are distinct
- between S W U
- between T W V
- \\angle T:U:W = \\angle U:T:W -> (\\angle  T:U:W) = (\\angle  U:T:W)
- SV || TU -> parallel (line through S and V) (line through T and U)
- Conclusion: (\\angle  S:V:W) = (\\angle  V:S:W)

But we must define the lines.

Let's assign:
- (T U W V S : Point)
- (TU UW WT : Line) for triangle TUW
- (VS SW WV : Line) for triangle VSW
- (SU TV : Line) for the intersecting lines

But note: SU must be the line through S and U -- but we also have SW and WU? Not necessarily.

We need to ensure that S, W, U are collinear on SU, and T, W, V are collinear on TV.

But we are given that SU and TV intersect at W, and W is between S and U, and between T and V.

So we can define:

```
- SU: line through S and U -- and W is on it
- TV: line through T and V -- and W is on it

But in the triangle definitions:
- For triangle TUW: we need lines TU, UW, WT
- But UW is part of SU? Not necessarily -- unless U, W, S are collinear, which they are -- since W is on SU
    and between S and U.

So points S, W, U are collinear on line SU.

Similarly, T, W, V are collinear on line TV.

So we can define:
- Line TU: through T and U
- Line UW: but U and W are on SU -> so UW is part of SU? But we need a line for side UW of triangle TUW.

But in `formTriangle`, we need three distinct lines: AB, BC, CA.

So for triangle TUW:
- side TU: line TU
- side UW: line UW -- but U and W are on SU -> so line UW must be the same as line SU? Only if S, U, W are
    collinear -- which they are.

But we cannot assume that line UW is the same as SU unless we define it.

But in formalization, we can let:
- line UW be the same as line SU? But SU contains S, W, U -- so yes, line through U and W is the same as line
    SU.

But in Lean, lines are extensional -- if two lines have two points in common, they are the same.

But in our formal system, we are assigning labels.

So to avoid confusion, we should define:
- Let line SU be the line containing S and U -- and since W is between S and U, W is on SU.
- Then for triangle TUW, side UW is on line SU -- so we can use SU as the line for UW? But in `formTriangle T
    U W TU UW WT`, we need:
  - T and U on TU
  - U and W on UW
  - W and T on WT

So UW must be a line containing U and W -- which SU does.

Similarly, for triangle VSW:
- V and S on VS
- S and W on SW -- again, S and W are on SU? No -- S and W are on SU, but W and S are on SU.

But VS is a different line.

Wait -- triangle VSW:
- V and S on line VS
- S and W on line SW -- but S and W are on SU -> so line SW is part of SU
- W and V on line WV

So line SW can be taken as SU? But only if we are allowed to reuse.

But in the formal system, we can have multiple names for the same line, but in `formTriangle`, we need to
    assign specific lines.

So to be safe, we define:
- Line SU: contains S, W, U
- Line TV: contains T, W, V
- Line TU: contains T, U
- Line VS: contains V, S
- Line WT: contains W, T -- but W and T are on TV -> so WT is part of TV
- Line WV: contains W, V -- also on TV

So for triangle TUW:
- TU: line TU
- UW: line SU (since U and W on SU)
- WT: line TV (since W and T on TV)

Similarly, for triangle VSW:
- VS: line VS
- SW: line SU (S and W on SU)
- WV: line TV (W and V on TV)

So we can reuse lines SU and TV in both triangles.

Now, we must ensure that the lines are distinct in each triangle.

For triangle TUW: lines TU, SU, TV -- must be distinct
For triangle VSW: lines VS, SU, TV -- must be distinct

We are not told they are distinct, but in a typical configuration they are.

But the `formTriangle` predicate requires the three lines to be distinct.

So we must assume that.

Similarly, `twoDistinctLinesIntersectAtPoint SU TV W` requires SU != TV.
```

Now, let's write the formal statement.

---

### Step 3: List all premises

We need:

1. formTriangle T U W TU UW WT
   But UW is SU, WT is TV -> so:
   -> formTriangle T U W TU SU TV
   But the arguments are: a b c AB BC CA
   So: T, U, W, TU, SU, TV
   Meaning:
   - T and U on TU
   - U and W on SU
   - W and T on TV
   And TU, SU, TV distinct.

2. formTriangle V S W VS SW WV
   But SW is SU, WV is TV -> so:
   -> formTriangle V S W VS SU TV
   Arguments: V, S, W, VS, SU, TV
   Meaning:
   - V and S on VS
   - S and W on SU
   - W and V on TV
   And VS, SU, TV distinct.

3. distinctPointsOnLine S U SU -- to ensure S and U are distinct and on SU
   But `formTriangle` already requires distinctPointsOnLine for each side?
   Looking back: `formTriangle` requires distinct points on each line:
   - distinctPointsOnLine a b AB
   - distinctPointsOnLine b c BC
   - distinctPointsOnLine a c CA

   So in formTriangle T U W TU SU TV:
   - distinctPointsOnLine T U TU
   - distinctPointsOnLine U W SU
   - distinctPointsOnLine W T TV

   Similarly for V S W VS SU TV:
   - distinctPointsOnLine V S VS
   - distinctPointsOnLine S W SU
   - distinctPointsOnLine W V TV

   So we get distinctness of points on lines.

   But we also need that W is between S and U, and between T and V.

   So:

4. between S W U
5. between T W V

6. twoDistinctLinesIntersectAtPoint SU TV W
   This gives:
   - SU != TV
   - W on SU, W on TV
   - SU intersects TV at W

   But `formTriangle` already requires W on SU and W on TV?
   Yes -- from distinctPointsOnLine U W SU and W T TV, etc.
   But `twoDistinctLinesIntersectAtPoint` also gives that they are distinct and intersect at W -- which we
        need.

   But is it redundant? Possibly, but we need to ensure SU != TV, which might not be implied otherwise.

   So include it.

7. \\angle T:U:W = \\angle U:T:W
   -> (\\angle  T:U:W) = (\\angle  U:T:W)

8. SV || TU
   But SV is the line VS (same line), TU is line TU
   -> parallel VS TU

Conclusion: \\angle S:V:W = \\angle V:S:W
-> (\\angle  S:V:W) = (\\angle  V:S:W)

Note: \\angle S:V:W is the angle at V in triangle VSW: points S, V, W -- so vertex V, with S and W on the
     sides.
Similarly, \\angle V:S:W is angle at S: points V, S, W -- vertex S.

So we need to output:

<<< \\forall  (T U W V S : Point) (TU SU TV VS : Line), [premises] -> (\\angle  S:V:W) = (\\angle  V:S:W) >>>

But wait -- in triangle VSW, we used lines VS, SU, TV.
But SU is used for SW, TV for WV.

But we also need to refer to angle at V: \\angle S:V:W -- this requires points S, V, W.

```
Similarly, angle at S: \\angle V:S:W -- points V, S, W.

We have all points.

But in the premises, we have formTriangle V S W VS SU TV -- this ensures that:
- V and S on VS
- S and W on SU
- W and V on TV

So the triangle is well-formed.

Now, write the full formalization.

---

### Final Formalized Statement

Variables:
- Points: T, U, W, V, S
- Lines: TU (for side TU), SU (for side UW and SW), TV (for side WT and WV), VS (for side VS)

We use:
- formTriangle T U W TU SU TV
- formTriangle V S W VS SU TV
- between S W U
- between T W V
- twoDistinctLinesIntersectAtPoint SU TV W
- (\\angle  T:U:W) = (\\angle  U:T:W)
- parallel VS TU

Conclusion: (\\angle  S:V:W) = (\\angle  V:S:W)

Note: The line VS is the same as SV, TU is the same as UT -- direction doesn't matter for lines or parallel.

So:

<<< \\forall  (T U W V S : Point) (TU SU TV VS : Line), formTriangle T U W TU SU TV \\land  formTriangle V S W
      VS SU TV \\land  between S W U \\land  between T W V \\land  twoDistinctLinesIntersectAtPoint SU TV W
      \\land  (\\angle  T:U:W) = (\\angle  U:T:W) \\land  parallel VS TU -> (\\angle  S:V:W) = (\\angle  V:S:W
      ) >>>
```

The following is the complete reasoning trace produced by Qwen3-235B when using our four-stage hierarchical decomposition pipeline:

```
/*
    To semi-formalize the English Statement into a structure of premises and conclusions, we will examine the
        English Statement sentence by sentence, and break down each sentence into a natural language clause
        that is as atomic as possible.
*/

{
    "premises": [
        /*
        The sentence "Two triangles TUW and VSW share a point W." declares two triangles.
        */
        "The distinct points T, U, and W form a triangle \\triangle  T U W",
        "The distinct points V, S, and W form a triangle \\triangle  V S W",

        /*
        The sentence "The lines SU and TV intersect at point W." declares two lines and their intersection.
        */
        "S and U are distinct points on line SU",
        "T and V are distinct points on line TV",
        "The distinct lines SU and TV intersect at point W",

        /*
        The sentence "The point W lies between S and U, and also lies between T and V."
        */
        "Among the distinct points S, W, U, the point W is between S and U",
        "Among the distinct points T, W, V, the point W is between T and V",

        /*
        The sentence "Given \\angle  T U W = \\angle  U T W"
        */
        "The measure of angle T U W equals the measure of angle U T W",

        /*
        The sentence "S V || T U"
        */
        "S and V are distinct points on line SV",
        "T and U are distinct points on line TU",
        "Line SV is parallel to line TU"
    ],
    "conclusions": [
        "The measure of angle S V W equals the measure of angle V S W"
    ]
}

/*
    To convert the Semi-Formalized Structure into a Formalized Structure, we now formalize each clause.
```

```
*/

{
    "premises": [
        "formTriangle T U W TU UW WT",
        "formTriangle V S W VS SW WV",
        "distinctPointsOnLine S U SU",
        "distinctPointsOnLine T V TV",
        "twoDistinctLinesIntersectAtPoint SU TV W",
        "between S W U",
        "between T W V",
        "(\\angle  T:U:W) = (\\angle  U:T:W)",
        "distinctPointsOnLine S V VS",
        "distinctPointsOnLine T U TU",
        "parallel VS TU"
    ],
    "conclusions": [
        "(\\angle  S:V:W) = (\\angle  V:S:W)"
    ]
}

/*
    Variables: (T U W V S : Point) (TU UW WT VS SW WV SU TV : Line)
*/

<<< \\forall  (T U W V S : Point) (TU UW WT VS SW WV SU TV : Line),
  formTriangle T U W TU UW WT \\land
  formTriangle V S W VS SW WV \\land
  distinctPointsOnLine S U SU \\land
  distinctPointsOnLine T V TV \\land
  twoDistinctLinesIntersectAtPoint SU TV W \\land
  between S W U \\land
  between T W V \\land
  (\\angle  T:U:W) = (\\angle  U:T:W) \\land
  distinctPointsOnLine S V VS \\land
  distinctPointsOnLine T U TU \\land
  parallel VS TU
  ->
  (\\angle  S:V:W) = (\\angle  V:S:W) >>>
```

**Direct Formalization Without Decomposition.** When prompted to formalize directly, Qwen3-235B produces the following output:

```
\\forall  (T U W V S : Point) (TU SU TV VS : Line),
  formTriangle T U W TU SU TV \\land  formTriangle V S W VS SU TV \\land
  between S W U \\land  between T W V \\land
  twoDistinctLinesIntersectAtPoint SU TV W \\land
  (\\angle  T:U:W) = (\\angle  U:T:W) \\land  parallel VS TU
  -> (\\angle  S:V:W) = (\\angle  V:S:W)
```

This formalization is missing two critical premises: `distinctPointsOnLine S U SU` and `distinctPointsOnLine T V TV`. Without these, the statement is weaker than intended because it does not enforce that S and U are distinct points on line SU, nor that T and V are distinct points on line TV. This error type corresponds to the "Stronger/Weaker Translation" category in our error analysis (Table 2).

Notably, the model's reasoning trace explicitly identifies these requirements. In Step 3 of its chain-of-thought, the model states: "`distinctPointsOnLine S U SU` – to ensure S and U are distinct and on SU" and initially considers adding this premise. However, the model then incorrectly reasons: "But `formTriangle` already requires distinctPointsOnLine for each side?" and lists that `formTriangle T U W TU SU TV` provides `distinctPointsOnLine U W SU`. This reasoning is flawed: in the formal language, `SU` is simply a variable name for a line—it could equivalently be called $l_1$ or any other identifier—and has no inherent semantic connection to the points S and U. The only way to establish that S and U are actually distinct points on this line is by explicitly stating `distinctPointsOnLine S U SU`. While `formTriangle` does constrain U and W to lie on the line named `SU`, it says nothing about whether S lies on this line. This illustrates a common failure mode in direct formalization: the extended free-form reasoning process can lead to flawed logical conclusions where the model is misled by suggestive variable naming into believing semantic relationships are already captured when they are not.

**Formalization With Hierarchical Decomposition.** In contrast, our four-stage decomposition pipeline guides the model through a structured process that prevents such errors. When processing the sentence "The lines SU and TV intersect at point W" in Stage 2 (Semi-Formalized Structure), the model is required to break it down into atomic clauses. This forces explicit representation of implicit assumptions: the model produces three separate clauses—"S and U are distinct points on line SU", "T and V are distinct points on line TV", and "The distinct lines SU and TV intersect

at point W". In Stage 3 (Formalized Structure), each atomic clause is independently translated to its formal correspondent. The clause "S and U are distinct points on line SU" simply becomes `distinctPointsOnLine S U SU`, regardless of what other predicates might or might not imply. This one-to-one translation prevents information loss and eliminates the flawed dependency reasoning that caused the direct formalization to fail, yielding:

```
\\forall  (T U W V S : Point) (TU UW WT VS SW WV SU TV : Line),
  formTriangle T U W TU UW WT \\land  formTriangle V S W VS SW WV \\land
  distinctPointsOnLine S U SU \\land  distinctPointsOnLine T V TV \\land
  twoDistinctLinesIntersectAtPoint SU TV W \\land
  between S W U \\land  between T W V \\land
  (\\angle  T:U:W) = (\\angle  U:T:W) \\land
  distinctPointsOnLine S V VS \\land  distinctPointsOnLine T U TU \\land
  parallel VS TU
  -> (\\angle  S:V:W) = (\\angle  V:S:W)
```

The decomposed output correctly includes all necessary premises. The hierarchical structure ensures that each semantic unit from the English statement is explicitly represented as an atomic clause, preventing information loss during the translation process.

**Token Efficiency.** Beyond correctness, our decomposition approach also demonstrates improved token efficiency. Using OpenAI's BPE tokenizer, the direct formalization consumed 4,661 output tokens for the free-form reasoning trace, whereas the decomposed approach used only 1,892 output tokens, which is a 59.4% significant reduction. The structured intermediate representations serve as a more compact and organized "working memory" compared to verbose natural language reasoning, reducing the cost, inference time, and most importantly the long-context challange for the model.

# D  PROMPT TEMPLATES

## D.1  ABSTRACTION LEARNING

### D.1.1  STEP 1: CONCEPT EXTRACTION

```
You are an expert in mathematics and logic with deep knowledge of all fields of mathematics. You are given
    some English mathematical statements and a list of previously extracted mathematical concepts. Your task
    is to extract ALL mathematical concepts mentioned in the current statements. {additional_specs}

## Mathematical Concepts
There are 3 types of mathematical concepts to extract:
1. **Definitions of Mathematical Objects**: Point, Line, Integer, Real Number, Series, Group, Ring, etc.
2. **Relations between Mathematical Objects**: 'a point being on a line', 'an integer being even', 'a group
    being a subgroup of another group', etc.
3. **Functions mapping Mathematical Objects to Mathematical Objects**: 'the Euclidean distance function
    mapping two points in Euclidean space to a real number', 'the Determinant function mapping a square
    matrix to a real number', etc.

## Guidelines
1. **Precision**: When extracting definitions, relations and functions, you MUST be as precise and detailed as
    possible. For example 'congruent' is not a precise mathematical relation! You MUST specify how many
    arguments and what type of arguments the relation or function takes like 'two integers being congruent
    modulo 3', 'two triangles being congruent', 'a list of line segments being congruent to each other', etc
    .

2. **Well-Definedness**: Please **MAKE SURE** that each mathematical object, relation, and function you
    extract is well-defined, or has a conventionally well-accepted definition. For example, 'two lines being
    on opposing sides of a point' is not well-defined because a point can't partition a plane into sides.

3. **Abstractness**: The definitions, relations and functions you extract MUST be abstract i.e. not involving
    any particular objects, variables, names in the current statements. For example, 'line AB being parallel
    to line CD' is not abstract because it involves the names 'AB' and 'CD'.

4. **Naming Consistency**: When naming concepts, you MUST align with the previously extracted concepts if they
    refer to the same thing. For example, if previous concepts contain 'two lines being parallel in
    euclidean space' and the current statements mention 'parallel lines in euclidean space', you should use
    exactly the name 'two lines being parallel in euclidean space' for this concept extracted from the
    current statements. Only add new concept names if they are truly new and not covered by previously
    extracted concepts.

4. **Reasoning**: Please carefully analyze the all English statements sentence by sentence, identify all
    potential definitions, relations and functions, and double-check that they are well-defined. Provide
    your detailed step-by-step reasoning **BEFORE** outputing your final extraction answer.

## Output Format
You **MUST** provide your final extraction answer i.e. a list of extracted concepts within triple angle
    brackets and separated by semi-colons. Each concept is a short phrase containing any necessary
    characters except semi-colon!!!

<<< concept1; concept2; ... >>>

## Task Context:
```

### Previously Extracted Concepts:

{previous_concepts}

### Current English Statements:

{english_statements}

## D.1.2   STEP 2: CONCEPT FILTERING

You are an expert in mathematics, logic, programming lanaguages, and formal verification with deep knowledge of all fields of mathematics and formalization of mathematics. You are given a list of mathematical concepts in English, and the documentation of a Domain-Specific Language (DSL) for formal mathematics. {additional_specs}

I want to include and formalize some of these concepts into the current formal DSL. Your task is to filter out concepts that do NOT meet the criteria for inclusion.

## Filtering Criteria
You MUST filter out concepts that satisfy ANY of the following criteria:
1. **Duplication of Other Concepts**: The concept is a duplication of another concept. For example, the concepts 'a point being the midpoint of a line segment' and 'a point dividing a line segment into two equal parts' are duplications because they describe the exact same mathematical relationship using different wording. Please keep the one with a clearer and more conventional wording. In this case, please keep the concept 'a point being the midpoint of a line segment' and filter out the other one.

2. **Already Defined in Current DSL**: The concept has a direct corresponding formal relation/function/type already defined in the current DSL. For example, there is a concept 'two distinct points being on a line', and in the DSL there is a formal relation `distinctPointsOnLine` with the description: def distinctPointsOnLine (a b : Point) (L : Line) -- points a and b are distinct and on line L. The syntax is `distinctPointsOnLine a b L`. Then the concept 'two distinct points lies on a line' should be filtered out because it is already defined in the current DSL.

## Guidelines
1. **Conservative Filtering**: When in doubt, **KEEP** the concept. Only filter out concepts that clearly violate one of the criteria.

2. **DSL Comparison**: Carefully compare each concept against the current DSL to identify redundancy or easy expressibility. Don't filter out concepts by mistake! Note that some concepts might be very similar to a formal relation/function/type in the current DSL, but are different in some subtle aspects. For example, the concept 'two points being on a line' is very similar to the formal relation `def distinctPointsOnLine (a b : Point) (L : Line)`, but the former allows the two points to be the same point, while the latter requires the two points to be distinct. For another example, the concept 'a set of points being on the same side of a line' is very similar to the formal relation `def sameSide (a b : Point) (L : Line)`, but the latter only formalizes the concept that 'two points being on the same side of a line', while the former allows the set of points to be more than two.

3. **Reasoning**: Please carefully analyze concept by concept, compare them with other concepts and the current DSL to identify redundancy, determine which concepts should be FILTERED OUT (NOT KEEPING), and provide your detailed step-by-step reasoning **BEFORE** outputing your final filtering answer.

## Output Format
You **MUST** provide your final filtering decision as a list of concepts to FILTER OUT (NOT KEEPING) within triple angle brackets and separated by semi-colons. The name of the concepts you provide should be **EXACTLY THE SAME** as the name of the concepts in provided concept list!!!

<<< concept1; concept2; ... >>>

If there is no concept to filter out (KEEPING ALL CONCEPTS), please output an empty list: <<< >>>

## Task Context:
### **Current DSL Documentation:**

{dsl_doc}

### **Concepts for Filtering:**

{concept_list}

## D.1.3   STEP 3: CDG CONSTRUCTION

You are an expert in mathematics, logic, programming languages, and formal verification with deep knowledge of all fields of mathematics and formalization of mathematics. You are given a list of mathematical concepts in English, a list of previously analyzed concepts, and the documentation of a Domain-Specific Language (DSL) for formal mathematics. {additional_specs}

Your task is to analyze the dependency structure of each concept and determine how they can be formalized in the DSL.

## Dependency Analysis
For each concept in the list, you need to determine:

1. **Direct Expressibility**: Can this concept be directly expressed using existing formal types, relations, and functions in the current DSL? If yes, list the specific DSL elements that can express this concept.

2. **Concept Dependencies**:If not directly expressible, what other mathematical concepts (with no direct correspondents in the current DSL) does this concept depend on? List all the prerequisite concepts needed to define this concept. These dependencies should be abstract mathematical concepts, not specific instances.

3. **Formalization Status**: Based on the analysis, classify the concept as:
   - `directly_expressible`: Can be directly expressed using existing DSL elements
   - `needs_dependencies`: Requires other concepts to be defined first, but ultimately formalizable
   - `impossible`: Cannot be expressed in the current DSL and would require fundamental DSL extensions

## Mathematical Concepts
There are 3 types of mathematical concepts you can add as additional dependencies:
1. **Definitions of Mathematical Objects**: Point, Line, Integer, Real Number, Series, Group, Ring, etc.
2. **Relations between Mathematical Objects**: 'a point being on a line', 'an integer being even', 'a group being a subgroup of another group', etc.
3. **Functions mapping Mathematical Objects to Mathematical Objects**: 'the Euclidean distance function mapping two points in Euclidean space to a real number', 'the Determinant function mapping a square matrix to a real number', etc.

## Guidelines
1. **Conservative Classification**: When in doubt between 'needs_dependencies' and 'impossible', choose 'needs_dependencies'. Only mark as 'impossible' if the concept fundamentally cannot be expressed in the mathematical framework of the DSL. When in doubt between 'directly_expressible' and 'needs_dependencies', choose 'directly_expressible' since the DSL is designed to be expressive enough for these concepts and you should **TRY YOUR BEST** to find directly expressible formal correspondents for these concepts.

2. **Direct Correspondents Analysis**: Be precise about what DSL elements (types, relations, functions) can express each concept. Quote the exact DSL definitions and explain how they relate to the concept.

3. **Concept Dependencies Analysis**: First of all, **TRY YOUR BEST NOT TO** introduce new concepts unless absolutely necessary!!! If you must introduce new concepts, note that the list of concepts you are analyzing might have dependencies on each other. It's great if we can define one concept on top of another, but be careful about circular dependencies! NO TWO CONCEPTS CAN DEPEND ON EACH OTHER!!!

4. **Concept Dependencies Criteria**: When adding new mathematical concepts as dependencies, you MUST follow these criteria:
   a. **Precision**: When adding new mathematical concepts as dependencies, you MUST be as precise and detailed as possible. For example 'congruent' is not a precise mathematical relation! You MUST specify how many arguments and what type of arguments the relation or function takes like 'two integers being congruent modulo 3', 'two triangles being congruent', 'a list of line segments being congruent to each other', etc.
   b. **Well-Definedness**: Please **MAKE SURE** that each mathematical object, relation, and function you add as a dependency is well-defined, or has a conventionally well-accepted definition. For example, 'two lines being on opposing sides of a point' is not well-defined because a point can't partition a plane into sides.
   c. **Abstractness**: The definitions, relations and functions you add as a dependency MUST be abstract i.e. not involving any particular objects, variables, names in the current statements. For example, 'line AB being parallel to line CD' is not abstract because it involves the names 'AB' and 'CD'.

4. **Reasoning**: Please carefully examine concept by concept, recall the definition of each concept, analyze which other concepts are necessary to define the current concept, which of those have direct correspondents in the current DSL, and provide your detailed step-by-step reasoning **BEFORE** generating the JSON output surrounded by triple angle brackets.

## Output Format
You **MUST** provide your final analysis in JSON format within triple angle brackets. The JSON should have the following structure:

```json
{
  "concept_name_1": {
    "analysis": "detailed step-by-step reasoning for dependency analysis"
    "formal_correspondents": ["list of DSL elements that can express this concept"],
    "concept_dependencies": ["list of prerequisite concepts"],
    "status": "directly_expressible|needs_dependencies|impossible",
  },
  "concept_name_2": {
    ...
  }
}
```

<<< JSON_OUTPUT_HERE >>>

## Task Context:
### **Current DSL Documentation:**

{dsl_doc}

### **Previously Analyzed Concepts:**

{previous_analysis}

### **Concepts for Analysis:**

{concept_list}

# D.1.4 STEP 4: CONCEPT FORMALIZATION

You are an expert in mathematics, logic, programming languages, and formal verification with deep knowledge of all fields of mathematics, proof assistants like Lean, and Domain-Specific Language (DSL) design. You are given a concept dependency graph (CDG) of mathematical concepts, previous extension file(s) containing the concepts you have already implemented, and the documentation of a Domain-Specific Language (DSL) for formal mathematics.

Your task is to extend the current DSL by implementing / formalizing the provided concepts in the CDG based on the current DSL.

{additional_specs}

## Guidelines
1. **CDG Interpretation**: In the CDG, for each concept, the field 'status' indicates if we can directly formalize this concept as a relation, function, abbreviation, alias, etc. in the current DSL. If it is 'directly_expressible', then the field 'formal_correspondents' lists the specific DSL elements that can express this concept. You **MUST** first try your best to implement the concepts using these 'formal_correspondents'. However, if they contradicts any specifications given above, You should carefully use your own knowledge and judgement to determine the most appropriate way to implement / formalize each concept based on the current DSL. If the field 'status' is 'needs_dependencies', then the field 'concept_dependencies' lists the prerequisite concepts that we need to formalize first before we can formalize this concept.
2. **Faithfulness to Original Concept**: Please try your best to faithfully implement / formalize the original concept, nothing more and nothing less. For example, if the original concept is a 'a quadrilateral in 2-dimensional Euclidean space', then you should only invovle the four points and four lines that form the quadrilateral, **no other elements such as the diagonal lines**!!! Please **PAY EXTRA ATTENTION** to this!!! Such subtlety will result in entirely different formalizations.

3. **Code Reuse**: Please try your best to reuse the helper functions or formalized concepts from the previous extension file(s) to implement / formalize the current new concepts. Please don't reinvent the wheel!!!

4. **Consistency with Current DSL**: Please ensure your extension matches the current DSL in spirit and style. For example, naming, API convention, formalization choices, etc..
5. **Documentation**: Please carefully document every new type, relation, function, axiom, abbrevition, alias, notation, etc. in your implementation. Please make sure that the style and level of detail are consistent with the current DSL documentation.

6. **Reasoning**: Please carefully examine concept by concept, analyze the dependency structure of each concept, determine the best way to implement / formalize each concept based on the current DSL, and provide your detailed step-by-step reasoning **BEFORE** outputing your final implementation.

## Output Format
You don't need to re-generate the previous extension file(s), **ONLY** output your implementation of the new concepts, which will be appended to the previous extension file(s). You **MUST** provide your final implementation within triple angle brackets.
<<< your entire implementation goes here >>>

## Task Context:
### **Current DSL Documentation:**

{dsl_doc}

### **Previous Extension File(s):**

{previous_extension_files}

### **Concept Dependency Graph (CDG):**

{cdg_list}

## D.1.5   STEP 5: FORMALIZATION REFACTORING

You are an expert in mathematics, logic, programming languages, and formal verification with deep knowledge of all fields of mathematics, proof assistants like Lean, and Domain-Specific Language (DSL) design. You are given the documentation of the current Domain-Specific Language (DSL) for formal mathematics and file(s) containing extensions (new definitions, relations, functions, axioms, theorems, notations, etc.) to this DSL.

Your task is to refactor the extension file to improve its quality, maintainability, and consistency with the DSL design principles.

{additional_specs}

## Guidelines
1. **Aggressive Refactoring**: You should be bold in refactoring the extension file:
   - **Eliminate Redundancy**: If multiple definitions express the exact same mathematical concept with different levels of generality, keep only the most general and well-abstracted version.
   - **Consolidate Similar Patterns**: When you find multiple similar definitions that can be unified, remove the individual ones and create a single, parameterized abstraction
   - **Quality Over Quantity**: It's better to have fewer, well-designed abstractions than many specific, poorly abstracted ones. You are not required to preserve every single definition from the original file. Focus on creating a clean, maintainable, and mathematically sound extension.
   - **Pick Best One Only**: If there are many formal relations that are similar to each other, or can substitute each other, you should either carefully choose one of them, or merge them into a single, more general and canonical definition. Multiple similar APIs can be confusing to the user, **BE CONCISE, BE CONCISE, BE CONCISE**!!!

2. **Preserve The Necessary**: Even though you are required to refactor aggressively, you should preserve the formal concepts are not truely not in the current DSL. It's ok if they might be easy to implement using the current DSL primitives, but this will enhance the readability and further extendability of the DSL.

3. **Documentation**: Please carefully update the documentation of every relation, function, axiom, abbrevition, notation, etc. you refactored. Please make sure that the style and content are consistent with the current DSL documentation, NOT the comments in the extension file(s). If there are comments in the extension file(s) that are not consistent with the DSL documentation, please refactor them as well.

4. **Reasoning**: Please carefully analyze the extension file, identify areas for improvement according to the
      refactoring criteria, and provide your detailed step-by-step reasoning **BEFORE** outputting your
      refactored implementation.

## Output Format
You **MUST** provide your refactored Lean file within triple angle brackets. The output should be a complete,
      valid Lean file that can be appended to the DSL. Do not include any import statements - they will be
      handled separately.

<<< your refactored implementation goes here >>>

## Task Context:
### **Current DSL Documentation:**

{dsl_doc}

### **Extension File to Refactor:**

{extension_file}

### D.1.6 STEP 6: DOCUMENTATION UPDATE

You are an expert in mathematics, logic, programming languages, and formal verification with deep knowledge of
      all fields of mathematics, proof assistants like Lean, and Domain-Specific Language (DSL) design. You
      are given the documentation of the current Domain-Specific Language (DSL) for formal mathematics and
      extension file(s) containing new definitions, relations, functions, axioms, theorems, notations, etc.
      that have been added to this DSL. Your task is to generate a new DSL documentation to incorporate all
      the new elements from the extension file(s), while preserving the previous documentation intact.

{additional_specs}

## Guidelines
1. **Preserve Existing Content**: Keep all existing documentation content unchanged. This is a **MUST**.

2. **Consistency with Current Doc**: Keep the style and content consistent with the current DSL documentation.
       For example, if the current documentation explains the syntax or gives usage examples, then you should
      do the same.

3. **Reasoning**: Please carefully analyze the extension file(s), identify all new elements that need to be
      documented, determine the best way to integrate them into the existing documentation structure, and
      provide your detailed step-by-step reasoning **BEFORE** outputting the new updated DSL documentation.

## Output Format
You **MUST** provide the new DSL documentation within triple angle brackets. The output should be a complete,
      new version of the DSL documentation that includes everything: both elements from the current DSL
      documentation and new documentation for the extension file(s).

<<< your entire new DSL documentation goes here >>>

## Task Context:
### **Current DSL Documentation:**

{dsl_doc}

### **Extension File(s) to Document:**

{extension_files}

### D.2 HIERARCHICAL DECOMPOSITION

### D.2.1 LEANEUCLIDPLUS INSTRUCTIONS WITH LEARNED ABSTRACTION

You are given an English Statement of a theorem from Euclidean Geometry. Note that all points and lines
      mentioned in the statement are distinct, unless otherwise implied by some premises.

Your task is to first semi-formalize the English Statement into a json-style structure (see Guidlines #2),
      then formalize each clause in the Semi-Formalized Structure resulting in a Formalized Structure, and
      finally convert the Formalized Structure into a formal statement in Lean 4 strictly adhering to the
      following formal definitions and guidelines.

```
-- Basic Geometric Sorts --
axiom Point : Type
axiom Line : Type
axiom Circle : Type

-- Inductive Types for Geometric Entities --
inductive Angle | right | ofPoints (A B C : Point)
inductive Segment | endpoints (a b : Point)
inductive Triangle | ofPoints (a b c : Point)

-- Notations and Macros for Geometric Entities --
```

```
"|(a-b)|" means the length of the line segment (type \\mathbb{R}, not type Segment) between point a and point
      b.

"\\angle  a:b:c" means the degree of the angle (type \\mathbb{R}, not type Angle) formed by points a, b, and c
      , where b is the vertex of the angle and a and c are points respectively on the two sides of the angle.
      Add parentheses around the angle notation like "(\\angle  a:b:c)" to avoid ambiguity.

"\\perp " means the degree of the right angle (type \\mathbb{R}, not type Angle).

"\\triangle  a:b:c" means the triangle (type Triangle) formed from points a, b and c.
"Triangle.area \\triangle  a:b:c" means the area of the triangle formed by points a, b and c. Add parentheses
      around the triangle notation like "(\\triangle  a:b:c)" to avoid ambiguity.

-- Relations and Axioms for Geometric Sorts --
namespace Point
def onLine (a : Point) (L : Line) -- point a is on line L. The syntax is `a.onLine L`.

def sameSide (a b : Point) (L : Line) -- point a and b are on the same side of line L. They both are not on
      line L, but can be the same point. The syntax is `a.sameSide b L`.

def opposingSides (a b : Point) (L : Line) -- distinct point a and b are on opposite sides of line L. They
      both are not on line L. The syntax is `a.opposingSides b L`.

def onCircle (a: Point) (C: Circle) -- point a is on circle C. The syntax is `a.onCircle C`.

def insideCircle (a: Point) (C: Circle) -- point a is inside circle C. It can't be on the circle. The syntax
      is `a.insideCircle C`.

def outsideCircle (a: Point) (C: Circle)-- point a is outside circle C. It can't be on the circle. The syntax
      is `a.outsideCircle C`.

def isCentre (a: Point) (C: Circle) -- point a is on the unique center circle C. The syntax is `a.isCentre C`.

def isMidpointOf (m a b : Point) -- point m is the midpoint of segment AB iff m is between a and b (hence all
      three are distinct and collinear), and |AM| = |MB|. The syntax is `m.isMidpointOf a b`.

def isMidpointOfSegmentEndpoints (m a b : Point) (SP : Segment) -- same as `m.isMidpointOf a b` with an
      explicit segment witness. Requires `SP = Segment.endpoints a b`. The syntax is `m.
      isMidpointOfSegmentEndpoints a b SP`.

def onExtensionBeyondB (p a b : Point) -- point p lies on the extension of segment AB beyond endpoint B iff `
      between a b p`. The syntax is `p.onExtensionBeyondB a b`.

def onExtensionBeyondA (p a b : Point) -- point p lies on the extension of segment AB beyond endpoint A iff `
      between b a p`. The syntax is `p.onExtensionBeyondA a b`.
end Point

namespace Line
def intersectsLine (L M : Line) -- two lines L and M intersect at some point. They can be the same line. The
      syntax is `L.intersectsLine M`.

def intersectsCircle (L : Line) (C : Circle) -- line L and circle C intersect. The syntax is `L.
      intersectsCircle C`.
end Line

namespace Circle
def intersectsCircle (C1 C2: Circle) -- circle C1 and C2 intersect. The syntax is `C1.intersectsCircle C2`.
end Circle

namespace Triangle
def sideAB (a b : Point) (_c : Point) : Segment -- the side AB of triangle ABC as a segment, i.e. `Segment.
      endpoints a b`. The syntax is `Triangle.sideAB a b c`.

def sideBC (_a : Point) (b c : Point) : Segment -- the side BC of triangle ABC as a segment, i.e. `Segment.
      endpoints b c`. The syntax is `Triangle.sideBC a b c`.

def sideCA (a : Point) (_b : Point) (c : Point) : Segment -- the side CA of triangle ABC as a segment, i.e. `
      Segment.endpoints c a`. The syntax is `Triangle.sideCA a b c`.
end Triangle

-- Geometric Relations --
def distinctPointsOnLine (a b : Point) (L : Line) -- points a and b are distinct and on line L. The syntax is
      `distinctPointsOnLine a b L`.

def between (a b c : Point) -- mutually distinct points a, b and c are collinear and ordered. Point b is
      between point a and c. The syntax is `between a b c`.

def formTriangle (a b c : Point) (AB BC CA : Line) -- mutually distinct points a, b and c form a triangle,
      where point a and b are on line AB, point b and c are on line BC, point a and c are on line CA. The
      lines AB, BC, and CA must be distinct. Note that the order and correspondence of arguments a, b, c, AB,
      BC, CA is strictly required i.e. a and b must be on AB, b and c must be on BC, a and c must be on CA,
      and they must be passed in the exact order! The syntax is `formTriangle a b c AB BC CA`.

def formRectilinearAngle (a b c : Point) (AB BC : Line)  -- points a, b and c form a rectilinear angle, where
      b is the vertex of the angle and a and c are points respectively on the sides AB and BC. The sides AB
```

and BC can be the same line, the points a and c can be the same point, but the vertex b must be distinct from a and c. The syntax is `formRectilinearAngle a b c AB BC`.

def formParallelogram (a b c d : Point) (AB CD AC BD : Line) -- mutually distinct points a, b, d, and c (in clockwise/counterclockwise order i.e. ad is a diagonal) form a parallelogram, where points a and b are on line AB, points c and d are on line CD, and points a and c are on line AC, and points b and d are on line BD. The lines AB, CD, AC, and BD must be distinct. Note that the order and correspondence of arguments a, b, c, d, AB, CD, AC, BD is strictly required i.e. a and b must be on AB, c and d must be on CD, a and c must be on AC, b and d must be on BD, and they must be passed in the exact order! The syntax is `formParallelogram a b c d AB CD AC BD`.

def twoDistinctLinesIntersectAtPoint (L M : Line) (i : Point) -- two distinct lines L and M intersect at point i, i.e. L != M, i lies on both L and M, and `L.intersectsLine M`. The syntax is `twoDistinctLinesIntersectAtPoint L M i`.

def sequentiallyAligned (pts : List Point) -- a list of points is sequentially aligned (ordered collinear chain) iff every consecutive triple (p\_i, p\_i+\_1, p\_i\_2) satisfies `between p\_i p\_i+\_1 p\_i\_2`. No extra line or global distinctness is required beyond betweenness. The syntax is `sequentiallyAligned [p\_0, p\_1, p\_2, ...]`.

def sequentiallyAlignedThreeOrMore (pts : List Point) -- like `sequentiallyAligned`, additionally requiring the list to contain at least three points. The syntax is `sequentiallyAlignedThreeOrMore [p\_0, p\_1, p\_2, ...]`.

def twoSetsOpposingSidesOnLine (xs ys : List Point) (L : Line) -- two lists of points lie on opposite sides of line L: (1) no point from either list is on L; (2) every cross-pair (x \\in xs, y \\in ys) is on opposing sides of L. The syntax is `twoSetsOpposingSidesOnLine [x\_1, x\_2, ...] [y\_1, y\_2, ...] L`.

def supplementaryAngles (a b c d e f : Point) -- angles \\angle a:b:c and \\angle d:e:f are supplementary iff (\\angle a:b:c) + (\\angle d:e:f) = \\perp + \\perp . The syntax is `supplementaryAngles a b c d e f`.

def parallel (L M : Line) -- two lines L and M are parallel iff they are distinct and do not intersect. The syntax is `parallel L M`.

def nonCollinearPoints (a b c : Point) -- points a, b, c are non-collinear iff Triangle.area (\\triangle a:b:c) != 0. The syntax is `nonCollinearPoints a b c`.

def segmentFromVertexToOppositeSide (v s1 s2 p : Point) (SP : Segment) -- a segment SP connects vertex v to a point p on the opposite side s1s2, witnessed by `between s1 p s2`. Requires `SP = Segment.endpoints v p`. The syntax is `segmentFromVertexToOppositeSide v s1 s2 p SP`.

def congruentSegments (a b c d : Point) -- segments AB and CD are congruent iff |AB| = |CD|. The syntax is `congruentSegments a b c d`.

def perpendicularAt (L M : Line) (i a c : Point) -- lines L and M are perpendicular with right angle at witness point i using points a on L and c on M iff: (1) `L.intersectsLine M`; (2) `formRectilinearAngle a i c L M` (so a,i lie on L and i,c lie on M, with i as the vertex); (3) (\\angle a:i:c) = \\perp . The syntax is `perpendicularAt L M i a c`.

def equalLengthRatios (a b c d e f g h : Point) -- equality of two segment-length ratios: (|AB| / |CD|) = (|EF| / |GH|). This is a real-number equation; no non-zero-length guard is enforced. The syntax is `equalLengthRatios a b c d e f g h`.

def triangleAngleSum (a b c : Point) -- for non-degenerate triangle ABC (i.e. `nonCollinearPoints a b c`), the interior angle sum is \\perp + \\perp : (\\angle b:a:c) + (\\angle a:b:c) + (\\angle a:c:b) = \\perp + \\perp . The syntax is `triangleAngleSum a b c`.

def trianglesSimilar (a b c d e f : Point) -- triangles ABC and DEF are similar iff: (1) corresponding angles are equal: \\angle BAC = \\angle EDF, \\angle ABC = \\angle DEF, \\angle ACB = \\angle DFE; (2) corresponding sides are proportional: |AB|/|DE| = |BC|/|EF| and |BC|/|EF| = |CA|/|FD|. The correspondence is A<->D, B<->E, C<->F. No non-degeneracy is assumed. The syntax is `trianglesSimilar a b c d e f`.

def trianglesCongruent (a b c d e f : Point) -- triangles ABC and DEF are congruent iff: (1) corresponding angles are equal; (2) corresponding sides are equal: |AB|=|DE|, |BC|=|EF|, |CA|=|FD|. The correspondence is A<->D, B<->E, C<->F. The syntax is `trianglesCongruent a b c d e f`.

def formConvexQuadrilateral (a b c d : Point) (AB BC CD DA : Line) -- ordered points a, b, c, d with side-lines AB, BC, CD, DA form a convex quadrilateral iff: (1) consecutive vertices lie on their side lines with distinct endpoints: `distinctPointsOnLine a b AB`, `distinctPointsOnLine b c BC`, `distinctPointsOnLine c d CD`, `distinctPointsOnLine d a DA`; (2) for each side line, the two nonincident vertices lie on the same side of that line: c.sameSide d AB, d.sameSide a BC, a.sameSide b CD, b.sameSide c DA. No further distinctness, intersection, or non-collinearity is imposed. The order and correspondence are strict: AB is the side ab, BC is the side bc, CD is the side cd, DA is the side da, and they must be passed in this exact order. The syntax is `formConvexQuadrilateral a b c d AB BC CD DA`.

def diagonalACOfQuadrilateral (a b c d : Point) (SP : Segment) -- SP is the diagonal AC of the ordered quadruple (a, b, c, d), i.e. `SP = Segment.endpoints a c`. The syntax is `diagonalACOfQuadrilateral a b c d SP`.

def diagonalBDOfQuadrilateral (a b c d : Point) (SP : Segment) -- SP is the diagonal BD of the ordered quadruple (a, b, c, d), i.e. `SP = Segment.endpoints b d`. The syntax is `diagonalBDOfQuadrilateral a b c d SP`.

def segmentBisectsAngleAtVertex (a b c p : Point) (SP : Segment) (AB BC BP : Line) -- segment BP bisects angle ABC at vertex b, with: a,b on AB; b,c on BC; b,p on BP; interior witnessed by same-side constraints a.sameSide p BC and c.sameSide p AB; sub-angles equal (\\angle a:b:p) = (\\angle p:b:c); and `SP = Segment.endpoints b p`. The order is strict: AB is the side ab, BC is the side bc, BP is the carrier line of the bisector bp, and they must be passed in this exact order. The syntax is `segmentBisectsAngleAtVertex a b c p SP AB BC BP`.

```
def trianglesShareSideBy
  (sel\_1 : Point -> Point -> Point -> Segment)
  (sel\_2 : Point -> Point -> Point -> Segment)
  (a b c d e f : Point) -- triangles ABC and DEF share a side witnessed by segment selectors `sel\_1` and `sel
      \_2` iff `sel\_1 a b c = sel\_2 d e f`. Typical selectors are `Triangle.sideAB`, `Triangle.sideBC`, or
      `Triangle.sideCA`. The syntax is `trianglesShareSideBy Triangle.sideAB Triangle.sideBC a b c d e f`.

def equilateralTriangle (a b c : Point) -- triangle ABC is equilateral iff |AB| = |BC| and |BC| = |CA|. The
    syntax is `equilateralTriangle a b c`.

def isoscelesAtA (a b c : Point) -- triangle ABC is isosceles at vertex A iff |AB| = |AC|. The syntax is `
    isoscelesAtA a b c`.

def isoscelesAtB (a b c : Point) -- triangle ABC is isosceles at vertex B iff |BA| = |BC|. The syntax is `
    isoscelesAtB a b c`.

def isoscelesAtC (a b c : Point) -- triangle ABC is isosceles at vertex C iff |CA| = |CB|. The syntax is `
    isoscelesAtC a b c`.

def lineCutsTriangleOnABandACAt (a b c p q : Point) (AB BC CA L : Line) -- line L intersects sides AB and AC
    of triangle ABC at points p and q respectively, with: `formTriangle a b c AB BC CA`; betweenness `
    between a p b` and `between c q a`; and both p and q on L. The order and correspondence are strict: AB
    is the side ab, BC is the side bc, CA is the side ca, and L is the cutting line; they must be passed in
    this exact order. The syntax is `lineCutsTriangleOnABandACAt a b c p q AB BC CA L`.

def lineCutsTriangleOnABandBCAt (a b c p q : Point) (AB BC CA L : Line) -- line L intersects sides AB and BC
    at points p and q respectively, with: `formTriangle a b c AB BC CA`; betweenness `between a p b` and `
    between b q c`; and both p and q on L. The order and correspondence are strict as above. The syntax is `
    lineCutsTriangleOnABandBCAt a b c p q AB BC CA L`.

def lineCutsTriangleOnACandBCAt (a b c p q : Point) (AB BC CA L : Line) -- line L intersects sides AC and BC
    at points p and q respectively, with: `formTriangle a b c AB BC CA`; betweenness `between c p a` and `
    between b q c`; and both p and q on L. The order and correspondence are strict as above. The syntax is `
    lineCutsTriangleOnACandBCAt a b c p q AB BC CA L`.

def lineCutsABandACParallelBC (a b c p q : Point) (AB BC CA L : Line) -- line L cuts sides AB and AC at p and
    q and is parallel to side BC; requires `lineCutsTriangleOnABandACAt a b c p q AB BC CA L` and `parallel
    L BC`. The syntax is `lineCutsABandACParallelBC a b c p q AB BC CA L`.

def lineCutsABandBCParallelCA (a b c p q : Point) (AB BC CA L : Line) -- line L cuts sides AB and BC at p and
    q and is parallel to side CA; requires `lineCutsTriangleOnABandBCAt a b c p q AB BC CA L` and `parallel
    L CA`. The syntax is `lineCutsABandBCParallelCA a b c p q AB BC CA L`.

def lineCutsACandBCParallelAB (a b c p q : Point) (AB BC CA L : Line) -- line L cuts sides AC and BC at p and
    q and is parallel to side AB; requires `lineCutsTriangleOnACandBCAt a b c p q AB BC CA L` and `parallel
    L AB`. The syntax is `lineCutsACandBCParallelAB a b c p q AB BC CA L`.

-- Guidelines --
```

1. Formalized Statement Format: Your formalized statement must be of the form <<< \\forall (...), P_1 \\land
   P_2 ... \\land P_n -> Q_1 \\land Q_2 ... \\land Q_m >>> where where each P_i and Q_i is built from
   the above building blocks using conjunction (\\land ) disjunction (\\lor ) and negation (\\neg ). All
   variable declarations must be placed in parentheses after the universal quantifier. You shouldn't
   declare variables or use quantifiers in any other places! For different types of variables, you should
   put them in different parentheses. For example, if the English statement contains "The points A, B are
   on line AB", then you should declare (A B : Point) (AB : Line) in the formalized statement.

2. Semi-Formalized Structure: For the Semi-Formalized Structure, you must output a json5-style structure (you
   can interleave comments in between data) with 2 fields "premises" and "conclusions", and the value of
   each filed is a list of clauses.

```
{
    "premises": [
        English clause 1,
        English clause 2,
        ...
    ],
    "conclusions": [
        English clause 1,
        English clause 2,
        ...
    ]
}
```

Each clause is an English sentence representing a proposition. The relationship between premises and
   conclusions is implication. The premises are the antecedent and the conclusions are the consequent. The
   relationship between each clause is conjunction. You should try to make each clause as atomic as
   possible i.e. within a clause, there shouldn't be any logical operators. If there has to be some, you
   should use as few logical operators within a clause as possible, and use multiple clauses instead to
   express the same meaning.

3. Formalized Structure: The structure is exactly the same as the Semi-Formalized Structure, but you will
   formalize each English clauses in premises and conclusions into Lean expressions, to get the Formalized
   Structure.

```
{
    "premises": [
        Formalized clause 1,
        Formalized clause 2,
        ...
    ],
    "conclusions": [
```

```
        Formalized clause 1,
        Formalized clause 2,
        ...
    ]
}
```

For the formalization of each clause, you should first try to find if there is a direct corresponding formal
    relation provided. For example, the English sentence "distinct points a is between b and c" has a
    potential directly corresponding formal relation `between`. You should then double check the detailed
    description of the formal relation to see if the details match the actual English clause, and pass the
    arguments in the correct order specified in the description: `between b a c`. If no direct formalization
    is available, or the relation description does not match, you should formalize it equivalently using
    available operators, constants, and other formal relations. For example, "the points A, B, C form an
    inscribed triangle in circle \\alpha " doesn't have any direct formalization, but it's equivalent to "
    the points A, B, C form a triangle, and they are all on circle \\alpha ", which can be formalized as `
    formTriangle A B C AB BC CA \\land  A.onCircle \\alpha  \\land  B.onCircle \\alpha  \\land  C.onCircle
    \\alpha `.

4. Implication: There can be only a single implication in the formula; either side of the implication must be
    a conjunction of formulae.

5. Variable Naming: You should always use the EXACT SAME variable name in the formalized statement as the one
    in the English statement. For example, if the English statement contains "Point A, B is on line AB",
    then you should use the variable names (A B : Point) (AB : Line) in the formalized statement. **You must
    not add any prefix or suffix to the variable names like "point_A" or "AB_line"!!!** Somtimes the
    English statement might first refer to a line as "AB", but later use "BA" for the same line, in which
    case you should use the first occurring name "AB" for that same line consistently in your formalized
    statement.

6. Numeric Values Restrictions: Denote 90-degree angle by \\perp , 180-degree angle by \\perp  + \\perp , etc.
     Also, when referring to segments, we always mean its length (i.e. |(a-b)|).

7. Quantified Variables: Your quantified variables must be limited to primitive geometric types: points, lines
    , and circles. ALL bound variables that you declared must be mentioned in some clauses later.

8. Intermediate Variables: You should never define an intermediate variable inside the proposition. For
    example, "let \\alpha  := (something);" is not allowed.

9. Numeric Operators: You should only uses  addition (+), subtraction (-), multiplication (*), and division
    (/). Avoid using other mathematical operators such as exponentiation.

10. Equality Relation: You can use the equality relation (=) and non-equality relation (!=) to compare points,
     lines, circles, angles, and line segments. However, please **avoid** using the equality relation to
     compare expressions and numeric values as it's not supported by the current formal language
     implementation. For example, you should write "|(a-b)| = |(c-d)|" instead of "|(a-b)| / |(c-d)| = 1".

11. Angle Notation: When you see a short-hand angle notation like "\\angle  X", this means that X is the
     vertex of the angle. You should always expand it to the full angle notation "\\angle  A X B" (formally,
     \\angle  A:X:B) where A and B are points respectively on the two sides of the angle.

12. Syntax Tip 1: You should only use the provided relations and axioms. Please examine the provided axioms
     and relations carefully to find the best way to express your proposition. Don't create or use new ones
     that are not provided above! When you see an error message that contains "unknown identifier", it means
     that you might have used some relations that are not provided in the guidelines. Please double-check
     this, or your formalized statement won't compile!

13. Syntax Tip 2: You must declare all Points, Lines, and Circles that you will use in the formal statement.
     You shouldn't declare any extra Points, Lines, or Circles that are not used in the formalized statement.
      When you see an error message that contains "unknown identifier", it means that you might have
     forgotten to declare some Points, Lines, or Circles that are used in the formalized statement. Please
     double-check this, or your formalized statement won't compile!

14. Syntax Tip 3: You shouldn't declare any extra Points, Lines, or Circles that are not used in the
     formalized statement. When you see an error message that contains "Unexpected expression", it means that
      you might have declared some extra variables that are not used in the formalized statement. Please
     double-check this, or your formalized statement won't compile!

15. Syntax Tip 4: A formalized angle \\angle  A:B:C always expects three point identifiers separated by colons
     ! Even though you will see the English expressions like "\\angle  Y", but "\\angle  Y" is not a valid
     formalization. You should formalize it into "\\angle  X:Y:Z" where X, Y, and Z are the three points that
      form the angle. When you see an error message like "unexpected token ')'; expected ':'", it means that
     you have not provided the correct number of identifiers for the angle, or you need to surround the angle
      with parentheses like (\\angle  X:Y:Z) to avoid ambiguity. Please double-check this, or your formalized
     statement won't compile!

16. Reasoning: Please think carefully step-by-step and state your intermediate reasoning steps before you semi
     -formalize each clause when converting the English Statement to Semi-Formalized Structure, formalize
     each clause when converting the Semi-Formalized Structure into Formalized Structure, and write down the
     final Formalized Statement. **This is crucial!!! Please do not skip this!!! If there are examples
     provided by the user, you must strictly follow the format and reasoning in the examples!!!**

17. Response Format: Present your formalized Lean expression within triple angle brackets (<<< Lean expression
     here >>>). Do not add any annotations/explanations, or markdown syntax.

## D.2.2   LEANEUCLIDPLUS ONE-SHOT EXAMPLE

Here is an example:

English Statement:

```
There are two triangles \\triangle GHJ and \\triangle IFJ with a shared vertex J. The lines GI and HF
    intersect at point J. The point J lies between G and I, and also lies between H and F.

Given G J / I J = H J / F J. Complete the proof that \\triangle  G H J ~ \\triangle  I F J.

Semi-Formalized Structure:

/*
    To semi-formalize the English Statement into a structure of premises and conclusions, we will examine the
        English Statement sentence by sentence, and break down each sentence into a natural language clause
        that is as atomic as possible.
*/

{
    "premises": [
        /*
        The sentence "There are two triangles \\triangle GHJ and \\triangle IFJ with a shared vertex J." first
            declares 2 triangles \\triangle  G H J and \\triangle  I F J.

        From the instructions, we know that all points and lines are distinct, so it also declares 5 distinct
            points: H, J, G that form triangle \\triangle  G H J, and I, F, J that form triangle \\triangle
            I F J.

        We can break down the sentence into the following 2 premises:
        */
        "The distinct points G, H, and J form a triangle \\triangle  G H J",
        "The distinct points I, F, and J form a triangle \\triangle  I F J",
        /*
        For now, the premises above only partially express the fact that points G, H, J, I, F are all distinct
            , and some lines (sides of the triangles) are distinct.

        However, we might be able to incorporate and express this distinctness relation in later premises, so
            we don't create additional premises for them right now.

        Instead, we will first examine all sentences in the English statement and eventually create premises
            for the remaining unexpressed distinctness relations if necessary.
        */

        /*
        The sentence "The lines GI and HF intersect at point J." first declares 2 lines: GI, HF.

        From the instructions, we know that all points and lines are distinct, so it also declares 4 distinct
            points and 2 distinct lines: H, F on line HF, G, I on line GI.

        It then states an intersection relation: the lines GI and HF intersect at point J.

        If we don't explicitly state the distinctness of the points and lines, it might be interpreted that
            either G and I can be the same point, H and F can be the same point, or GI and HF are the same
            line, which is not what we want.

        Therefore, we can break down the sentence into the following 3 premises:
        */
        "H and F are distinct points on line HF",
        "G and I are distinct points on line GI",
        "The distinct lines GI and HF intersect at point J",
        /*
        Since we had explicitly expressed that the points G, H, J are mutually distinct, and that the points I
            , F, J are mutually distinct, we have also expressed all distinctness relations between the
            points G, H, J, I, F except for the distinctness relation between G and F, H and I.

        If G were the same as F, then GI (FI) would intersects HF at point F as well as point J, which means
            that either J = F or GI = HF, which contradicts the fact that J and F are distinct points or GI
            and HF are distinct lines. Therefore, G and F must be distinct points.

        If H were the same as I, following the exact same reasoning, we will get a contradiction. Therefore, H
            and I must be distinct points.

        Therefore, we checked that all distinctness relations among the points G, H, I, F are expressed. There
            is only one more distinctness relation left to check: the lines GH and FI are distinct.

        If GH were the same as FI, we would have the intersection of GI and HF, point J also lying on GH (FI),
            which contradicts the fact that G, H, and J (I, F, and J) form a triangle. Therefore, GH and FI
            must be distinct lines.

        Therefore, we checked that all distinctness relations among the points and lines are expressed.
        */

        /*
        The sentence "The point J lies between G and I, and also lies between H and F." states order relations
            : the point J is between G and I, and also between H and F.

        If we don't explicitly state the distinctness of the points, it might be interpreted that J can be the
            same as G or I, or J can be which is not what we want.

        Therefore, we have the following 2 premises:
        */
        "Among the distinct points G, J, I, the point J is between G and I",
        "Among the distinct points H, J, F, the point J is between H and F",

        /*
```

```
        The sentence "Given G J / I J = H J / F J." states a numeric equation between the ratios of line
                segments: G J / I J = H J / F J.

        Therefore, we have the following premise:
        */
        "The ratio of segment GJ to segment IJ equals the ratio of segment HJ to segment FJ, denoted as G J /
                I J = H J / F J",
    ],
    "conclusions": [
        /*
        The sentence "Complete the proof that \\triangle  G H J ~ \\triangle  I F J." states the goal of the
                proof: to show that the triangles \\triangle  G H J and \\triangle  I F J are similar.

        Therefore, we have the following conclusion:
        */
        "The triangles \\triangle  G H J and \\triangle  I F J are similar, denoted as \\triangle  G H J ~ \\
                triangle  I F J"
    ]
}

Formalized Structure:

/*
    To convert the Semi-Formalized Structure into a Formalized Structure, we will examine the natural language
            clauses in the Semi-Formalized Structure one by one, and formalize each into a formal clause that
         is well-formed and well-defined according to the guidelines.
*/

{
    "premises": [
        /*
        The clause "The distinct points G, H, and J form a triangle \\triangle  G H J" has a potential direct
                formalization in the guidelines: `formTriangle`.

        We double check the description of the `formTriangle` relation: "mutually distinct points a, b and c
                form a triangle, where point a and b are on line AB, point b and c are on line BC, point a and c
                 are on line CA. The lines AB, BC, and CA must be distinct".

        The mutual distinctness of points in our target clause matches the description of "mutually distinct
                points a, b and c". By definition, the 3 points that form a triangle must be non-collinear,
                which matches the description "The lines AB, BC, and CA must be distinct". Therefore, this
                relation is indeed equivalent to our target clause.

        `formTriangle` takes (a b c : Point) (AB BC CA : Line) as arguments in order, so we can formalize the
                clause as: `formTriangle G H J GH HJ JG`.

        The following clause "The distinct points I, F, and J form a triangle \\triangle  I F J" can be
                formalized in the exact same way.

        Therefore, we can formalize the first 2 clauses as follows:
        */
        "formTriangle G H J GH HJ JG",
        "formTriangle I F J IF FJ JI",

        /*
        The clause "H and F are distinct points on line HF" has a potential direct formalization in the
                guidelines: `distinctPointsOnLine`.

        We double check the description of the `distinctPointsOnLine` relation: "points a and b are distinct
                and on line L". It is indeed equivalent to our target clause.

        It takes (a b : Point) (L : Line) as arguments in order, so we can formalize the clause as: `
                distinctPointsOnLine H F HF`

        The following clause "G and I are distinct points on line GI" can be formalized in the exact same way.

        Therefore, we can formalize the next 2 clauses as follows:
        */
        "distinctPointsOnLine H F HF",
        "distinctPointsOnLine G I GI",

        /*
        The clause "The distinct lines GI and HF intersect at point J" has a potential direct formalization in
                 the guidelines: `twoDistinctLinesIntersectAtPoint`.

        We double check the description of the `twoDistinctLinesIntersectAtPoint` relation: "two distinct
                lines L and M intersect at point i, i.e. L != M, i lies on both L and M, and `L.intersectsLine M
                `". It is indeed equivalent to our target clause since it guarantees that the lines are distinct
                .

        It takes (L M : Line) (i : Point) as arguments in order, so we can formalize the clause as follows:
        */
        "twoDistinctLinesIntersectAtPoint GI HF J",

        /*
        The clause "Among the distinct points G, J, I, the point J is between G and I" has a potential direct
                formalization in the guidelines: `between`.

        We double check the description of the `between` relation: "mutually distinct points a, b and c are
                collinear and ordered. Point b is between point a and c". It is indeed equivalent to our target
                clause since it guarantees that the points are mutually distinct.
```

```
            It takes (a b c : Point) as arguments in order, with Point b being the one that is between Point a and
                Point c, so we can formalize the clause as: `between G J I`.

            The following clause "Among the distinct points H, J, F, the point J is between H and F" can be
                formalized in the exact same way.

            Therefore, we can formalize the next 2 clauses as follows:
            */
            "between G J I",
            "between H J F",

            /*
            The clause "The ratio of segment GJ to segment IJ equals the ratio of segment HJ to segment FJ,
                denoted as G J / I J = H J / F J" has a potential direct formalization in the guidelines: `
                equalLengthRatios`.

            We double check the description of the `equalLengthRatios` relation: "equality of two segment-length
                ratios: (|AB| / |CD|) = (|EF| / |GH|). This is a real-number equation; no non-zero-length guard
                is enforced". It is indeed equivalent to our target clause.

            It takes (a b c d e f g h : Point) as arguments in order. Therefore, the parameters a, b corresponds
                our concrete arguments G and J, the parameters c, d corresponds to I and J, the parameters e, f
                corresponds to H and J, and the parameters g, h corresponds to F and J, so we can formalize the
                clause as follows:
            */
            "equalLengthRatios G J I J H J F J"
        ],
        "conclusions": [
            /*
            The clause "The triangles \\triangle  G H J and \\triangle  I F J are similar, denoted as \\triangle
                G H J ~ \\triangle  I F J" has a potential direct formalization in the guidelines: `
                trianglesSimilar`.

            We double check the description of the `similar` relation: "triangles ABC and DEF are similar iff: (1)
                 corresponding angles are equal: \\angle  BAC = \\angle  EDF, \\angle  ABC = \\angle  DEF, \\
                angle  ACB = \\angle  DFE; (2) corresponding sides are proportional: |AB|/|DE| = |BC|/|EF| and |
                BC|/|EF| = |CA|/|FD|. The correspondence is A<->D, B<->E, C<->F. No non-degeneracy is assumed".
                It is indeed equivalent to our target clause.

            We check that the point G corresponds to the point I, the point H corresponds to the point F, and the
                point J corresponds to the point J.

            The `trianglesSimilar` relation takes (a b c d e f : Point) as arguments in order. Therefore, the
                parameters a, b, c corresponds our concrete arguments G, H, J, the parameters d, e, f
                corresponds to I, F, J, so we can formalize the clause as follows:
            */
            "trianglesSimilar G H J I F J"
        ]
}
```

Formalized Statement:

Before converting the Formalized Structure into a Formalized statement, we first need to declare all the
    geometric objects that are mentioned in the clauses of the Formalized Structure as variables.

According to the guidelines, since all our declared variables will be bounded by a universal quantifier \\
    forall , we shouldn't declare any extra variables that are not mentioned in any clauses of Formalized
    Structure.

We will examine the clauses in the Formalized Structure one by one, and declare variables for the mentioned
    geometric objects in order.

The first 2 clauses, `formTriangle G H J GH HJ JG`, and `formTriangle I F J IF FJ JI`, require the declaration
     of Points: G, H, J, I, F, and Lines: GH, HJ, JG, IF, FJ, JI.

The next 2 clauses, `distinctPointsOnLine H F HF`, and `distinctPointsOnLine G I GI`, requires the additional
    declaration of Lines: HF, GI.

The remaining clauses require no additional declarations.

Therefore, we need to declare all and only the following geometric objects as variables:

(G H J I F : Point) (GH HJ JG IF FJ JI HF GI : Line)

Now we can combine the declarations, premises, and conclusions to convert the Formalized Structure to a
    Formalized Statement. According to the guidelines, we will do the following:

1. Quantify the declared variables correctly.
2. Use the conjunction of all formal clauses in the premises as the antecedent.
3. Use the conjunction of all formal clauses in the conclusions as the consequent.
4. Connect the antecedent and consequent with an implication (->).
5. Wrap the entire Formalized Statement with triple angle brackets (<<< Lean expression here >>>) for parsing.

Finally, we have the Formalized Statement:

<<< \\forall  (G H J I F : Point) (GH HJ JG IF FJ JI HF GI : Line), formTriangle G H J GH HJ JG \\land
    formTriangle I F J IF FJ JI \\land  distinctPointsOnLine H F HF \\land  distinctPointsOnLine G I GI \\
    land  twoDistinctLinesIntersectAtPoint GI HF J \\land  between G J I \\land  between H J F \\land
    equalLengthRatios G J I J H J F J -> trianglesSimilar G H J I F J >>>

### D.2.3 PROOFNET-HARD INSTRUCTIONS WITH LEARNED ABSTRACTION

```
You are given an English Statement of a mathematical theorem. Target environment: Lean 4.7.0-rc2 with Mathlib4
    . Do NOT use Lean 3 or deprecated identifiers. Use only current Lean 4/Mathlib4 names and notations.

Your task is to formalize the English Statement into a formal theorem in Lean 4 using the Mathlib library and
    no other libraries, strictly adhering to the following formal definitions and guidelines.

Here are an extra set of helpers you can **DIRECTLY USE** in addition to Mathlib:

import Mathlib

/-!
This file extends Mathlib with a small, self-contained collection of reusable
definitions used in undergraduate-level formalization tasks. The design
emphasizes generality, concise naming, and consistency with Mathlib conventions.

Contents:

1. Iterated square-root map on \\mathbb{R}>=0 and a recursive sequence:
   - `nnrealSqrt2PlusSqrt` : x |->  sqrt(2 + sqrt(x)) on \\mathbb{R}>=0
   - `nnrealSqrt2`         : the nonnegative real sqrt(2)
   - `nnrealIterSqrtSeq`   : the sequence defined by s\_0 = sqrt(2), s_{n+1} = sqrt(2+sqrt(s\_n))

2. Simple sequence predicates and examples:
   - `StrictlyBoundedAboveBy` : strict upper bound predicate for sequences
   - `sqrtSuccDiff`           : the real sequence n |->  sqrt(n+1) - sqrt(n)

3. Comparability of topologies:
   - `TopologiesComparable` : t <= u \\lor  u <= t

4. Subbasis and generated topologies on \\mathbb{R}:
   - `IsSubbasisFor` : a family is a subbasis for a topology if the topology is generated from it
   - `LowerLimitSubbasis` and `sorgenfreyTopology` (lower limit / Sorgenfrey line)
   - `KNatRecip`, `KTopologySubbasis`, and `kTopology` (K-topology on \\mathbb{R})
   - `RationalIooSubbasis` and `realTopologyFromRationalIoo` (topology from rational intervals)

5. Pointwise convergence on a set and limit points:
   - `UnitInterval` : the type Icc 0 1
   - `powSeqOnUnitInterval` : the sequence of functions x |->  x^n on the unit interval
   - `powLimitOnUnitInterval` : its pointwise limit
   - `PointwiseConvergesOn` : pointwise convergence on a set
   - `IsLimitPoint` and `LimitPointCompact` : limit points and limit point compactness

6. Nested closed nonempty families:
   - `NestedClosedNonempty` : antitone closed nonempty families
   - `NestedClosedNonemptyInterNonempty` : their intersection is nonempty

All definitions are marked with `@[simp]` as requested.
-/

noncomputable section

open Real Filter Set TopologicalSpace
open scoped NNReal

/-- The function on \\mathbb{R}>=0 given by `x |->  sqrt(2 + sqrt(x))`. -/
@[simp]
def nnrealSqrt2PlusSqrt (x : \\mathbb{R}>=0) : \\mathbb{R}>=0 :=
  \\langle Real.sqrt ((2 : \\mathbb{R}) + Real.sqrt x), Real.sqrt_nonneg _\\rangle

/-- The nonnegative real number `sqrt(2)`. -/
@[simp]
def nnrealSqrt2 : \\mathbb{R}>=0 :=
  \\langle Real.sqrt (2 : \\mathbb{R}), Real.sqrt_nonneg 2\\rangle

/-- The recursively defined sequence on \\mathbb{R}>=0:
`s 0 = sqrt(2)` and `s (n+1) = sqrt(2 + sqrt(s n))`. -/
@[simp]
def nnrealIterSqrtSeq : \\mathbb{N} -> \\mathbb{R}>=0
  | 0 => nnrealSqrt2
  | n+1 => nnrealSqrt2PlusSqrt (nnrealIterSqrtSeq n)

/-- A predicate expressing that a sequence `s : \\mathbb{N} -> \\alpha ` is strictly bounded above by `c`,
i.e. `\\forall  n, s n < c`. -/
@[simp]
def StrictlyBoundedAboveBy {\\alpha  : Type*} [LT \\alpha ] (s : \\mathbb{N} -> \\alpha ) (c : \\alpha ) :
    Prop :=
  \\forall  n, s n < c

/-- The real sequence `n |->  sqrt(n + 1) - sqrt(n)`. -/
@[simp]
def sqrtSuccDiff (n : \\mathbb{N}) : \\mathbb{R} :=
  Real.sqrt ((n : \\mathbb{R}) + 1) - Real.sqrt (n : \\mathbb{R})

/-- Two topologies on the same type are comparable if one is included in the other
(i.e. `t <= u \\lor  u <= t`). -/
@[simp]
def TopologiesComparable {\\alpha  : Type*} (t u : TopologicalSpace \\alpha ) : Prop :=
```

```
  t <= u \\lor  u <= t

/-- A family `S : Set (Set X)` is a subbasis for a topology `t` on `X` if
`t` is generated from `S`. -/
@[simp]
def IsSubbasisFor {X : Type*} (S : Set (Set X)) (t : TopologicalSpace X) : Prop :=
  t = TopologicalSpace.generateFrom S

/-- The generating family for the lower limit (Sorgenfrey) topology on `\\mathbb{R}`:
all half-open intervals `[a, b)` with `a < b`. -/
@[simp]
def LowerLimitSubbasis : Set (Set \\mathbb{R}) :=
  {U | \\exists  a b : \\mathbb{R}, a < b \\land  U = Set.Ico a b}

/-- The lower limit (Sorgenfrey) topology on `\\mathbb{R}`,
generated by `{ [a, b) | a < b }`. -/
@[simp]
def sorgenfreyTopology : TopologicalSpace \\mathbb{R} :=
  TopologicalSpace.generateFrom LowerLimitSubbasis

/-- The set `K := { 1/n | n \\in  \\mathbb{N}+ } \\subseteq  \\mathbb{R}`, used in the definition of the K-
    topology. -/
@[simp]
def KNatRecip : Set \\mathbb{R} :=
  {x | \\exists  n : PNat, x = (1 : \\mathbb{R}) / ((n : \\mathbb{N}) : \\mathbb{R})}

/-- The generating family for the K-topology on `\\mathbb{R}`:
all open intervals `(a, b)` and the punctured intervals `(a, b) \ K`. -/
@[simp]
def KTopologySubbasis : Set (Set \\mathbb{R}) :=
  {U | \\exists  a b : \\mathbb{R}, a < b \\land  (U = Set.Ioo a b \\lor  U = Set.Ioo a b \ KNatRecip)}

/-- The K-topology on `\\mathbb{R}`, generated by the subbasis consisting of all sets
of the form `(a, b)` and `(a, b) \ K`, where `K = { 1/n | n \\in  \\mathbb{N}+ }`. -/
@[simp]
def kTopology : TopologicalSpace \\mathbb{R} :=
  TopologicalSpace.generateFrom KTopologySubbasis

/-- The generating family for the topology on `\\mathbb{R}` with a basis of intervals having
rational endpoints: `Ioo a b` with `a, b \\in  \\mathbb{Q}` and `a < b`. -/
@[simp]
def RationalIooSubbasis : Set (Set \\mathbb{R}) :=
  {U | \\exists  a b : \\mathbb{Q}, a < b \\land  U = Set.Ioo (a : \\mathbb{R}) (b : \\mathbb{R})}

/-- The topology on `\\mathbb{R}` generated by the subbasis of open intervals with rational endpoints. -/
@[simp]
def realTopologyFromRationalIoo : TopologicalSpace \\mathbb{R} :=
  TopologicalSpace.generateFrom RationalIooSubbasis

/-- The unit interval `[0,1]` seen as a subtype of `\\mathbb{R}`. -/
@[simp] abbrev UnitInterval : Type := Set.Icc (0 : \\mathbb{R}) 1

/-- The sequence of functions on the unit interval `[0,1]` given by
`(powSeqOnUnitInterval n) x = x^n`. -/
@[simp]
def powSeqOnUnitInterval : \\mathbb{N} -> UnitInterval -> \\mathbb{R} :=
  fun n x => (x : \\mathbb{R}) ^ n

/-- The pointwise limit function on `[0,1]` associated with the sequence `x |->  x^n`:
it is `0` for `x < 1` and `1` otherwise (in particular at `x = 1`). -/
@[simp]
def powLimitOnUnitInterval : UnitInterval -> \\mathbb{R} :=
  fun x => if (x : \\mathbb{R}) < 1 then 0 else 1

/-- Pointwise convergence on a set: a sequence of functions `F : \\mathbb{N} -> X -> \\beta ` converges
pointwise on `S` to `g : X -> \\beta ` if `\\forall  x \\in  S, Tendsto (fun n => F n x) atTop (nhds (g x))`.
    -/
@[simp]
def PointwiseConvergesOn {X \\beta  : Type*} [TopologicalSpace \\beta ]
    (F : \\mathbb{N} -> X -> \\beta ) (g : X -> \\beta ) (S : Set X) : Prop :=
  \\forall  x \\in  S, Tendsto (fun n => F n x) atTop (nhds (g x))

/-- A point `x` is a limit point (accumulation point) of a subset `A` of a topological space `X`
if every neighborhood of `x` meets `A \ {x}`. -/
@[simp]
def IsLimitPoint {X : Type*} [TopologicalSpace X] (A : Set X) (x : X) : Prop :=
  \\forall  U \\in  nhds x, (U \cap (A \ {x})).Nonempty

/-- A topological space is limit point compact if every infinite subset has a limit point. -/
@[simp]
def LimitPointCompact {X : Type*} [TopologicalSpace X] : Prop :=
  \\forall  A : Set X, A.Infinite -> \\exists  x : X, IsLimitPoint A x

/-- A nested (decreasing) sequence `C : \\mathbb{N} -> Set X` of closed, nonempty sets in a
topological space:
`Antitone C \\land  (\\forall  n, IsClosed (C n)) \\land  (\\forall  n, (C n).Nonempty)`. -/
@[simp]
def NestedClosedNonempty {X : Type*} [TopologicalSpace X] (C : \\mathbb{N} -> Set X) : Prop :=
  Antitone C \\land  (\\forall  n : \\mathbb{N}, IsClosed (C n)) \\land  (\\forall  n : \\mathbb{N}, (C n).
      Nonempty)

/-- The property that every nested (decreasing) sequence of closed nonempty subsets of `X`
```

```
has nonempty intersection `\\bigcap  n, C n`. -/
@[simp]
def NestedClosedNonemptyInterNonempty {X : Type*} [TopologicalSpace X] : Prop :=
  \\forall  (C : \\mathbb{N} -> Set X),
    Antitone C ->
    (\\forall  n : \\mathbb{N}, IsClosed (C n)) ->
    (\\forall  n : \\mathbb{N}, (C n).Nonempty) ->
    (\\bigcap  n, C n).Nonempty
```

1. Target & Environment
   - Formalize an English statement into Lean 4 using **ONLY** Mathlib
   - **NO** other libraries or import should be used.
   - Environment: Lean 4.7.0-rc2 with Mathlib4. Use only current Lean 4/Mathlib4 identifiers. Do NOT use Lean 3-era names.

2. Forbidden Lean 3-Era Identifiers & Notations
   - Identifiers: `Convergent`, `Function.IsFieldHom`, `QuotientGroup.quotient`, `Metric.bounded`, `IsPerfect`, `Real.cbrt`.
   - Notations: `\\mathbb{R}^m`, `Z[i]`, raw `\int x in \\mathbb{R}, \ldots`, binder shorthand `\\forall  x y \\in  S, \ldots`, negative exponents with `^` (e.g., `^ (-p)`).
   - Use `RingHom`/`AlgHom` (not `Function.IsFieldHom`).
   - Use `Subgroup.Quotient` (not `QuotientGroup.quotient`).
   - Use `Bornology.IsBounded E` (not `Metric.bounded E`).
   - Avoid `Real.cbrt`; use `Real.rpow` or rational exponents.

3. Header Format:
   - You need to generate **BOTH** the header and the theorem in your final response
   - The **ONLY** import statement you should generate is `import Mathlib`.
   - After that, write `open ...` and `open scoped ...` for the namespaces you need to use in the theorem.
   - **MAKE SURE** that the names and notations you are using are opened in the header!!! Please check this if you receive syntax errors!!!

4. Theorem Format: ALWAYS include theorem name, and it MUST be exactly `thm_Q`.
   - Required format: `theorem thm_Q (params) : conclusion := by sorry`
   - You **MUST** use `by sorry` as the placeholder. Do **NOT** generate a proof!!!
   - Note that some formalizations of a statement might not have parameters/binders, only the type/goal. For example, "Prove that $\sqrt[3]{2}+\sqrt[3]{3}$ is irrational." can be formalized as `theorem thm_Q : Irrational (2^((1:\\mathbb{R})/3) + 3^((1:\\mathbb{R})/3)) := by sorry`
   - You should try to use the binder/parameter form as much as possible. For example, try to convert existential statements to universal statements so that you can use the binder/parameter form.

5. Parameter Format: Use explicit parentheses for ALL parameters/binders `(\\Omega  : Set \\mathbb{C}) (f : \\mathbb{C} -> \\mathbb{C}) (h : IsOpen \\Omega )`. NEVER use implicit `{\\Omega }`.

6. Parameter/Binder Naming:
   - For object names, use the **EXACT SAME** names as the in problem text. For example, if the problem mentions "complex function f" then you should name the corresponding parameter `f` like `(f : \\mathbb{C} -> \\mathbb{C})`.
   - For hypothesis names, use the object name with a prefix `h` and index them from 1. For example, if the problem mentions "complex function f that is holomorphic on X" then you should name the corresponding hypothesis `hf_1` like `(hf_1 : DifferentiableOn \\mathbb{C} f X)`.

7. Reasoning
   - Please think carefully step-by-step and state your intermediate reasoning steps before write down the final Formalized Statement.
   - This is **EXTREMELY CRUCIAL!!!** Please do not skip this!!! **PAY EXTRA ATTENTION TO THIS!!!**
   - If there are examples provided by the user, you must strictly follow the format and reasoning in the examples!!! **THINK EXTRA HARD!!!**

8. Response Format: Return the header and the theorem together, surrounded by triple angle brackets <<< import Mathlib
open ...
open scoped ...

theorem thm_Q ... := by sorry >>>.

Do **NOT** include **ANY** comments or proof tactics in your final response!!! **PAY EXTRA ATTENTION TO THIS !!!**

Example of CORRECT output (declare all variables, name must be thm_Q):
<<< import Mathlib
open ...
open scoped ...

theorem thm_Q : \\forall  N : \\mathbb{N}, \\exists  n >= N, (3*n+1).Prime \\land  (3*n+1) >= N := by sorry >>>

Example of INCORRECT output (wrong theorem name, missing variable N declaration, includes comments, missing `by sorry`):
<<< import Mathlib
open ...
open scoped ...

-- some random comment
theorem some_random_name : \\exists  n >= N, (3*n+1).Prime \\land  (3*n+1) >= N := >>>

## D.2.4 PROOFNET-HARD ONE-SHOT EXAMPLE

```
Here is an example:

English Statement:

Let $U \subset \mathbb{C}$ be a (non-empty) connected open set and let $f_n$ be a sequence of holomorphic
    functions defined on $U$. Suppose that $f_n$ converges uniformly to a function $f$ on every compact
    subset of $U$. Show that $f$ is holomorphic in $U$.

Semi-Formalized Structure:

/*
    To semi-formalize the English Statement into a nested structure of quantifications, premises and
        conclusions, we will examine the English Statement sentence by sentence, and break down each
        sentence into natural language clauses that are as atomic as possible.
*/

{
    /*
    The sentence "Let $U \subset \mathbb{C}$ be a (non-empty) connected open set and let $f_n$ be a sequence
        of holomorphic functions defined on $U$." indicates that the choice of $U$ and $f_n$ are arbitrary,
        so we should choose the universal quantifier for the entire statement that quantifies over $U$,
        $f_n$, and possibly other variables.

    Since $f_n$ is a sequence of functions with certain properties, we will further break it into a nested sub
        -structure. Now, we will first universally quantify over $U$ as follows:
    */
    "quantification": "for all $U$, $f_n$, possibly other variables",
    "premises": [
        /*
        The sentence first declares that $U$ is a subset of the complex plane $\mathbb{C}$, that it is a non-
            empty, a connected, and an open, so we have the following 4 premises:
        */
        "$U$ is a subset of the complex plane $\mathbb{C}$",
        "$U$ is non-empty",
        "$U$ is connected",
        "$U$ is open",
        /*
        The sentence also declares that $f_n$ is a sequence of complex functions, so we have the following
            premise:
        */
        "$f_n$ is a sequence of functions from $\mathbb{C}$ to $\mathbb{C}$",
        /*
        Now we can break down the specification on $f_n$, a sequence of holomorphic functions defined on $U$.
            Recall our mathematical knowledge, a sequence is countably indexed, which means that for all $i
            \in \mathbb{N}$, $f_i$ is holomorphic on $U$.

        By the definition of holomorphic function, each $f_i$ is a function from $U$ to $\mathbb{C}$ and is
            complex-differentiable at every point of $U$.

        Therefore, we break it into a nested sub-structure by universally quantifying over $n$ as follows:
        */
        {
            "quantification": "for all $i$",
            "premises": [
                "$i$ is a natural number i.e. $i \in \mathbb{N}$",
            ],
            "conclusions": [
                "$f_i$ in the sequence $f_n$ is holomorphic on $U$, which means that $f_i$ is complex-
                    differentiable at every point of $U$"
            ]
        },
        /*
        The next sentence "Suppose that $f_n$ converges uniformly to a function $f$ on every compact subset of
            $U$" indicates that there exists a function $f$ with a certain property, which indicates a
            nested existential quantifier over $f$.

        However, our conclusion "Show that $f$ is holomorphic in $U$" requires $f$ to be declared beforehand.
            Therefore, the correct conclusion should really be "Show that if there exists a functions $f$
            such that $f_n$ converges uniformaly to $f$ on every compact subset of $U$, then $f$ is
            holomorphic in $U$".

        According to the guidelines, we should convert statement to universally quantified as much as possilbe
            so that later we can easily formalize them in the binder form. Recall our mathematical
            knowledge, if a limit exists, it is unique. Then we can rewrite the existential quantified
            conclusion as "for all $f$, such that $f_n$ converges uniformaly to $f$ on every compact subset
            of $U$, $f$ is holomorphic in $U$".

        In addition, since no other premises before depends on $f$, we can move the quantification over $f$ to
            our top-level quantifications i.e. $f$ can be quantified by the outmost universal quantifier.

        Therefore, we declare $f$ as follows:
        */
        "$f$ is a function from $\mathbb{C}$ to $\mathbb{C}$",
        /*
        Now we can break down the specification on $f$, a function that for all compact subset $X$ of $U$, the
            sequence $f_n$ converges uniformly to $f$ on $X$, which indicates a nested universal quantifier
            over $X$.

        Therefore, we break it into a nested sub-structure by universally quantifying over $X$ as follows:
        */
        {
```

```
            "quantification": "for all $X$",
            "premises": [
                "$X$ is a subset of $U$",
                "$X$ is compact"
            ],
            "conclusions": [
                "the sequence $f_n$ converges uniformly to $f$ on $X$"
            ]
        }
    ],
    "conclusions": [
        /*
        The final sentence "Show that $f$ is holomorphic in $U$" means that $f$ is complex-differentiable at
                every point of $U$. Since we have declared $f$ beforehand in the premises, we can directly refer
                 to the name $f$ in the conclusion.

        Therefore, we have the following conclusion:
        */
        "$f$ is holomorphic on $U$, which means that $f$ is complex-differentiable at every point of $U$"
    ]
}
```

```
Formalized Structure:

/*
    To convert the Semi-Formalized Structure into a Formalized Structure, we will examine the natural language
            clauses in the Semi-Formalized Structure one by one, and formalize each into a formal clause that
            is well-formed and well-defined according to the guidelines.
*/

{
    /*
    First of all, we notice that the premises and conclusions in the Semi-Formalized Structure are quantified
            by a universal quantifier over $U$. According to the guidelines, for outmost universally quantified
            statements, we should use the binder form as much as possible.

    Therefore, we put no formal quantifier over $U$ in the Formalized Structure, but instead formalize the
            premises "$U$ is subset of the complex plane $\mathbb{C}$", "$U$ is non-empty", "$U$ is connected",
            etc. as binders like `(U : some_type)`, `(hU_1 : some_property_of_U)`, etc.
    */
    "quantification": "omitted since we will use the binder form",
    "premises": [
        /*
        Now we should formalize the premises one by one. The first premise is "$U$ is a subset of the complex
                plane $\mathbb{C}$". Check the provided helpers and recall our knowledge of the required
                versions of Lean and Mathlib in the guidelines, there is no direct correspondents of "the set of
                 all complex numbers", so we need to come up with an equivalent way to formalize this premise
                alternatively.

        We notice that this premise is equivalent to just saying that $U$ is a set of complex numbers.
                Therefore, we can use the type former `Set` and the complex number type `\\mathbb{C}` to say
                that $U$ is of type `Set \\mathbb{C}`.

        Therefore, we can formalize this premise as follows:
        */
        "(U : Set \\mathbb{C})",
        /*
        The next premise is "$U$ is non-empty". Check the provided helpers and recall our knowledge of the
                required versions of Lean and Mathlib in the guidelines, there is a direct corresponding formal
                relation `Nonempty`. Therefore, we can formalize this property of $U$ as follows:
        */
        "(hU_1 : Nonempty U)",
        /*
        The next premise is "$U$ is connected". Check the provided helpers and recall our knowledge of the
                required versions of Lean and Mathlib in the guidelines, there is a direct corresponding formal
                relation `IsConnected`. Therefore, we can formalize this property of $U$ as follows:
        */
        "(hU_2 : IsConnected U)",
        /*
        The next premise is "$U$ is open". Check the provided helpers and recall our knowledge of the required
                 versions of Lean and Mathlib in the guidelines, there is a direct corresponding formal relation
                 `IsOpen`. Therefore, we can formalize this property of $U$ as follows:
        */
        "(hU_3 : IsOpen U)",
        /*
        The next premise is "$f_n$ is a sequence of complex functions". Check the provided helpers and recall
                our knowledge of the required versions of Lean and Mathlib in the guidelines, since a sequence
                is countably indexed, we can say that $f_n$ is of the curried function type `\\mathbb{N} -> \\
                mathbb{C} -> \\mathbb{C}`. When we want to talk about the $i$-th function in the sequence, we
                can use the function application notation `f_n i`.

        Therefore, we can formalize this premise as follows:
        */
        "(f_n : \\mathbb{N} -> \\mathbb{C} -> \\mathbb{C})",
        /*
        The next premise is a universally quantified statement (a sub-structure in our semi-formalized
                structure) about each $f_i$ in the sequence $f_n$: "for all $i \in \mathbb{N}$" we have "$f_i$
                in the sequence $f_n$ is holomorphic on $U$, which means that $f_i$ is complex-differentiable at
                 every point of $U$".
```

```
            For "$i$ being a natural number", check the provided helpers and recall our knowledge of the required
                    versions of Lean and Mathlib in the guidelines, we can say that $i$ is of type `\\mathbb{N}`, so
                    we can formalize the quantification over $i$ as `\\forall  i : \\mathbb{N}`.

            For "$f_i$ is complex-differentiable at every point of $U$", check the provided helpers and recall our
                    knowledge of the required versions of Lean and Mathlib in the guidelines, there is a direct
                    corresponding formal relation `DifferentiableOn` to talk about complex-differentiability by
                    passing \\mathbb{C} as the first argument.

            Therefore, we can formalize this property of $f_i$ as follows:
            */
            "(hf_n_1 : \\forall  i : \\mathbb{N}, DifferentiableOn \\mathbb{C} (f_n i) U)",
            /*
            The next premise is "$f$ is a function from $\\mathbb{C}$ to $\\mathbb{C}$". Check the provided helpers
                    and recall our knowledge of the required versions of Lean and Mathlib in the guidelines, we can
                    say that $f$ is of the function type `\\mathbb{C} -> \\mathbb{C}`.

            Therefore, we can formalize this premise as follows:
            */
            "(f : \\mathbb{C} -> \\mathbb{C})",
            /*
            The next premise is a universally quantified statement (a sub-structure in our semi-formalized
                    structure) about each subset $X$ of $U$: "for all subsets $X$ of $U$, if $X$ is compact, then
                    the sequence $f_n$ converges uniformly to $f$ on $X$".

            For "$X$ being a subset of $U$", check the provided helpers and recall our knowledge of the required
                    versions of Lean and Mathlib in the guidelines, there is a direct corresponding formal notation
                    `\\subseteq `. However, according to the guidelines, we should always declare the type of any
                    variable first. Therefore, we can formalize the quantificaiton over $X$ as `\\forall  X : Set \\
                    mathbb{C}` and the subset relation as `X \\subseteq  U`.

            For "$X$ being compact", check the provided helpers and recall our knowledge of the required versions
                    of Lean and Mathlib in the guidelines, there is a direct corresponding formal relation `
                    IsCompact`. Therefore, we can formalize this property of $X$ as follows as `IsCompact X`.

            For "the sequence $f_n$ converges uniformly to $f$ on $X$", check the provided helpers and recall our
                    knowledge of the required versions of Lean and Mathlib in the guidelines, there is a direct
                    corresponding relation `TendstoUniformlyOn F f l X` for `F : \\iota  -> \\alpha  -> \\beta ` and
                    `f : \\alpha  -> \\beta ` along a filter `l`. Here `F` is our sequence of functions `\\lambda
                    n x, f_n n x`, the limit is `f`, the index filter is `atTop` (we can directly use this name
                    since we opened the namespace `Filter`), and the set is `X`. We can formalize the conclusion of
                    this quantified statement as `TendstoUniformlyOn (\\lambda  n x => f_n n x) f atTop X`.

            Therefore, assembling every piece together, we can formalize this universally quantified statement as
                    follows:
            */
            "(hf_1 : \\forall  X : Set \\mathbb{C}, X \\subseteq  U \\land  IsCompact X -> TendstoUniformlyOn (\\
                    lambda  n x => f_n n x) f atTop X)",
    ],
    "conclusions": [
        /*
        The conclusion is "$f$ is holomorphic on $U$", which means that $f$ is complex-differentiable at every
                point of $U$". Check the provided helpers and recall our knowledge of the required versions of
                Lean and Mathlib in the guidelines, there is a direct corresponding formal relation `
                DifferentiableOn` to talk about complex-differentiability by passing \\mathbb{C} as the first
                argument.

        Therefore, we can formalize this property of $f$ as follows:
        */
        "DifferentiableOn \\mathbb{C} f U",
    ]
}
```

Formalized Statement:

Now we can convert the Formalized Structure to a Formalized Statement. According to the guidelines, we will do
        the following:

1. Assemble the quantification, premises, and conclusions properly into a Formalized Statement. Name the
        theorem following the guidelines as `thm_Q`. Use the correct syntax if we are using the binder form.
2. Wrap the entire Formalized Statement with triple angle brackets (<<< Lean expression here >>>) for parsing.

Finally, we have the Formalized Statement:

```
<<< theorem thm_Q (U : Set \\mathbb{C}) (hU_1 : Nonempty U) (hU_2 : IsConnected U) (hU_3 : IsOpen U) (f_n : \\
        mathbb{N} -> \\mathbb{C} -> \\mathbb{C}) (hf_n_1 : \\forall  n : \\mathbb{N}, DifferentiableOn \\mathbb{
        C} (f_n n) U) (f : \\mathbb{C} -> \\mathbb{C}) (hf_1 : \\forall  X : Set \\mathbb{C}, X \\subseteq  U \\
        land  IsCompact X -> TendstoUniformlyOn (\\lambda  n x => f_n n x) f atTop X) : DifferentiableOn \\
        mathbb{C} f U := by sorry >>>
```

