# OpenReview forum: "Divide and Abstract: Autoformalization via Decomposition and Abstraction Learning"
_ICLR.cc/2026/Conference — ICLR 2026 Poster_

### Official Review · Reviewer_v3kv · 2025-10-29

**Soundness:** 3
**Presentation:** 3
**Contribution:** 3
**Rating:** 6
**Confidence:** 4

**Summary:**

This paper presents Divide and Abstract (DNA), a novel zero-training framework for autoformalization that addresses the generalizability challenges in translating informal mathematics into formal languages. The framework consists of two phases: Phase I extracts common mathematical concepts from corpora to extend the target language's capabilities, while Phase II decomposes complex statements into structured clauses for systematic formalization. Evaluated on LeanEuclidPlus and ProofNet-Hard benchmarks, DNA demonstrates significant performance improvements across multiple model families, achieving up to 9.6× gains compared to baseline approaches and enabling smaller models to match larger baselines on complex mathematical statements.

**Strengths:**

1.The framework's combination of corpus-driven abstraction learning and hierarchical statement decomposition represents a novel contribution to autoformalization research, addressing fundamental limitations of existing approaches that rely on direct generation of complete formalizations.
2. The comprehensive evaluation across diverse model architectures (GPT-4.1/5, Claude-4-Sonnet, Qwen3 variants) and benchmarks demonstrates consistent performance improvements, with particularly impressive results on challenging domains where specialized models previously failed entirely.
3. The zero-training design makes DNA highly practical for low-resource domain-specific languages, eliminating the need for extensive training data while still achieving state-of-the-art performance through its innovative abstraction learning mechanism.

**Weaknesses:**

1. The six-step language extension process and four-step decomposition pipeline introduce significant complexity that may limit adoption, particularly for researchers and practitioners without extensive expertise in natural language processing or formal mathematics.​
2. While the paper demonstrates performance improvements, there is limited qualitative analysis of how the learned abstractions compare to human-crafted ones beyond basic correctness metrics, which would strengthen the validation of the framework's core innovation.

**Questions:**

The abstract mentions that code is available at https://github.com/anonymousauthor567/DivdedAndAbstract, but the repository appears to be empty. Will the code be available?

---

> ### Author Response · Authors · 2025-12-03
>
> Dear Reviewer v3kv,
>
> Thank you so much for your thoughtful and supportive assessment of our work, and for your detailed and insightful feedback. We have updated the manuscript to fix typos and improve clarity. In addition, we hope to address your concerns and questions here:
>
>
> ## Weakness 1: Complexity of the DNA Framework
>
> Thanks for such insightful feedback! The main cost of adapting our technique to new domains is writing one-shot examples. Our one-shot examples for LeanEuclidPlus and ProofNet-Hard are shown in **Appendix D.2** and **Appendix D.4**. While they are lengthy, they did not require an unreasonable amount of effort to create (it took one of the authors about two hours to construct each one).
>
> We believe that this effort is very worthwhile given the performance gains, and it is substantially less work than fine-tuning models. More importantly, as we see in **Table 5**, specialized, fine-tuned autoformalizer models completely fail on low-resource DSLs like that in **LeanEuclidPlus**.
>
> In addition to `Kimina Autoformalizer 7B` and `Mathesis Autoformalizer 7B`, we evaluated newer and larger specialized fine-tuned models on LeanEuclidPlus. `Goedel Formalizer V2 8B` and `Goedel Formalizer V2 32B` are fine-tuned based on `Qwen3`, and `StepFun Formalizer 7B` and `StepFun Formalizer 32B` are fine-tuned based on `Qwen2.5 distilled from DeepSeek R1`.
>
> We see that the fine-tuned `Goedel Formalizer V2 32B` performs way worse than the corresponding base model `Qwen3-32B` (even in the non-reasoning mode) on LeanEuclidPlus. One reason is that the fine-tuned model is so overfitted to Lean Mathlib that it has lost the general instruction-following capabilities to even output correct syntax and formatting.
>
>
> | Model | Baseline | Divide | Abstract | DNA | OracleA | DNOracleA |
> | :--- | :--- | :--- | :--- | :--- | :--- | :--- |
> | **Fine-tuned Models** | | | | | | |
> | Kimina-Autoformalizer-7B | 0.0 | 0.0 | 0.0 | 0.0 | 0.0 | 0.0 |
> | Mathesis-Autoformalizer-7B | 0.0 | 0.0 | 0.0 | 0.0 | 0.0 | 0.0 |
> | Goedel-Formalizer-V2-8B | 0.0 | 0.0 | 0.0 | 0.0 | 0.0 | 0.0 |
> | Goedel-Formalizer-V2-32B | 0.0 | 0.0 | 0.0 | 0.0 | 0.0 | 0.0 |
> | StepFun-Formalizer-7B | 0.0 | 0.0 | 0.0 | 0.0 | 0.0 | 0.0 |
> | StepFun-Formalizer-32B | 0.4 | 0.6 | **1.4** | **1.4** | 2.0 | 0.8 |
> | **Non-Reasoning Models** | | | | | | |
> | Qwen3-14B | 1.0 | 5.0 | 4.8 | **9.6** | 3.0 | 13.4 |
> | Qwen3-32B | 4.6 | 17.7 | 4.0 | **20.6** | 9.2 | 26.4 |
> | Qwen3-235B-A22B-Instruct-2507 | 40.8 | 43.6 | 45.4 | **64.4** | 55.4 | 71.4 |
> | **Reasoning Models** | | | | | | |
> | Qwen3-14B | 25.4 | 27.8 | 33.8 | **38.6** | 33.0 | 51.6 |
> | Qwen3-32B | 29.6 | 31.6 | 39.4 | **40.8** | 46.8 | 58.2 |
> | Qwen3-235B-A22B-2507-Thinking | 40.2 | 45.4 | 41.6 | **55.6** | 58.4 | 70.4 |
>
> More importantly, in **Appendix B**, we provide an ablation of the 1-shot example; it turns out that for most models (especially the larger ones with better instruction-following capabilities like ``Qwen3-235B`` and ``Claude 4 Sonnet``), even 0-shot decomposition is enough to elicit more than **2$\times$** improvement over the baselines.
>
> For the abstraction learning phase, the prompt templates we designed (included in **Appendix D.1**) are general and domain-agnostic. The only domain-specific information that is incorporated is the version of Lean to use and a 3-4 sentence description of the target domain. This requires minimal effort on behalf of users to adapt to new domains. The additional information we included can be found in our [implementation](https://github.com/anonymousauthor567/DivdedAndAbstract/blob/main/Abstraction_Learning/extend_dsl.py).
>
> ## Weakness 2: Qualitative Analysis
>
> Thanks a lot for mentioning the lack of qualitative study. This seems to be a shared concern among all reviewers and indeed a missing angle that will better illustrate the effectiveness of our framework.
>
> We have updated the manuscript to include a comprehensive Qualitative Analysis in **Appendix C** covering both LeanEuclidPlus and ProofNet-Hard. This section provides detailed examinations of each step in both the Abstraction Learning and the Hierarchical Decomposition phases.
>
> Specifically, in **Appendix C.2**, we compare the learned abstractions with the oracle human-crafted abstractions on both LeanEuclidPlus and ProofNet-Hard.

---

> ### Author Response · Authors · 2025-12-03
>
> ## Question 1: Code Availability
>
> Thank you so much for pointing that out! We accidentally included the wrong link in the manuscript, the codebase should be now available at the original (anonymous) [link](https://github.com/anonymousauthor567/DivdedAndAbstract).
>
> We plan to release the refactored versions of the 2 benchmarks and the DNA framework with more user-friendly interfaces and better documentation in the near future.
>
>
> ## Thank You Note
>
> Please let us know if the above clarification resolves your concerns. We'd be more than pleased to elaborate more. Thanks again for your time and consideration!

---

### Official Review · Reviewer_HnGH · 2025-10-31

**Soundness:** 2
**Presentation:** 3
**Contribution:** 3
**Rating:** 6
**Confidence:** 3

**Summary:**

This paper proposes Divide and Abstract (DNA), a two-stage, training-free framework for autoformalization of mathematical statements. The Divide stage first extracts common mathematical concepts from a corpus and formalizes them into reusable abstractions. The Abstract stage decomposes each statement into structured informal clauses and uses the abstractions produced above to formalize them. Concretely, the Divide stage consists of concept extraction, concept filtering, concept dependency graph construction, concept formalization, refactoring, and documentation update steps; the Abstract stage consists of decomposition, formalization, and composition steps. Evaluation on subsets of LeanEuclid and ProofNet demonstrates the effectiveness of the DNA method across many LLMs.

**Strengths:**

1. The overall approach is very intuitive, while the experimental results show that it is effective. The results in Table 5 and the analysis in Section 4.3 indicate that the Divide and Abstract stages are complementary and produce synergistic benefits; in particular, their combination appears to yield a substantial performance boost on LeanEuclidPlus.

2. The proposed method has high potential value for low-resource formal mathematical languages. Beyond low-resource languages, it should also be effective when facing rapidly evolving formal languages (for example, different versions of mathlib in Lean 4 as noted in Line 106). This aligns with the current trend of fast development in formal systems and their mathematical standard libraries.

3. The paper addresses a long-standing but thorny problem: [missing mathematical dependencies](https://github.com/jsm28/IMOLean). The lack of formalized expressions for certain mathematical concepts in libraries such as mathlib creates difficulties for formalization; this issue has been widely overlooked by many autoformalization efforts. This paper partially addresses the problem by constructing the Abstract stage. (Although the example in Figure 1 is a simple composition of concepts, fully and rigorously formalizing complex mathematical concepts and structures and feeding them back into libraries like mathlib remains beyond the reach of current models.)

**Weaknesses:**

1. The effectiveness of the Divide stage appears to be limited to problems the LLMs have not seen (e.g., certain geometry problems). From Table 5, the performance gains of Divide stage on the ProofNet-Hard dataset are marginal. Those problems largely depend on mathlib (which is likely included in LLM pretraining data).

2. The experimental baseline setup is somewhat misleading: Line 292 states that helper definitions are withheld in the baselines. Given that these Lean definitions are typically absent from training corpora and resemble a DSL, withholding helpers imposes an unnecessary handicap on the baselines and may exaggerate the contribution of Divide. A more reasonable comparison would retain the existing pipeline but add a baseline that is provided the helper definitions (without the Divide stage).

3. The method’s generalizability is questionable. The approach seems to cover a single topic well, but extending the same method to other domains or languages appears to require substantial prompt rewriting for each component of the whole pipeline, demanding significant effort and therefore limiting transferability across different DSLs or math domains.

4. Minor issues suggesting the manuscript may have been prepared in haste: the code link given on Line 29 does not contain any actual code; Line 37 mentions "three transformative applications" but does not specify what they are; and on Line 160 both quotation marks are closing quotes.

**Questions:**

1. How are the correctness and recall metrics in Table 1, Table 2, and Table 3 defined? The main text does not appear to provide definitions or evaluation protocols for these metrics—please supply precise definitions and describe how they were computed.
2. Dataset choice and subsets. The paper evaluates on subsets (LeanEuclidPlus and ProofNet-Hard) rather than the full LeanEuclid and ProofNet collections. Please clarify:
   - What specific practical difficulties may arise if one attempts to apply the DNA method directly to the full LeanEuclid and ProofNet datasets?
   - Will the selected subsets and any scripts used to produce them be released publicly to facilitate reproducibility?

---

> ### Author Response · Authors · 2025-12-03
>
> Dear Reviewer HnGH,
>
> Thank you very much for your constructive and encouraging assessment of our work, and for your detailed and valuable feedback. We have updated the manuscript to fix typos and improve clarity. In addition, we hope to address your concerns and questions here:
>
> ## Weakness 1: Effectiveness of the Divide (Decomposition) Stage
>
>
> The reviewer mentioned that
>
> > The Divide stage first extracts common mathematical concepts from a corpus and formalizes them into reusable abstractions. The Abstract stage decomposes each statement into structured informal clauses and uses the abstractions produced above to formalize them.
>
> There might be some confusion about the **Divide** and **Abstract** phases. We first clarify that:
>
> - **Abstract** is the first phase that extracts common mathematical concepts from a corpus and formalizes them into reusable abstractions.
> - **Divide** is the second phase that decomposes each statement into structured informal clauses and uses the learned abstractions produced to formalize them.
>
> Assuming that the reviewer is asking about the true **Divide** stage, we acknowledge that if we only compare the `Baseline` column and the `Divide` column for **ProofNet-Hard**, they are indeed both **0.0** for all models and thus indistinguishable.
>
> However, this is expected as **ProofNet-Hard** problems are designed to be way more challenging than the average ones. We will also explain the selection of this benchmark further when answering **Question 2: Benchmark Choices**.
>
> If we compare the `Abstract` (LLMs are provided with the learned abstractions produced by the **Abstract** stage) column and the `DNA` (learned abstractions + the hierarchical decomposition, i.e., the `Divide` phase) column, we can see that the performance gains of the **Divide** stage are significant on average, up to **7.6X** better.
>
> If we further compare the `OracleA` column (where the LLMs are provided with the oracle human-crafted abstractions), and the `DNOracleA` column (oracle human-crafted abstractions + the hierarchical decomposition, i.e., the `Divide` phase), we can see that the performance gains of the **Divide** stage are much more significant for all reasoning and non-reasoning models, up to **7X** better. The two specialized fine-tuned autoformalizer models `Kimina Autoformalizer 7B` and `Mathesis Autoformalizer 7B` didn't follow any of the instructions to perform hierarchical decomposition, resulting in `DNOracleA` being worse than `OracleA` for both models.
>
> Therefore, the results show that the **Divide** phase is effective, and it is the most effective **when we have better abstractions to work with**. As we discussed in **Section 4.3**, the DNA framework is not just a simple addition of abstraction learning and hierarchical decomposition, but a systematic framework where the **two phases strongly synergize with each other**.
>
> **Abstract** and **Divide** are the two sides of the same coin: the former performs **bottom-up formalization**, growing the library of reusable abstractions, while the latter performs **top-down decomposition**, simplifying complex statements into simpler clauses where direct translation to a learned abstraction is possible.

---

> ### Author Response · Authors · 2025-12-03
>
> ## Weakness 2: Baseline Choices
>
> Thank you for the important methodological question. There may be a misunderstanding about the **ProofNet-Hard** benchmark.
>
> In this example ground truth formalization from **ProofNet-Hard**, what we meant by helper definitions are `countably_compact` and `limit_point_compact`. These definitions are **not** from Lean Mathlib, but handwritten as part of ground truth in ProofNet:
>
> ```lean
> import Mathlib
>
> open Filter Set TopologicalSpace
> open scoped Topology
> noncomputable section
>
> def countably_compact (X : Type*) [TopologicalSpace X] :=
>   ∀ U : ℕ → Set X,
>   (∀ i, IsOpen (U i)) ∧ ((univ : Set X) ⊆ ⋃ i, U i) →
>   (∃ t : Finset ℕ, (univ : Set X) ⊆ ⋃ i ∈ t, U i)
>
> def limit_point_compact (X : Type*) [TopologicalSpace X] :=
>   ∀ U : Set X, Infinite U → ∃ x ∈ U, ClusterPt x (𝓟 U)
>
> theorem exercise_28_4 {X : Type*}
>   [TopologicalSpace X] (hT1 : T1Space X) :
>   countably_compact X ↔ limit_point_compact X :=
> sorry
> ```
>
> Even though they are provided along with import and open statements in the benchmark, they are not defined in `Mathlib`. Thus, in a real-world scenario, we can't expect human users to first provide these helper definitions to the LLMs; instead, **the LLMs must define them themselves.** Therefore, as discussed in **Section 3**, our baseline setting provides LLMs with only the import and open statements, but not the helper definitions.
>
> In the settings `Baseline` and `Divide`, the LLMs are expected to generate the formalization directly. The `Divide` setting adds hierarchical decomposition to the baseline.
>
> In the settings `Abstract` and `DNA`, the LLMs first learn a library of reusable abstractions, which is highly likely to include the ground truth helper definitions (supported by the recall results in **Table 1** and the qualitative comparison in **Appendix C.2**). The `DNA` setting, which is our main method, adds hierarchical decomposition to the abstraction learning phase.
>
> In the settings `OracleA` and `DNOracleA`, we provide the human-crafted helper definitions as oracle abstractions to the LLMs. The `DNOracleA` setting adds hierarchical decomposition to the oracle abstractions.
>
> Therefore, to isolate the effectiveness of hierarchical decomposition, i.e., Divide, a fair comparison is between `Baseline` and `Divide`, between `Abstract` and `DNA`, and between `OracleA` and `DNOracleA`. As shown in **Table 5**, the settings with hierarchical decomposition always significantly outperform the version without decomposition, **no matter whether we have no abstraction, learned abstractions, or oracle abstractions to work with**.

---

> ### Author Response · Authors · 2025-12-03
>
> ## Weakness 3: Generalizability of the DNA Framework
>
> Thanks for the insightful feedback! The main cost of adapting our technique to new domains is writing one-shot examples. Our one-shot examples for LeanEuclidPlus and ProofNet-Hard are shown in **Appendix D.2** and **Appendix D.4**. While they are lengthy, they did not require an unreasonable amount of effort to create (it took one of the authors about two hours to construct each one).
>
> We believe that this effort is very worthwhile given the performance gains, and it is substantially less work than fine-tuning models. More importantly, as we see in **Table 5**, specialized, fine-tuned autoformalizer models completely fail on low-resource DSLs like that in **LeanEuclidPlus**.
>
> In addition to `Kimina Autoformalizer 7B` and `Mathesis Autoformalizer 7B`, we evaluated newer and larger specialized fine-tuned models on LeanEuclidPlus. `Goedel Formalizer V2 8B` and `Goedel Formalizer V2 32B` are fine-tuned based on `Qwen3`, and `StepFun Formalizer 7B` and `StepFun Formalizer 32B` are fine-tuned based on `Qwen2.5 distilled from DeepSeek R1`.
>
> We see that the fine-tuned `Goedel Formalizer V2 32B` performs way worse than the corresponding base model `Qwen3-32B` (even in the non-reasoning mode) on LeanEuclidPlus. One reason is that the fine-tuned model is so overfitted to Lean Mathlib that it has lost the general instruction-following capabilities to even output correct syntax and formatting.
>
>
> | Model | Baseline | Divide | Abstract | DNA | OracleA | DNOracleA |
> | :--- | :--- | :--- | :--- | :--- | :--- | :--- |
> | **Fine-tuned Models** | | | | | | |
> | Kimina-Autoformalizer-7B | 0.0 | 0.0 | 0.0 | 0.0 | 0.0 | 0.0 |
> | Mathesis-Autoformalizer-7B | 0.0 | 0.0 | 0.0 | 0.0 | 0.0 | 0.0 |
> | Goedel-Formalizer-V2-8B | 0.0 | 0.0 | 0.0 | 0.0 | 0.0 | 0.0 |
> | Goedel-Formalizer-V2-32B | 0.0 | 0.0 | 0.0 | 0.0 | 0.0 | 0.0 |
> | StepFun-Formalizer-7B | 0.0 | 0.0 | 0.0 | 0.0 | 0.0 | 0.0 |
> | StepFun-Formalizer-32B | 0.4 | 0.6 | **1.4** | **1.4** | 2.0 | 0.8 |
> | **Non-Reasoning Models** | | | | | | |
> | Qwen3-14B | 1.0 | 5.0 | 4.8 | **9.6** | 3.0 | 13.4 |
> | Qwen3-32B | 4.6 | 17.7 | 4.0 | **20.6** | 9.2 | 26.4 |
> | Qwen3-235B-A22B-Instruct-2507 | 40.8 | 43.6 | 45.4 | **64.4** | 55.4 | 71.4 |
> | **Reasoning Models** | | | | | | |
> | Qwen3-14B | 25.4 | 27.8 | 33.8 | **38.6** | 33.0 | 51.6 |
> | Qwen3-32B | 29.6 | 31.6 | 39.4 | **40.8** | 46.8 | 58.2 |
> | Qwen3-235B-A22B-2507-Thinking | 40.2 | 45.4 | 41.6 | **55.6** | 58.4 | 70.4 |
>
>
>
> More importantly, in **Appendix B**, we provide an ablation of the 1-shot example; it turns out that for most models (especially the larger ones with better instruction-following capabilities like ``Qwen3-235B`` and ``Claude 4 Sonnet``), even 0-shot decomposition is enough to elicit more than **2$\times$** improvement over the baselines.
>
> For the abstraction learning phase, the prompt templates we designed (included in **Appendix D.1**) are general and domain-agnostic. The only domain-specific information that is incorporated is the version of Lean to use and a 3-4 sentence description of the target domain. This requires minimal effort on behalf of users to adapt to new domains. The additional information we included can be found in our [implementation](https://github.com/anonymousauthor567/DivdedAndAbstract/blob/main/Abstraction_Learning/extend_dsl.py).
>
>
>
> ## Weakness 4: Missing Sentences & Codebase Link
>
> Thank you so much for pointing that out! We have updated the manuscript to include the missing sentence after **Line 37** and the codebase should be visible now at the original (anonymous) [link](https://github.com/anonymousauthor567/DivdedAndAbstract).

---

> ### Author Response · Authors · 2025-12-03
>
> ## Question 1: Metrics in Table 1, 2, and 3
>
> Along with the qualitative analysis, we have updated the definitions and evaluation protocols for the metrics in **Tables 1, 2, and 3** in **Appendix C.1**. We provide a summary below:
>
> All metrics are averaged across 5 independent runs conditioned on the same input. All correctness and recall metrics are evaluated manually due to the freeform natural language nature of intermediate outputs and the existence of multiple semantically valid formalizations that cannot be reliably assessed by automated checkers.
>
> **Table 1 (Steps 1 & 2):**
> - **Step 1 Correctness**: The percentage of extracted concepts that satisfy the criteria of being non-ambiguous, well-defined, and abstract.
> - **Step 1 Recall**: The percentage of oracle human-expert concepts that are successfully identified and extracted by the system.
> - **Step 2 Correctness**: The percentage of filtered-out concepts that are correctly identified as duplicates or already formalized in the target formal language.
>
> **Table 2 (Steps 3 & 4):**
> - **Step 3 Correctness**: The percentage of concept dependency analyses that are correctly constructed.
> - **Step 4 Correctness**: The percentage of formal definitions that faithfully capture the intended semantics (helper definitions are counted separately).
>
> **Table 3 (Steps 5 & 6):**
> - **Step 5 Correctness**: The percentage of refactored definitions that preserve semantic correctness.
> - **Step 5 Compression Ratio**: The ratio of the number of formal definitions before refactoring to the number after refactoring, indicating the effectiveness of redundancy elimination and helper function reuse. This metric is evaluated automatically by counting the number of function definitions.
> - **Step 6 Correctness**: The percentage of documented definitions that accurately convey the formal semantics without omitting essential details or introducing ambiguity.
> - **Step 6 Downstream Compilation Rate**: A quantitative validation of documentation quality, measured by the compilation success rate of strong models (Qwen3-235B and GPT-5) when using the generated documentation for downstream formalization tasks. This metric is evaluated automatically by running the generated formalizations through the Lean compiler.

---

> ### Author Response · Authors · 2025-12-03
>
> ## Question 2: Benchmark Choices
>
> Thanks for the clarification question! As described in **Section 3**, the first benchmark **LeanEuclidPlus** is not a subset of **LeanEuclid** but an upgraded version. The following are the main changes we made:
>
> 1. Fix ambiguous, ill-defined, or under-specified natural language inputs
> 	  - ambiguous: "point A and B are closer to C and D"
>     - ill-defined: "two lines are on opposing sides of a point"
>     - under-specified: "quadrilateral" while the statement only holds for "convex quadrilateral"
>
> 2. Fix incorrect or less faithful ground truth formalizations
>     - Incorrect formalizations: wrong variable names, wrong input order, etc.
>     - Less faithful formalizations: when the problem text describes a rectangle with a named diagonal, we construct the rectangle with the diagonal instead of using two right triangles that share the hypotenuse
>
> 3. Fix ambiguous and incorrect descriptions in the language documentation
> 	  - ambiguous descriptions: missing description of whether the input points can be distinct or not
>     - operators described in the documentation but not defined in the code implementation
> 	- operators described in a different namespace than defined in the code
>
> 4. Upgrade the automated equivalence checker
>     - Add pre-checks for contradictory premises and trivially valid conclusions, so that the soundness of the checker is improved
>     - Add pre-checks for false statements, so that the completeness of the checker is improved
>     - Separately check the equivalence of the premises and the conclusions, which is easier for the SMT solver, so that the completeness of the checker is improved
>     - Fix bug in the Lean-to-SMT translation for Angle construction from 3 points
>
> 5. Removed the 48 problems from the Elements book due to the archaic language.
>
> 6. Added 40 more complex problems. As shown in **Table 6**, we use them to evaluate the generalizability of our learned abstractions on problems out of the training corpus, and the usefulness of our hierarchical decomposition on longer and harder problems.
>
> For the second benchmark **ProofNet-Hard**, it is indeed a subset. There are no theoretical or technical obstacles to evaluate on the entire **ProofNet**, but we intentionally selected the **ProofNet-Hard** problems as they are way more challenging than the average ones.
>
> For example, the natural language input "Show that the lower limit topology $\mathbb{R}_l$ and $K$-topology $\mathbb{R}_K$ are not comparable." has the following formalization:
>
> ```lean
> import Mathlib
>
> open Filter Set TopologicalSpace
> open scoped Topology
> noncomputable section
>
> def lower_limit_topology (X : Type) [Preorder X] :=
>   generateFrom {S : Set X | ∃ a b, a < b ∧ S = Ico a b}
>
> def Rl := lower_limit_topology ℝ
>
> def K : Set ℝ := {r | ∃ n : ℕ, r = 1 / n}
>
> def K_topology := generateFrom
>   ({S : Set ℝ | ∃ a b, a < b ∧ S = Ioo a b} ∪ {S : Set ℝ | ∃ a b, a < b ∧ S = Ioo a b \ K})
>
> theorem exercise_13_6 :
>   ¬ (∀ U, Rl.IsOpen U → K_topology.IsOpen U) ∧ ¬ (∀ U, K_topology.IsOpen U → Rl.IsOpen U) :=
> sorry
> ```
>
> This formalization is difficult as the LLM needs to first formalize the definitions of lower limit topology and K-topology, and then use them to formalize the statement, involving multiple layers of concept dependencies.
>
> As shown in **Table 5**, the baselines for **ProofNet-Hard** are all **0** for all models, including specialized fine-tuned models specifically trained on **ProofNet** and strong reasoning models like `GPT-5` and `Claude 4 Sonnet`. In contrast, our DNA framework is able to improve the best performing model to **15.8**, even better than the case where we have the human-crafted oracle abstractions (the OracleA column). The results quantitatively demonstrate the difficulty of **ProofNet-Hard** and the effectiveness of the DNA framework.
>
> Regarding the codebase and the benchmark, we accidentally included the wrong link in the manuscript, they should both be available now at the original (anonymous) [link](https://github.com/anonymousauthor567/DivdedAndAbstract).
>
> Please let us know if you are able to access it now.
>
> We promise to release the refactored versions of the 2 benchmarks and the DNA framework with more user-friendly interfaces and better documentation in the near future.
>
> ## Thank You Note
>
> Please let us know if the above clarification resolves your concerns. We'd be more than pleased to elaborate more. Thanks again for your time and consideration!

---

### Official Review · Reviewer_mecL · 2025-11-03

**Soundness:** 3
**Presentation:** 2
**Contribution:** 2
**Rating:** 4
**Confidence:** 4

**Summary:**

The paper “Divide and Abstract: Autoformalization via Decomposition and Abstraction Learning” introduces DNA, a zero-training framework for translating informal mathematics into formal languages. It combines abstraction learning, which extracts and formalizes reusable concepts to extend target libraries, with hierarchical decomposition, which breaks complex statements into manageable clauses for accurate formalization. Tested on LeanEuclidPlus and ProofNet-Hard, DNA achieves up to 8.6× gains over baselines and enables strong generalization to low-resource languages, showing that decomposition and abstraction jointly enhance autoformalization performance.

**Strengths:**

- Dividing a complex task like autoformalization into multiple stages is very natural, and I particularly appreciate the authors’ idea of constructing a dependency graph.
- The training-free framework is appealing and offers good portability.

**Weaknesses:**

- The core operations of 'abstraction learning' and 'hierarchical decomposition' are not rigorously formulated. For example, abstraction learning is described as extracting three types of concepts—definitions of mathematical objects, relations between objects, and functions mapping objects to objects—but the boundaries between these categories (e.g., relations vs. functions) are unclear. Moreover, the expected formal output for complex informal statements (such as “the space of continuous functions from a compact manifold to a Banach space forms a complete metric space under the supremum norm”) remains somewhat vague.
- The paper lacks a qualitative study, particularly on non-Euclidean problems. The chosen Euclidean geometry tasks appear somewhat artificial and simplistic. It would strengthen the work to validate and qualitatively justify the proposed framework on more realistic datasets such as FATE [1].

[1] https://frenzymath.com/blog/fate/

**Questions:**

- What are the algorithms for step 4 and 5? And how to interpret the results of Table 3?

---

> ### Author Response · Authors · 2025-12-03
>
> Dear Reviewer mecL,
>
> Thank you very much for your valuable and constructive feedback. We have updated the manuscript to fix typos and improve clarity. In addition, we hope to address your concerns and questions below:
>
> ## Weakness 1: Qualitative Analysis & Realistic Evaluations
>
> Thanks for pointing out the lack of qualitative study. We have updated the manuscript to include a comprehensive Qualitative Analysis in **Appendix C** covering both **LeanEuclidPlus** and **ProofNet-Hard**. This section provides detailed examinations of each step in both the Abstraction Learning and the Hierarchical Decomposition phases. **Appendix C.1** also explains in detail how the Quantitative Metrics presented in **Tables 1, 2, and 3** are designed and evaluated.
>
>
> Regarding realistic evaluations, we chose **ProofNet-Hard** in addition to **LeanEuclidPlus** because it is a realistic benchmark drawn from a wide range of undergraduate mathematics text (Real Analysis, Complex Analysis, Abstract Algebra, Topology, Putnam Competition, etc.), from which we selected the most challenging subset to rigorously evaluate our system's capabilities on non-trivial formalization tasks.
>
>
> ## Weakness 2: Clarification on Step 1 Concept Extraction of the Abstraction Learning Stage
> For the first step of abstraction learning, we prompt the LLM to extract 3 types of concepts: Objects, Relations, and Functions. These types are well-grounded in formal logic. We provide the LLM with an informal description along the following lines:
>
> * Objects: Primitives (e.g., points and lines, groups, etc.)
> * Relations: Tuples encoding relationships between objects (e.g., a point lies on a line, or one group is the subgroup of another)
> * Functions: Mappings from objects to objects (e.g., group operation or group inverse)
>
> Additional details for `Step 1: Concept Extraction` can be found in the prompt template in **Appendix D.1.1** of the updated paper.
>
> While we provide the LLM with an informal description, these ideas are standard and can be formalized as an $\mathcal{L}$-structure in mathematical logic. For example, here is a formal definition from Modern Mathematical Logic (Mileti, 2023):
>
> Definition 4.2.1 Let $\mathcal{L}$ be a language. An $\mathcal{L}$-structure, or simply a structure, is a tuple $\mathcal{M}=\left(M, g_{\text {ConSym }}, g_{\text {FuncSym }}, g_{\text {RelSym }}\right)$ where:
> - $M$ is a nonempty set, called the universe of $\mathcal{M}$.
> - $g_{\text {ConSym }}:$ ConSym $\rightarrow M$. For each $\mathrm{c} \in$ ConSym, we use $\mathrm{c}^{\mathcal{M}}$ to denote $g_{\text {ConSym }}(\mathrm{c})$.
> - $g_{\text {RelSym }}$ is a function on RelSym such that $g_{\text {RelSym }}(\mathrm{R})$ is a subset of $M^k$ for all $\mathrm{R} \in \operatorname{RelSym} m_k$. For each $\mathrm{R} \in \operatorname{RelSym}{ }_k$, we use $\mathrm{R}^{\mathcal{M}}$ to denote $g_{\text {RelSym }}(\mathrm{R})$.
> - $g_{\text {FuncSym }}$ is a function on FuncSym such that $g_{\text {FuncSym }}(\mathrm{f})$ is a $k$-ary function on $M$ for all $\mathrm{f} \in$ FuncSym ${ }_k$. For each $\mathrm{f} \in$ FuncSym ${ }_k$, we use $\mathrm{f}^{\mathcal{M}}$ to denote $g_{\text {FuncSym }}(\mathrm{f})$.
>
> The difference between relation and function is that
>
> - A relation is a subset of the universe of k-tuples $M^k$. For a certain k-tuple of objects, the relation holds if and only if the k-tuple is in the subset, i.e., the formula R(k-tuple) will be evaluated/interpreted as a truth value.
> - A (k-ary) function is a mapping from the universe of k-tuples $M^k$ to the universe of objects $M$. The formula f(k-tuple) will be evaluated/interpreted as an object, instead of a truth value.
>
>
> ## Weakness 3: Clarification on Formal Output for Complex Informal Statements
>
> We are not sure if by "formal output for complex informal statement", the reviewer is referring to the final formal statement or the intermediate decomposed state like the semi-formalized structure.
>
> A Lean formalization of “the space of continuous functions from a compact manifold to a Banach space forms a complete metric space under the supremum norm” is as follows:
>
> ```lean
> import Mathlib.Analysis.NormedSpace.Basic
> import Mathlib.Topology.ContinuousFunction.Compact
> import Mathlib.Topology.UniformSpace.CompactConvergence
>
> open scoped Topology
>
> theorem continuousMap_completeSpace_supNorm_of_compact_manifold
>     (𝕜 X E : Type*) {H : Type*} [NormedField 𝕜] [TopologicalSpace H]
>     [TopologicalSpace X] [ChartedSpace H X] [CompactSpace X] [NormedAddCommGroup E]
>     [NormedSpace 𝕜 E] [CompleteSpace E] : CompleteSpace C(X, E) := by
>   sorry
> ```
>
> In **Lines 97-105**, we use this statement as an example to motivate the need for decomposition. We are happy to further clarify if the reviewer was asking for something else.

---

> ### Author Response · Authors · 2025-12-03
>
> ## Question 1: Algorithm for Step 4 and 5 of the Abstraction Learning Stage
>
> Thanks for the clarification question.
>
> For `Step 4: Concept Formalization`, we prompt the LLM to generate the formalization of each concept in the Concept Dependency Graph (CDG) constructed in step 3 by batches of 5. For the formalization of each batch, the LLM has up to 10 attempts to fix compilation errors.
>
> For `Step 5: Formalization Refactoring`, we prompt the LLM to refactor the entire extension Lean file to improve its quality, maintainability, and consistency with the DSL design principles. The LLM has up to 10 attempts to fix compilation errors.
>
> The prompt template for ``Step 4: Concept Formalization`` is at **Appendix D.1.4**, and the prompt template for ``Step 5: Formalization Refactoring`` is at **Appendix D.1.5**. The code implementing steps 4 and 5 is available at this anonymous [link](https://github.com/anonymousauthor567/DivdedAndAbstract/blob/main/Abstraction_Learning/extend_dsl.py).
>
> ## Question 2: Interpretation of the Results of Table 3
>
> In **Table 1, 2, and 3**, we are evaluating the robustness of each step in the Abstraction Learning phase by sampling 5 outputs for a particular input. We have included detailed descriptions of these metrics in **Appendix C.1**.
>
> Specifically, in **Table 3**, we are evaluating ``Step 5: Formalization Refactoring`` and ``Step 6: Documentation Update``.
>
> For ``Step 5``, the correctness metric measures the proportion of refactored formal definitions that are correct, with helper definitions counted separately (essentially counting the number of function definitions), averaged across 5 runs.
>
> The correctness metric is evaluated manually despite the output being in formal language. For formalization, correctness is about alignment between the natural language concept and the formal statement, which requires human judgement beyond well-defined equivalence (e.g. logical equivalence, definitional equivalence, definitional equivalence with certain extensions, etc.). This cannot be evaluated automatically using a symbolic checker due to the lack of ground truth formalization, which necessitates manual evaluation by human experts.
>
> Additionally, we quantitatively analyze the compression ratio, which is the ratio of the number of formal definitions before refactoring to the number after refactoring, averaged across 5 runs. This compression ratio metric is important because it indicates how effectively common helper functions are reused and how many unnecessary formal definitions are eliminated. A high compression ratio facilitates better understanding and utilization of the learned abstractions by both humans and language models, as it reduces cognitive load and clarifies the essential conceptual structure.
>
>
>
> For ``Step 6``, the correctness metric measures the proportion of documented formal definitions that correctly convey the meaning of the definition without missing essential details or introducing ambiguity (using manual evaluation for the same reason as above), averaged across 5 runs.
>
> To further validate the quality of the generated documentation, we also evaluated downstream task performance on LeanEuclidPlus using two strong models, ``Qwen3-235B`` and ``GPT-5``, with our four-stage hierarchical decomposition pipeline. Under the hypothesis that good documentation should enable strong models to generate syntactically correct formalizations with high reliability, we report the compilation rate of the two models averaged across 5 runs.
>
> To further demonstrate the stability and robustness of this step, we also report detailed statistics across runs:
>
> | Model | Maximum | Minimum | Mean | Standard Deviation |
> |-------|---------|---------|------|-------------------|
> | Qwen3-235B | 98.2% | 96.8% | 97.4% | 0.47% |
> | GPT-5 | 100.0% | 99.0% | 99.8% | 0.45% |
>
> These consistently high compilation rates with low variance validate the quality and stability of the generated documentation.
>
> The correctness metric is evaluated manually for the same reason as Steps 1, 2, and 3, since the output consists of freeform natural language descriptions. The downstream task performance metrics are evaluated automatically using the same evaluation pipeline as the main experiments on LeanEuclidPlus.
>
>
> ## Thank You Note
>
> Please let us know if the above clarification resolves your concerns. We'd be more than pleased to elaborate more. Thanks again for your time and consideration!

---

### Author Response · Authors · 2025-12-03
**Summary of the Discussion Stage**

Dear Area Chairs and Senior Area Chairs,

We thank all reviewers for their valuable feedback. For your reference, we briefly summarize the major concerns we have addressed in the discussion stage, and the values we hope to provide to the entire ICLR and AI for Math research community via this paper.

## Addressed Concerns

1. **Qualitative Analysis:** This concern was shared by all 3 reviewers. We have added a comprehensive qualitative analysis in **Appendix C**, covering both LeanEuclidPlus and ProofNet-Hard. This includes detailed step-by-step examinations of the Abstraction Learning phase (**Appendix C.1**), comparison between learned and oracle abstractions (**Appendix C.2**), and a qualitative example demonstrating the effectiveness and efficiency of the Hierarchical Decomposition phase (**Appendix C.3**).

2. **Metric Definitions in Tables 1, 2, and 3:** This concern was shared by Reviewer mecL and Reviewer HnGH. We have provided detailed definitions and evaluation protocols for correctness, recall, and compression ratio metrics in **Appendix C.1**, clarifying how each metric is defined, calculated, and evaluated.

3. **Generalizability of the DNA Framework:** This concern was shared by Reviewer HnGH and Reviewer v3kv. We demonstrated that the main adaptation cost is writing one-shot examples (~2 hours per domain), which is substantially less than fine-tuning. The ablation in **Appendix B** shows that even 0-shot decomposition yields >2× improvement for strong models. Most importantly, we provide extra quantitative results to show the **complete failure of specialized fine-tuned models when adapting to low-resource DSLs**. In addition, our prompt templates in **Appendix D** are domain-agnostic, requiring minimal and optional domain-specific information.

4. **Choices of Benchmarks & Baselines:** These questions were raised by Reviewer HnGH. In our response, we have clarified the choices of benchmarks and baselines, and why they support our claims about the effectiveness of Abstraction Learning, the effectiveness of Hierarchical Decomposition, and most importantly **the synergy effect** of combining them.

5. **Code Availability:** This question was raised by all 3 reviewers. The codebase is now available at the [anonymous link](https://github.com/anonymousauthor567/DivdedAndAbstract) provided in the manuscript.


## Values to Research Community

1. **Addressing the Generalizability Gap:** We identify and tackle 3 fundamental generalizability challenges in existing autoformalization approaches:
    - **Bottleneck by Existing Libraries:** Mentioned as a strength by Reviewer HnGH, our Abstraction Learning phase learns reusable formal abstractions from the target natural language corpus, which can also be reused for problems outside the corpus (**Section 4.2**).
    - **Long Complex Statements:** Mentioned as a strength by Reviewer mecL, our Hierarchical Decomposition phase decomposes complex statements into smaller, more manageable sub-statements, which can be formalized more easily (**Section 4.2**).
    - **Transferability to Different Domains & Languages:** Mentioned as a strength by all 3 reviewers, our DNA framework is designed to be domain-agnostic and language-agnostic since it requires zero training, which makes it portable to different formal languages and domains with minimal effort. As shown in our additional experiments, the fine-tuned model `Goedel-Formalizer-V2-32B` fails to even output correct formatting and thus gets all **0.0** on LeanEuclidPlus, while its base model `Qwen3-32B` performs much better (**20.6** in non-reasoning mode, and **40.8** in reasoning mode).

2. **Synergistic Framework Design:** Mentioned as a strength by Reviewer HnGH and Reviewer v3kv, we demonstrate that **Abstraction Learning (Bottom-Up Formalization)** and **Hierarchical Decomposition (Top-Down Formalization)** are not independent techniques but **two sides of the same coin**—they work best when combined. DNA consistently outperforms both components in isolation, and remarkably often surpasses human-crafted oracle abstractions alone, validating our theoretical framework.

3. **Unlocking Potential of Smaller Models:** The DNA framework provides the most dramatic improvements for smaller, non-reasoning models (e.g., `Qwen3-14B`: 1.0 → 9.6, an **8.6×** gain; `GPT-4.1 mini`: 4.8 → 42.4, a **7.83×** gain), democratizing access to effective autoformalization without requiring expensive fine-tuning or frontier model APIs.

4. **Benchmark & Evaluation Contributions:** We release LeanEuclidPlus, an upgraded benchmark with fixed ambiguities, corrected ground truths, improved documentation, and an enhanced equivalence checker; and ProofNet-Hard, a challenging subset that exposes the limitations of current approaches (all baselines achieve 0% success).

---

### Meta-Review · Area_Chair_p9MV · 2026-01-06

**Summary:**

This paper introduces Divide and Abstract (DNA), a zero-training framework with two phases for autoformalization, to address three challenges: bottleneck by existing abstractions, difficulty of complex statements, and poor transferability across languages. DNA first extracts common mathematical concepts and formalizes them as reusable abstractions, then hierarchically decomposes each statement into informal clauses to conduct autoformalization and composition. In experiments, the framework is applied to Kimina Autoformalizer, Mathesis Autoformalizer, GPT, Claud, and Qwen families and evaluated on LeanEuclidPlus and ProofNet-Hard.

**Reviewer Concerns:**

Addressed reviewer concerns:
* Rigorous formulation of concepts "abstraction learning" and "hierarchical decomposition" (Reviewer mecL): Added explanations.
* Quality study (Reviewers mecL, v3kv): Added quality study in Appendix C.
* Benchmark choices (Reviewer v3kv): With detailed explanation.


Insufficiently addressed reviewer concerns:
* The performance gains on the ProofNet-Hard are marginal (HnGH)
* Complexity of the framework (Reviewer v3kv)
* Generalizability of the DNA Framework (Reviewers HnGH, v3kv)

**Reviewer Scores:**

Reviewer mecL's concerns are resolved, which may result in raising the score. For Reviewers HnGH and v3kv, some of their questions are answered, while some remaining concerns (as listed above) are partially addressed or less convincing. Overall, the final scores may justify a positive outcome.

---

### Decision · Program_Chairs · 2026-01-26

Accept (Poster)